

**The underappreciated impact of emission source profiles on the simulation of**
**PM2.5 components: New evidence from sensitivity analysis**
Zhongwei Luo[a,b,1], Yan Han[a,b,c,1], Kun Hua[a,b], Yufen Zhang[a,b*], Jianhui Wu[a,b], Xiaohui
Bi[a,b], Qili Dai[a,b], Baoshuang Liu[a,b], Yang Chen[c], Xin Long[c], Yinchang Feng[a,b*]
[a]State Environmental Protection Key Laboratory of Urban Ambient Air Particulate
Matter Pollution Prevention and Control & Tianjin Key Laboratory of Urban
Transport Emission Research, College of Environmental Science and Engineering,
Nankai University, Tianjin 300350, China.
[b]CMA-NKU Cooperative Laboratory for Atmospheric Environment-Health Research,
Tianjin 300350, China.
[c]Research Center for Atmospheric Environment, Chongqing Institute of Green and
Intelligent Technology, Chinese Academy of Sciences, Chongqing 400714, China.
*Corresponding authors:
Y. F. Zhang (zhafox@nankai.edu.cn). And Y. C. Feng (fengyc@nankai.edu.cn).
[1]Z. W. Luo and Y. Han equally contribute to this work



## Abstract

The chemical transport model (CTM) is an essential tool for air quality prediction and management, widely used in air pollution control and health risk assessment. However, the current models do not perform very well in simulating $PM_{2.5}$ components. Studies suggested that the uncertainties of model chemical mechanism, source emission inventory and meteorological field can cause inaccurate simulation results. Still, the emission source profile of $PM_{2.5}$ has not been fully taken into account in current numerical simulation. This study aims to answer (1) Whether the variation of source profile adopted in chemical transport models (CTMs) has an impact on the simulation of $PM_{2.5}$ chemical components? (2) How much does it impact? (3) How does the impact work? Based on the characteristics and variation rules of chemical components in typical $PM_{2.5}$ sources, different simulation scenarios were designed and the sensitivity of components simulation results to $PM_{2.5}$ sources profile was explored. Our findings showed that the influence of source profile changes on simulated $PM_{2.5}$ concentration was insignificant, but its impact on $PM_{2.5}$ components could not be ignored. The variations of simulated components ranged from 8% to 167% under selected different source profiles, and simulation results of some components were sensitive to the adopted $PM_{2.5}$ source profile in CTMs. These influences are connected to the chemical mechanisms of the model since the variation of species allocations in emission sources directly affected the thermodynamic equilibrium system. We also found that the perturbation of the $PM_{2.5}$ source profile caused the variation of simulated gaseous pollutants, which indirectly indicated that the perturbation of the source profile affected the simulation of secondary $PM_{2.5}$ components. Given the vital role of air quality simulation in environment management and health risk assessment, the representativeness and timeliness of source profile should be considered.

## Keywords

$PM_{2.5}$; source profile; component; numerical simulation; chemical transport model



**1. Introduction**

Ambient fine particulate matter ($PM_{2.5}$) pollution in some key regions of China has attracted much attention (Liang et al., 2020; Huang et al., 2021). The chemical components of $PM_{2.5}$, including elements (Al, Si, Fe, Mn, Ti, Cu, Zn, Pb, etc.), water-soluble ions ($SO_4^{2-}$, $NO_3^-$, $Cl^-$, $F^-$, $NH_4^+$, $Na^+$, $K^+$, $Mg^{2+}$, $Ca^{2+}$, etc.), and carbon-containing components (Organic Carbon, OC; Elemental Carbon, EC) (Yang et al., 2011; Li et al., 2013), have different physical and chemical properties, such as reactivity, thermal stability, particle size distribution, residence time, optical properties, health hazards, etc (Seinfeld and Pandis, 2006; Tang et al., 2006). According to long-term monitoring results, in most regions of China, $SO_4^{2-}$, $NO_3^-$, $NH_4^+$ and OC are the most important species in ambient $PM_{2.5}$ (Li et al., 2017a; Li et al., 2021), which has a certain adverse impact on human health (Shi et al., 2018) and ecosystem (Han et al., 2019; Zhou et al., 2018), such as acid rain in southwest China (Han et al., 2019), food security (Zhou et al., 2018), etc.

The chemical transport models (CTMs) play an important role in policy making for regulatory purposes. Based on the scientific understanding of atmospheric physical and chemical processes, CTMs are built to simulate the transport, reaction and removal of pollutants on a certain scale in horizontal and vertical directions. With the development of CTMs, the simulation accuracy of $PM_{2.5}$ concentration has been significantly improved. Higher requirements have been put forward for the precise simulation of $PM_{2.5}$ components so as to provide support for the use of CTMs in human health risk assessment, climate effects, pollution sources apportionment, and so on (Peterson et al., 2020; Lv et al., 2021). However, the current models perform not very well in simulating some components (for example, $PM_{2.5}$-bound sulfate, nitrate, ammonium, trace elements, etc.) (Zheng et al., 2015; Fu et al., 2016; Ying et al., 2018; Cao et al., 2021). In the current literature, the correlation coefficient (R) and normalized mean bias (NMB) are highly variable and inconsistent between the simulated and the observed values (listed in Table S1). This is mainly attributable to the uncertainties of



model chemical mechanism, source emission inventory and meteorological field
simulation.
The chemical mechanisms involved in CTMs are derived from parameterized
assumptions based on laboratory simulation and field observations. The actual
atmospheric chemical processes are very complex, and some reaction mechanisms are
still limitedly understood. In addition, the integration of chemical reactions and
simplified treatment methods in the model cannot fully reflect the correlation among
atmospheric pollutants. For example, in some model mechanisms, other important
sulfate and nitrate formation pathways were added through new heterogeneous
chemistry, including the chemical reaction between $SO_2$ and aerosol, $NO_2/NO_3/N_2O_3$
and aerosol (Zheng et al., 2015), nitrous acid oxidized $SO_2$ to produce sulfate (Zheng
et al., 2020), dust particles promoted the oxidation of $SO_2$ (Yu et al., 2020), modified
the uptake coefficients for heterogeneous oxidation of $SO_2$ to sulfate (Zhang et al.,
2019), updated the heterogeneous $N_2O_5$ parameterization (Foley et al., 2010). Even
though the aforementioned processes can significantly improve the simulation of $SO_4^{2-}$
and $NO_3^-$, there is still a gap between the modeled and the actual atmospheric chemical
processes.
The uncertainty of source emission inventory also significantly affects the
simulation results of $PM_{2.5}$ components (Shi et al., 2017; Sha et al., 2019). Due to
incomplete information or insufficient representativeness, pollutant emissions are
sometimes overestimated or underestimated, and the method for temporal and spatial
allocation also needs to be improved.
The uncertainty of meteorological field simulation is another crucial reason for the
simulation deviation, especially on heavy pollution days, the variation trends of $PM_{2.5}$
chemical components were not well-captured (Ying et al., 2018; Qi et al., 2019; Wang
et al., 2022). Precipitation is the key meteorological factor determining wet removal of
pollutants; boundary layer height and wind speed are the main factors affecting
convection and transport of pollutants; solar radiation, temperature and relative
humidity are the key factors affecting the formation of secondary particles (Huang et



al., 2019; Chen et al., 2020). Some literature reported that deviation from precipitation
and wind field simulation might lead to underestimation of $SO_4^{2-}$, $NO_3^-$ and $NH_4^+$
(Cheng et al., 2015; Zhang et al., 2017). Devaluation of liquid water path and cloud
cover cause a decrease of sulfate formation in cloud, and ultimately results in
significantly underestimated components in simulation values (Sha et al., 2019; Foley
et al., 2010). Underestimation of temperature and relative humidity may also cause
adverse effects of temperature- and/or relative humidity-dependence chemical reaction
in the simulation (Sha et al., 2019).

In particular, the emission source profile of $PM_{2.5}$ (Hereinafter referred to as

"source profile") has not been fully taken into account in the current numerical
simulation by CTMs. In the reported literature, $PM_{2.5}$ species allocation coefficients of
emission sources are commonly treated in the following ways: (1) allocated $PM_{2.5}$
components of source emissions by referring to source profile data in published
literature or database like the US SPECIATE (Fu et al., 2013; Wang et al., 2014; Ying
et al., 2018); (2) chemical profiles come from local measurement (Fu et al., 2013; Appel
et al., 2013). However, with the development of production technology and the
innovation of pollution treatment technology in recent years, some source profiles have
changed dramatically (Bi et al., 2019), such as $SO_4^{2-}$ from coal burning, $SO_4^{2-}$ content
in $PM_{2.5}$ is generally low in coal-fired power plant without desulfurizing facilities, while
existing coal-fired power plants using limestone/gypsum wet desulphurization, the
contents of $SO_4^{2-}$ in $PM_{2.5}$ are significantly higher than that without desulfurization
facilities (Zhang et al., 2020). The timeliness of $PM_{2.5}$ species allocation coefficients in
current CTMs also needs to be considered.

This paper attempts to answer the following questions: (1) Whether the variation

of the source profile adopted in the air quality model has an impact on the simulated
results of $PM_{2.5}$ chemical components? (2) How much does it impact? (3) How does
the impact work? Aiming at these problems above, chemical composition and its
variation law for typical $PM_{2.5}$ emission sources are summarized, on this basis,
sensitivity tests are designed to identify whether $PM_{2.5}$ source profiles and species





allocation in the model are important parameters that affect the simulation results of
chemical components in $PM_{2.5}$. The aim of this study is to provide support for the
effective utilization of source profiles in the CTMs and improvement of the simulation
schemes.

**2. Model and Data**

**2.1 Model configuration**

Weather Research and Forecasting model (WRF-3.7.1), the widely used
Community Multiscale Air Quality model (CMAQv5.0.2), and Multi-resolution
Emission Inventory for China (MEICv1.3) have been used in this study. MEIC provided
the emission inventory which is developed by Tsinghua University, mainly tracked
anthropogenic emissions in China including coal-fired power plants, industry, vehicles,
residents and agriculture (http://meicmodel.org/?page_id=135) (Li et al., 2017b; Zheng
et al., 2018). The WRF model was used to generate meteorological inputs for the
CMAQ model. Three nested modeling domains consisting of 36 km×36 km (Dom1),
12 km×12km (Dom2), and 4 km×4km (Dom3) horizontal grid sizes were set, as shown
in Fig. 1. The initial and boundary conditions for WRF were based on the North
American Regional Reanalysis data archived at National Center for Atmospheric
Research (NCAR). In addition, surface and upper air observations obtained from
NCAR were used to further refine the analysis data. The major configurations we used
in CMAQ were illuminated as follows: Gas-phase chemistry was based on the CB05
mechanism and the aerosol dynamics/chemistry was based on the aero6 module
(cb05tucl_ae6_aq). The detailed model configurations were shown in Table S2.



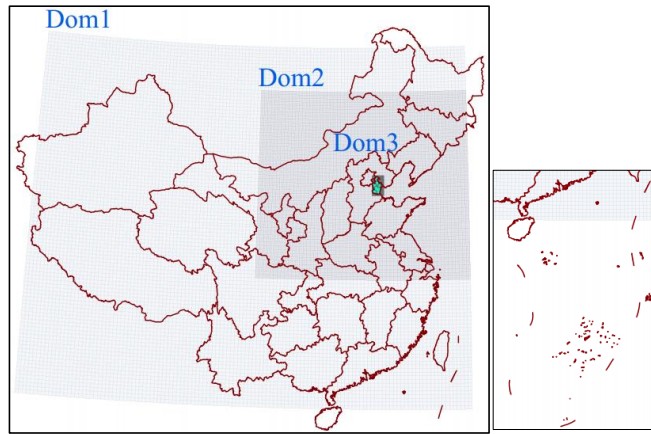

Fig.1 Modeling domains of the CMAQ model

**2.2 Selection and comparison of PM$_{2.5}$ source profile**

The PM$_{2.5}$ emission source profiles from database of Source Profiles of Air Pollution (SPAP) (http://www.nkspap.com:9091/), U.S. Environmental Protection Agency's (EPA) SPECIATE database (https://www.epa.gov/air-emissions-modeling/speciate) as well as from published literature were selected, respectively. The SPAP was developed by the State Environment Protection Key Laboratory of Urban Particulate Air Pollution Prevention, Nankai University, China. This database contains more than 3000 size-resolved source profiles of stationary combustion sources, industrial processes, vehicle exhaust, biomass burning, dust and cooking emissions and other sources, collected from more than 40 cities in China since 2001. In addition to inorganic elements, water-soluble ions, OC, EC and other conventional components, some source profiles also encompass a series of tracer information, such as organic markers, isotopes, single particle mass spectrometry, VOCs and other gaseous precursors. Based on species in the aerosol chemical mechanism (AERO6) (Appel et al., 2013; Chapel Hill, 2012), we selected 15 components in PM$_{2.5}$ source profiles including Al, Ca, Cl, EC, Fe, K, Mg, Mn, Na, OC, Si, Ti, NH$_4^+$, NO$_3^-$ and SO$_4^{2-}$, the remaining components are classified as "other". Emission sources are divided into four main categories referred to the classification in MEIC: coal combustion by power plants (PP), industrial processes (IN), residential emission (RE) and transportation sector (TR).

Coal-fired power plants remain the main coal consumers in China, which



accounted for 50.2% of total coal consumption in 2019 (NBS, 2021) and gained much
more attention (Wu et al., 2022), especially with the wide implementation of the
strictest ultralow emission standards, $PM_{2.5}$ emission characteristics have changed
accordingly (Wu et al., 2020). There are obvious differences in $PM_{2.5}$ source profiles
between SPAPPC (SPAP database and published source profiles in China) and
SPECIATE (SPECIATE database), detailed information is shown in Table S3. The
percentages of species in PP source profiles are plotted in Fig. 2. The main components
in SPAPPC are sorted by Si, $SO_4^{2-}$, OC, Ca with average values of 8.7±6.8%, 8.5±11.5%,
6.8±9.1% and 6.5±6.9%, respectively; The SPECIATE are enriched in $SO_4^{2-}$
(16.9%±20.0%), OC (12.7±21.8%), Si (9.6±5.0%) and Ca (9.3±7.3%), higher than
SPAPPC. Coal properties, burning conditions, pollution control measures and sampling
methods are the main reasons for those great percentage fluctuations. Different
treatment processes of flue gases, e.g. wet/dry limestone, ammonia and double-alkali
flue gas desulfurization, will affect the percentages of components in source profiles
(Zhang et al., 2020). It has been reported that the percentage of Ca, Mg, $SO_4^{2-}$ and $Cl^-$
in PP profiles increased after the limestone-gypsum method was used in coal-fired
power plants (Bi et al., 2019). Besides that, the percentage of $Cl^-$ in SPAPPC is
obviously higher than that in SPECIATE, which might attribute to the generally higher
$Cl^-$ content in raw coal in China (Guo et al., 2004).



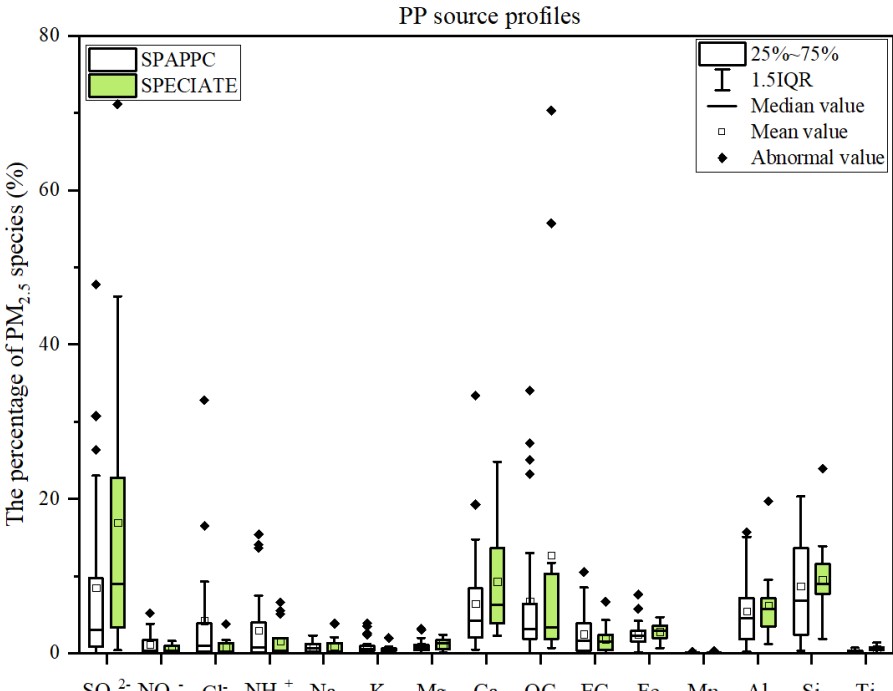

Fig. 2 Chemical profiles for PM$_{2.5}$ emitted from coal-fired power plant (PP). Data obtained from SPAPPC (SPAP database and published source profiles in China) and SPECIATE (U.S. EPA SPECIATE database)

Industrial emissions are one of the major sources of PM$_{2.5}$ (Hopke et al., 2020), the percentages of Ca, Fe, OC and SO$_4^{2-}$ are relatively high both in SPAPPC and SPECIATE of industrial processes, but the shares in different source profile database varied (Detailed information were shown in Table S4~S7). In SPAPPC, these four components account for 16.4±14.9%, 10.4±14.4%, 6.9±6.1%, 6.2±6.4%, the proportions in SPECIATE are 10.4±9.8%, 11.4±10.6%, 8.5±4.9%, 16.3±13.3%, respectively (Fig. 3). Large variations of components and their percentages in industrial processes are attributed to the manufacturing processes, raw material, pollution control measures and so on (Ji et al., 2017; Bi et al., 2019; Gao et al., 2022). For example, Ca, Al, OC and SO$_4^{2-}$ are found to have the highest percentage in cement sources (Guo et al., 2021); Fe, Si and SO$_4^{2-}$ are the most abundant species in steel industry emission (Guo et al., 2017).

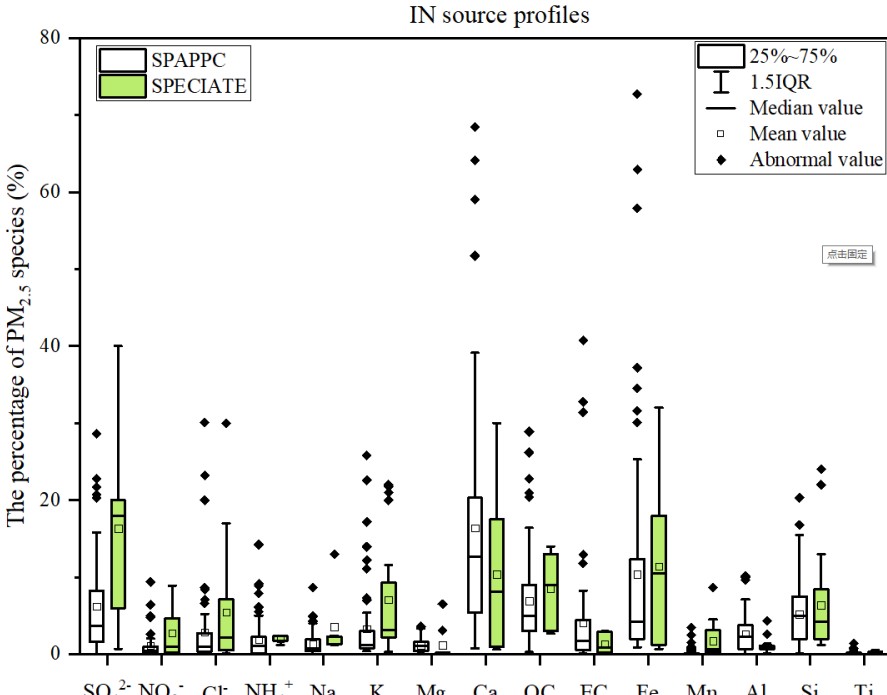

Fig. 3 Chemical profiles for PM$_{2.5}$ emitted from industry processes (IN). Data obtained from
SPAPPC (SPAP database and published source profiles in China) and SPECIATE (U.S. EPA
SPECIATE database)

Traffic contributed a large fraction of PM$_{2.5}$ in many locations (Hopke et al., 2022).

It is well-known that the transportation sector makes a dominant contribution of OC
and EC. The main components of PM$_{2.5}$ emitted from traffic sources are OC, EC and
SO$_4^{2-}$ both in SPAPPC and SPECIATE, but still vary in wide range (Detailed
information was given in Table S8~S10). In SPAPPC, the percentages of OC, EC and
SO$_4^{2-}$ are 40.8±15.0%, 23.1±13.8%, 3.1±3.7%, and in SPECIATE, the percentages are
40.6±16.4%, 36.1±21.5%, 6.4±9.9%, respectively (Fig. 4). These significant
differences mainly attribute to the vehicle type, fuel quality, mixing ratio between oil
and gas and the combustion phase in vehicle engine and so on (Xia et al., 2017).





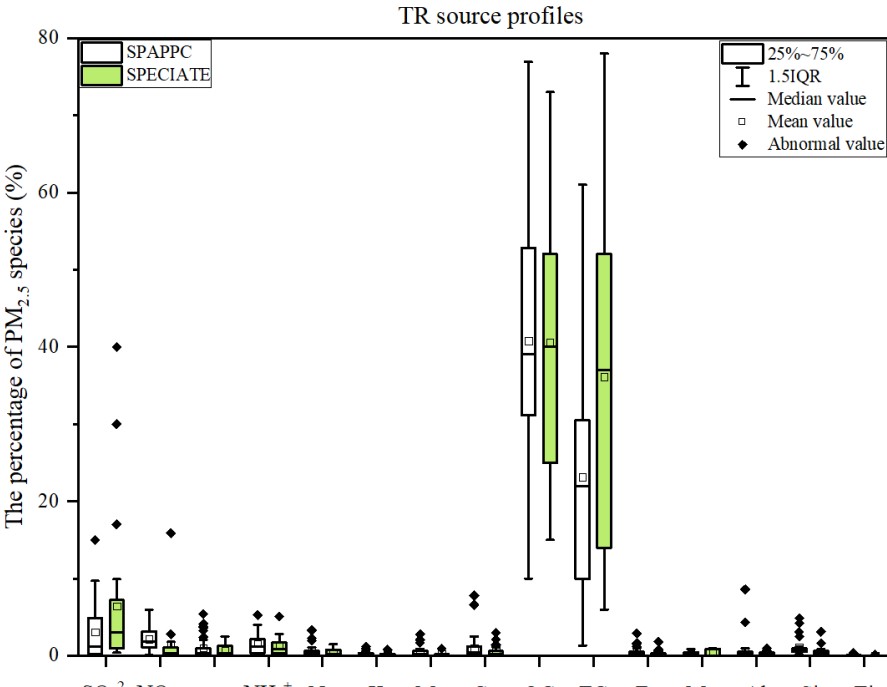

Fig. 4 Chemical profiles for PM$_{2.5}$ emitted from transportation sector (TR). Data obtained from
SPAPPC (SPAP database and published source profiles in China) and SPECIATE (U.S. EPA
SPECIATE database)

Residential coal combustion, as the leading source of global PM$_{2.5}$ emission

(Weagle et al., 2018), has a much higher emission factor than coal-fired power plant
(Wu et al., 2022). The fraction of components varied greatly in the profiles measured
from SPAPPC and SPECIATE (Detailed information was given in Table S11), SO$_4^{2-}$,
OC, NH$_4^+$ and EC make the main contribution to PM$_{2.5}$ emitted from residential coal
combustion. In SPAPPC, the average percentages of SO$_4^{2-}$, OC, NH$_4^+$, EC are
27.1±10.1%, 20.7±20.6%, 11.3±7.7%, 2.6±2.8%, respectively. In SPECIATE, the
average percentages are OC (58.2±14.0%), EC (24.6±5.4%), SO$_4^{2-}$ (3.2±2.3%) and
NH$_4^+$ (1.6±1.0%) (Fig. 5). Total percentages of OC and EC in SPECIATE are over 80%,
obviously higher than that in SPAPPC, while a higher percentage of SO$_4^{2-}$, Cl$^-$, K and
Si are observed in SPAPPC. The coal type and properties, burning condition are the
main factors affecting the percentages of PM$_{2.5}$ components, like the chunk coal burning
has relatively higher percentages of OC, EC, SO$_4^{2-}$, NO$_3^-$ and NH$_4^+$ than honeycomb





briquette (Wu et al., 2021; Song et al., 2021).

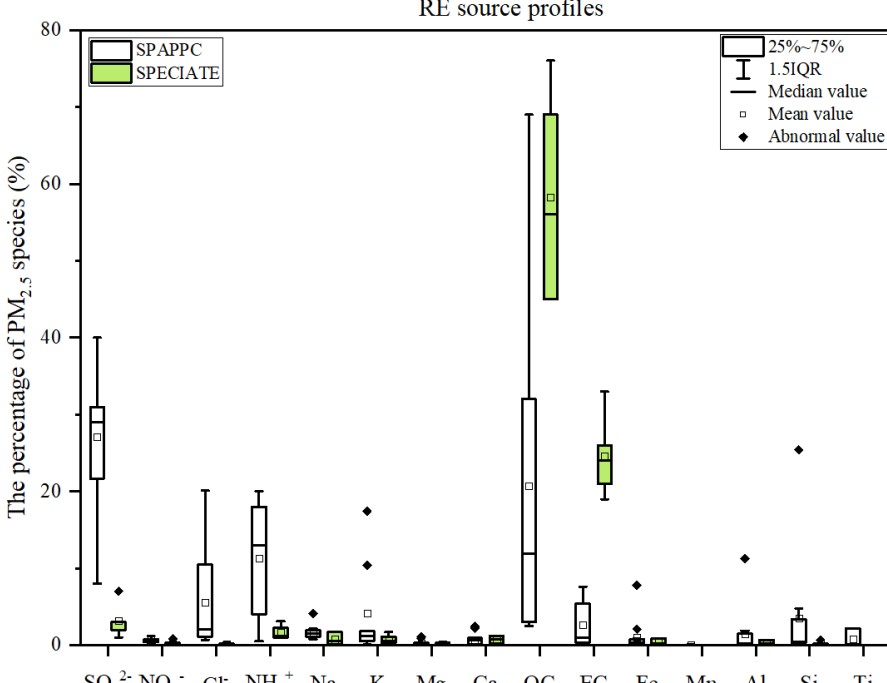

Fig. 5 Chemical profiles for PM$_{2.5}$ emitted from residential coal combustion (RE). Data obtained
from SPAPPC (SPAP database and published source profiles in China) and SPECIATE (U.S. EPA
SPECIATE database)
Briefly, many factors can affect PM$_{2.5}$ source profiles, and with the innovation of
manufacturing technique and pollution control technology, changes in fuel and raw and
auxiliary materials, the main chemical components and their percentages would change
dramatically. To explore whether the variations of source profile would be one of the
important factors affecting the simulation results of PM$_{2.5}$ species in CTMs, we
designed a series of simulation tests as follows.
**3 Whether the variation of source profile adopted in CTMs has an impact on the**
**simulation of chemical components in PM$_{2.5}$?**
In this part, we separately selected source profiles from SPAPPC and SPECIATE
databases and applied them in emission inventory for simulating PM$_{2.5}$ and its
components with other modeling conditions unchanged, corresponding to case



CMAQ_SPA and CMAQ_SPE. The detailed information on source profiles is shown
in Figure S1. To determine the similarity between the two groups of source profiles,
Coefficient divergence (CD) is calculated using the following formula
(Wongphatarakul et al., 1998):
$$CD_{jk} = \sqrt{\frac{1}{p}\sum_{i=1}^{p}\left(\frac{x_{ij}-x_{ik}}{x_{ij}+x_{ik}}\right)^2} \dots\dots\dots\dots\dots\dots\dots\dots \quad (1)$$
Where $CD_{jk}$ is the coefficient of divergence of source profile $j$ and $k$, $p$ was the
number of chemical components in source profile, $x_{ij}$ is the weight percentage for
chemical component i in source profile $j$, $x_{ik}$ is the weight percentage for $i$ in source
profile $k$ (%). The CD value is in the range of 0 to 1, if the two source profiles are
similar, the value of CD is close to 0; if the two are very different, the value was close
to 1.
By comparing the selected SPAPPC source profiles with the selected SPECIATE
source profiles, the coefficient divergences for the four main source categories were
$CD_{PP}(0.67) > CD_{RE}(0.62) > CD_{TR}(0.60) > CD_{IN}(0.60)$, which meant the selected source
profiles in the two simulation cases were quite different. The simulated concentration
of PM$_{2.5}$ and its components (For this part and each test case in next section) at 10
ambient air quality monitoring stations (Table S12) were extracted from CMAQ outputs
of the innermost simulation domain. We selected one air quality monitoring station to
study the influence of PM$_{2.5}$ source profile on numerical simulation of PM$_{2.5}$-bound
components and to explore the relevant laws in the atmosphere, then used the left 9 sites
to further illustrate the conclusions suggested.
The simulation results for PM$_{2.5}$ species under CMAQ_SPA and CMAQ_SPE
cases also showed big differences (as shown in Fig. 6 and Table S13), in which the
largest difference in simulated concentration was EC with CAMQ_SPE giving higher
by 167% than CMAQ_SPA; For OC and Mn, higher values were also given by
CMAQ_SPE than by CMAQ_SPA (45% and 126% on average, respectively); For the
remaining components, the simulated concentration by CMAQ_SPE was lower than
CMAQ_SPA with Ti (58%), Na (55%), Mg (53%), Ca (51%), Al (33%), Cl (30%), K

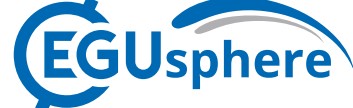

(29%), Si (22%), Fe (16%), $NH_4^+$ (9%), $SO_4^{2-}$ (9%), $NO_3^-$ (8%), separately. While the
simulated $PM_{2.5}$ concentrations under the two cases were quite close. The influence of
source profile variation on the simulated $PM_{2.5}$ concentration was not significant, but
the influence on the simulation of chemical components in $PM_{2.5}$ could not be ignored.

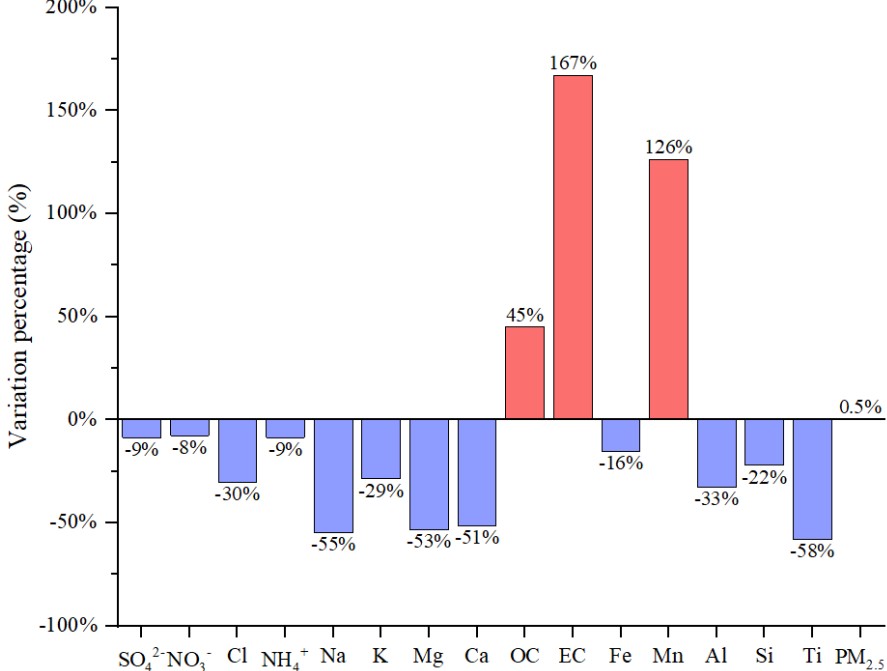


Fig. 6 The percentage difference of simulated concentration ($PM_{2.5}$ and its components) between
CMAQ_SPE and CAMQ_SPA (relative to CAMQ_SPA); $PM_{2.5}$ source profiles from SPAPPC and
SPECIATE database were applied in emission inventory for simulating $PM_{2.5}$ and its components,
corresponding to case CMAQ_SPA and CMAQ_SPE, respectively.
**4 How much did the variation of source profile adopted in CTMs impact on the**
**simulation of chemical components in PM₂.₅?**

In order to quantitatively characterize how much the source profiles affect the

simulation results of $PM_{2.5}$ and its components, we selected the chemical composition
of code 000002.5 (Variety of different categories, used for the overall average
composite profiles (Hsu et al., 2019)) in the US EPA Speciate_5.0_0 database as species



allocation of $PM_{2.5}$ components. The corresponding percentages of EC, OC, Mn, Fe, Ti,
Al, Si, Ca, Mg, K, Na, Cl, $NH_4^+$, $NO_3^-$ and $SO_4^{2-}$ in $PM_{2.5}$ were shown in Fig. 7 (SGL,
base case simulation).

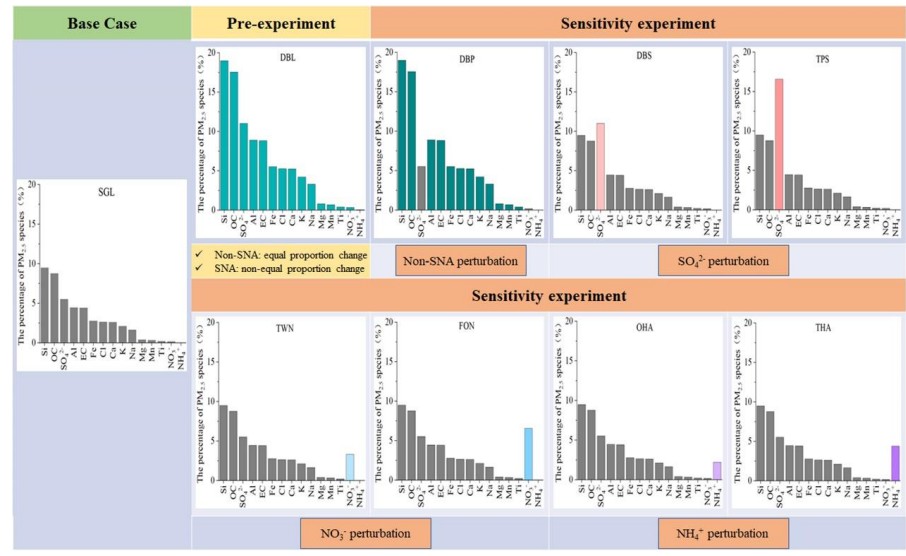


Fig. 7 The general roadmap of sensitivity tests

(The histogram in each case were the speciation profile in CTMs)

Table 1 The content of sensitivity experiment cases

| Cases | Description |
|---|---|
| Pre-experiment (DBL) | The percentage of all the listed components in the source profile of base case (SGL) were doubled, and the proportion of unlisted components (Other) decreased to 9%. |
| Case S1 (DBP): add perturbation to Non-SNA | The percentages of non-SNA were doubled and SNA( $SO_4^{2-}$, $NO_3^-$, $NH_4^+$) species stayed the same with that in SGL (the cumulative percentage of listed species was 85.3%), the proportion of unlisted components decreased to 14.7%. |
| Case S2 (DBS and TPS): add perturbation to $SO_4^{2-}$ | The percentage of $SO_4^{2-}$ was doubled (11%, DBS, represented Double Sulfate), tripled (16.5%, TPS, represented Triple Sulfate) and the other listed 14 species stayed the same with that in SGL (the cumulative percentage of listed species was 51% and 57%, respectively), the proportion of unlisted components decreased to 49% and 43%. |
| Case S3 (TWN and FON): add perturbation to $NO_3^-$ | The $NO_3^-$ content was raised up to 20 times (3.3%, TWN) and 40 times (6.6%, FON) of that in SGL (0.16%), the other 14 species stayed the same with SGL (the cumulative |



| | percentage of listed species was 48.6% and 51.9%, respectively), the proportion of unlisted components decreased to 51.4% and 48.1%. |
|---|---|
| Case S4 (OHA and THA): add perturbation to $NH_4^+$ | The $NH_4^+$ content was raised up to 100 times (2.2%, OHA), 200 times (4.4%, THA) of that in SGL (0.02%), the other 14 species stayed the same with SGL (the cumulative percentage of listed species was 47.7% and 49.9%, respectively), the proportion of unlisted components decreased to 52.3% and 50.1%. |

Note: The source profiles in all cases listed in the table were calculated based on the base case SGL. In the design of simulation cases, the reason why the disturbance amplitude of $NH_4^+$ and $NO_3^-$ were significantly higher than that of other components such as $SO_4^{2-}$ and Non-SNA, was because the percentages of $NH_4^+$ and $NO_3^-$ in the base source profile (SGL, based on the chemical composition of code 000002.5 in the EPA Speciate_5.0_0 database ) were very low, while the percentage of $NH_4^+$ and $NO_3^-$ in SPAPPC exhibited in section 2.2 were orders of magnitude higher than those in SGL.

Given the large number and complex chemical composition of $PM_{2.5}$, it is
advisable to classify it reasonably before designing sensitivity experiments. The pre-
experiment was to double the percentage of the listed 15 components mentioned above
(SGL) in $PM_{2.5}$ species allocation for emission sources (DBL case, the cumulative
percentage was 91%, the details were shown in Fig. 7 and Table 1). As the percentage
of these components increased, the proportion of unlisted components (represented by
Other) decreased to 9% in order to meet the requirement that the total percentage of all
components is 100%. Then we compared the simulation results before (SGL case) and
after perturbation (DBL case) in species allocation of $PM_{2.5}$ sources.
In the case DBL, when the percentage of all the components except "other" were
doubled in the source profile, the simulated concentrations of Al, Ca, Cl, EC, Fe, K,
Mg, Mn, Na, OC, Si and Ti doubled as well, while the simulated concentration of $NO_3^-$,
$SO_4^{2-}$ and $NH_4^+$ only increased at about 3%, 10% and 4%, respectively, although the
simulated concentration of $PM_{2.5}$ was not obviously changed (Detailed simulation
results were shown in Table S14). Through this pre-experiment, we found that the
results for SNA ($SO_4^{2-}$, $NO_3^-$, and $NH_4^+$) and Non-SNA were obviously different.
Therefore, we divided the components in the source profile into two groups (Non-SNA
and SNA) and designed a series of sensitivity tests listed in next section to further



explore how species allocation of PM$_{2.5}$ in emission sources of CTMs would affect the
simulation results.

**4.1 Sensitivity tests design**

Based on the pre-experiment results, sensitivity tests were designed by changing

the percentages of the target components and related components in the base case (SGL):
perturbation on each component of Non-SNA, perturbation on SO$_4^{2-}$, perturbation on
NO$_3^-$, and perturbation on NH$_4^+$. The general roadmap of sensitivity tests was shown in
Fig. 7, and the illustration of each case was summarized in Table 1. The basic rules must
be followed: a) perturbation on the percentage of each component in source profile fell
within the variation range of its measured value described in section 2.2. b) The sum of
the percentage of listed Non-SNA, SNA and Other components in PM$_{2.5}$ source profile
was 100%.

**4.2 Evaluation index for simulation result**

In order to quantify the concentration changes of simulated PM$_{2.5}$ components

caused by the perturbation in source profile, we proposed the sensitivity coefficient (δ)
as evaluation index. The calculation formula is as follows:
$$\delta_i = \begin{cases} \dfrac{\dfrac{C_{i\_case}}{C_{PM_{2.5}\_case}}\times100\% - \dfrac{C_{i\_base}}{C_{PM_{2.5}\_base}}\times100\%}{P_{i\_case} - P_{i\_base}} & (For\ DBL\ and\ DBP) \\[4ex] \dfrac{\dfrac{C_{i\_case}}{C_{PM_{2.5}\_case}}\times100\% - \dfrac{C_{i\_base}}{C_{PM_{2.5}\_base}}\times100\%}{P_{j\_case} - P_{j\_base}} & (For\ other\ cases) \end{cases} \qquad \text{………(2)}$$

Wherein, δ$_i$ is the sensitivity coefficient of component $i$, representing the change

of the simulated value of its content in ambient PM$_{2.5}$ corresponded to 1% perturbation
in the source profiles. $C_{i\_case}$ is the simulation result of component $i$ in different
sensitivity experiment cases, μg/m$^3$; $C_{i\_base}$ is the simulation result of components $i$ in
base case, μg/m$^3$; $C_{PM_{2.5}\_case}$ is the simulation result of PM$_{2.5}$ in different sensitivity
experiment cases, μg/m$^3$; $C_{PM_{2.5}\_base}$ is the simulation result of PM$_{2.5}$ in base case, μg/m$^3$;
$P_{i\_case}$ is the percentage of component $i$ in different source profile of sensitivity



experiment cases, %; $P_{j\_case}$ is the percentage of perturbed component $j$ in different
source profile of sensitivity experiment cases, %; $P_{i\_base}$ is the percentage of component
$i$ in base case source profile, %; ; $P_{j\_base}$ is the percentage of perturbed component $j$ in
base case source profile, %.

The positive value of δ means the simulated concentration of $PM_{2.5}$ component

increases (decreases) with the increase (decrease) of the perturbation to the percentage
of components in source profile, while the meaning of negative δ is just the opposite. If
the absolute value of δ is less than or equal to 0.1, the simulated result of $PM_{2.5}$ chemical
component is considered to be insensitive to the corresponding variation of source
profile; If the absolute value of δ falls between 0.1 and 0.4 (included), the simulated
results of $PM_{2.5}$ chemical component is considered to be sensitive to the variation of
source profile; If the absolute value of δ is larger than 0.4, the simulated results of $PM_{2.5}$
chemical component is very sensitive to the variation of source profile. The greater the
absolute value of δ is, indicates the variation of source profile adopted in CMAQ has
more obvious impact on the simulated results of $PM_{2.5}$ chemical components.
**4.3 The response of simulated $PM_{2.5}$ components**

Fig.8 listed the sensitivity coefficients of simulated ambient $PM_{2.5}$ components to

the perturbation of source profile under each test case. In case DBL (The percentage of
all the listed components in the source profile of base case (SGL) was doubled), the
sensitivity coefficient (δ) of $NH_4^+$ was negative, and the absolute value was the highest,
indicating that the simulated proportion of $NH_4^+$ in ambient $PM_{2.5}$ decreased, and it was
very sensitive to the variation of source profile. Conversely, the sensitivity coefficient
of $NO_3^-$ was close to 1, which illustrated that the simulated proportion of $NO_3^-$ in
ambient $PM_{2.5}$ increased proportionally with the change in source profile. The δ of $SO_4^{2-}$
also showed a very sensitive property. The simulated Non-SNA concentrations were
doubled when compared to the base case (SGL).



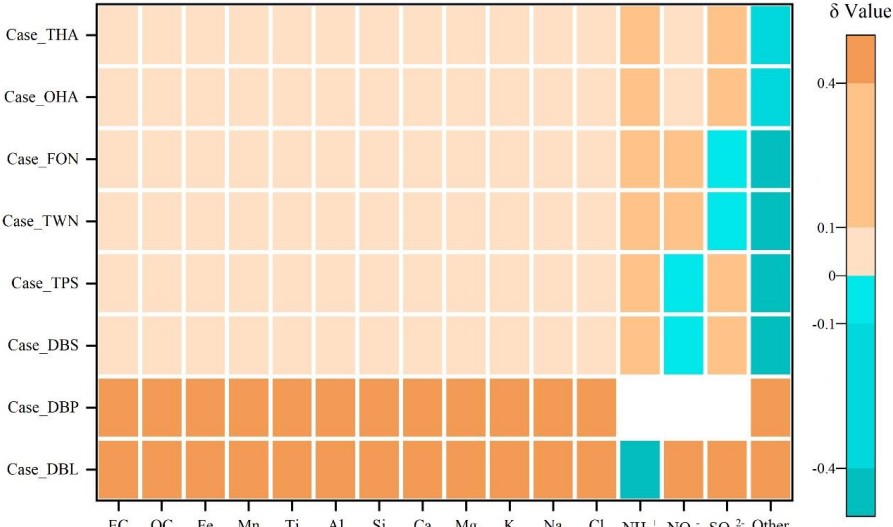

Fig. 8 The sensitivity coefficients (δ) of simulated components to the perturbation of adopted source profile in different cases. Note: Each small color box in the figure represented the sensitivity level (indicated by the legend on the right) of $PM_{2.5}$ components (the x-coordinate) in different cases (y-coordinate). The blank grids in DBP case indicated no perturbation to SNA in $PM_{2.5}$ source profile under this case.

In case DBP, when the percentages of listed Non-SNA in the source profile were doubled, the simulated proportions of Non-SNA (Al, Ca, Cl, EC, Fe, K, Mg, Mn, Na, OC, Si and Ti) in ambient $PM_{2.5}$ synchronous increased, and were very sensitive to the change in the adopted source profile with a sensitivity coefficient (δ) of 0.5. Interestingly, the simulated concentration of SNA in ambient $PM_{2.5}$ also changed although the SNA in source profile did not change, the concentration of $NO_3^-$ and $SO_4^{2-}$ increased by 2% and 3%, respectively, $NH_4^+$ decreased by 10% (Detail simulation results of different cases were shown on Table S15~S21).

Under $SO_4^{2-}$ perturbation cases (Case DBS and Case TPS), we found the simulated results of Non-SNA and $NO_3^-$ had no obvious variation when compared with the base case. Either in Case DBS or in Case TPS, the δ of Non-SNA and $NO_3^-$ were always between -0.1 to 0.1. But when the percentage of $SO_4^{2-}$ was doubled in $PM_{2.5}$ source profile (DBS), the simulated concentration of $NH_4^+$ and $SO_4^{2-}$ increased by 6% and 8%, respectively. In Case TPS (the percentage of $SO_4^{2-}$ was tripled), the simulated





395 concentration of $NH_4^+$ and $SO_4^{2-}$ were increased by 11% and 16%, respectively. The δ

396 of $NH_4^+$ and $SO_4^{2-}$ were 0.12 and 0.36, sensitive toward to positive direction with the

397 increase of $SO_4^{2-}$ in the source profile.

398   In the situation of $NO_3^-$ perturbation (Case TWN and Case FON), the simulated

399 concentrations of Non-SNA hardly change when compared to the base case, while the

400 changing characteristics of SNA concentrations were different. In cases TWN and FON,

401 the simulation concentration of $NH_4^+$ increased by 2.6% and 5.4% when compared with

402 the base case, the simulated $NO_3^-$ increased by 14% and 30%, the simulated $SO_4^{2-}$

403 decreased slightly, even could be neglected in some observation sites. The simulated

404 concentrations of Non-SNA and $SO_4^{2-}$ were insensitive to the perturbation of $NO_3^-$ in

405 $PM_{2.5}$ source profile; $NH_4^+$ was sensitive, and $NO_3^-$ was very sensitive.

406   When we put perturbation to $NH_4^+$ in the source profile (Case OHA and Case

407 THA), the simulation results of Non-SNA were almost not changed, the simulated

408 concentration of $SO_4^{2-}$, $NH_4^+$, $NO_3^-$ increased in OHA and THA. The δ of SNA to the

409 variation of $NH_4^+$ in the source profile were positive and $\delta(SO_4^{2-}) > \delta(NH_4^+) > \delta(NO_3^-)$,

410 $SO_4^{2-}$ and $NH_4^+$ were sensitive to the $NH_4^+$ perturbation in the source profile, but $NO_3^-$

411 was not so sensitive.

412   In general, the simulation results of components in ambient $PM_{2.5}$ were affected in

413 one way or another by the change of source profiles adopted by CMAQ. Both of the

414 simulated Non-SNA and SNA were very sensitive to the perturbation of Non-SNA in

415 source profile. When the percentage of SNA changed in the source profile, simulated

416 concentrations of Non-SNA generally have little change, but the simulation results of

417 SNA could change in different levels: the simulated $SO_4^{2-}$ was very sensitive and $NH_4^+$

418 was sensitive to the perturbation of $SO_4^{2-}$ in source profile, simulated $NO_3^-$ was very

419 sensitive and $NH_4^+$ was sensitive to the perturbation of $NO_3^-$, $SO_4^{2-}$ and $NH_4^+$ were

420 sensitive to the perturbation of $NH_4^+$. The simulated component such as $SO_4^{2-}$ was

421 influenced not only by the change of $SO_4^{2-}$ itself but also by other components like

422 some Non-SNA and $NH_4^+$ in the source profile. In other words, there was a linkage

423 effect, variation of some components in the source profile would bring changes to the



simulated results of other components.

**5 How the variation of source profile adopted in CTMs impact on the simulation**
**of chemical components in PM$_{2.5}$?**

The variation of species allocation in emission sources directly affected the
composition of aerosol system in CTMs. In CMAQv5.0.2, the aerosol thermodynamic
equilibrium process was carried out according to ISORROPIA II, including a SO$_4^{2-}$-
NO$_3^-$-Cl$^-$-NH$_4^+$-Na$^+$-K$^+$-Mg$^{2+}$-Ca$^{2+}$-H$_2$O system which was established on the basis of
ISORROPIA I by adding the effects of K$^+$, Ca$^{2+}$ and Mg$^{2+}$ (Detailed equilibrium
relations were shown in Table S22). Some assumptions had been made in the
ISORROPIA model to simplify the simulation system (Fountoukis and Nenes, 2007):
(1) Because the vapor pressure of sulfuric acid and metal salts (such as Na$^+$, Ca$^{2+}$, K$^+$,
Mg$^{2+}$) were very low, it was assumed that all the sulfuric acid and metal salts in the
system existed in the aerosol phase; (2) For ammonia in the system, it was preferred to
have an irreversible reaction with sulfuric acid to produce ammonium sulfate. Only
when there was still surplus NH$_3$ after the neutralization of H$_2$SO$_4$, can it have a
reversible reaction with HNO$_3$ and HCl to produce NH$_4$NO$_3$ and NH$_4$Cl. (3) For sulfuric
acid in the system, if there were metal ions (such as Ca$^{2+}$, Mg$^{2+}$, K$^+$, Na$^+$) in the system,
sulfuric acid would react with metal ions to produce metal salts. Only in the case of
insufficient sodium, sulfuric acid would react with ammonia. Based on these
assumptions, the ISORROPIA model introduced the following three judgment
parameters (R$_1$, R$_2$ and R$_3$ were calculated by the following formulas) to determine the
simulation subsystems. In this paper, R$_1$, R$_2$, R$_3$ and the corresponding solid phase
species under different perturbation cases on source profiles were shown in Table 3.
These components achieved thermodynamic equilibrium in the order of preference for
more stable salts, obviously, the simulation processes of these components may
influence each other.
$$R_1 = \frac{\left[NH_4^+\right] + \left[Ca^{2+}\right] + \left[K^+\right] + \left[Mg^{2+}\right] + \left[Na^+\right]}{\left[SO_4^{2-}\right]} \quad \text{.....................} \quad (3)$$





$$R_2 = \frac{\left[Ca^{2+}\right]+\left[K^+\right]+\left[Mg^{2+}\right]+\left[Na^+\right]}{\left[SO_4^{2-}\right]} \dots\dots\dots\dots\dots\dots\dots\dots\dots \ (4)$$

$$R_3 = \frac{\left[Ca^{2+}\right]+\left[K^+\right]+\left[Mg^{2+}\right]}{\left[SO_4^{2-}\right]} \dots\dots\dots\dots\dots\dots\dots\dots\dots\dots\dots \ (5)$$

Table 2 Potential aerosol species in ISORROPIA II under different cases

| Cases | $R_1$ | $R_2$ | $R_3$ | Solid phase species* |
|---|---|---|---|---|
| SGL、DBL TWN、FON | 2.53 | 2.52 | 1.9 | $CaSO_4$, $MgSO_4$, $K_2SO_4$, $Na_2SO_4$, $NaCl$, $NaNO_3$, $NH_4Cl$, $NH_4NO_3$ |
| DBS | 1.26 | 1.26 | 0.95 | $CaSO_4$, $MgSO_4$, $K_2SO_4$, $KHSO_4$, $Na_2SO_4$, $NaHSO_4$, $(NH_4)_2SO_4$, $NH_4HSO_4$, $(NH_4)_3H(SO_4)_2$ |
| TPS | 0.84 | 0.84 | 0.63 | $CaSO_4$, $KHSO_4$, $NaHSO_4$, $NH_4HSO_4$ |
| DBP | 5.04 | 5.03 | 3.79 | $CaSO_4$, $MgSO_4$, $K_2SO_4$, $CaCl_2$, $Ca(NO_3)_2$, $MgCl_2$, |
| OHA | 3.58 | 2.52 | 2.95 | $Mg(NO_3)_2$, $KCl$, $KNO_3$, $NaCl$, $NaNO_3$, $NH_4Cl$, |
| THA | 4.64 | 2.52 | 4.02 | $NH_4NO_3$ |

* The solid phase species were determined based on the research of (Fountoukis and Nenes, 2007)

In Non-SNA perturbation case, when the percentage of Non-SNA in source profile doubled (Case DBP), meant there were more Na, K, Mg, Ca, Cl participated in aerosol chemistry, the model system needed more $SO_4^{2-}$ and $NO_3^-$ on the basis of charge balance and the thermodynamic equilibrium shifted to the direction of consuming Ca Mg, K and Na, which resulted in the increase of the simulated concentration of $SO_4^{2-}$ and $NO_3^-$. Meanwhile, according to the rule of anions preferentially binding with nonvolatile cations in ISORROPIA, the increased cations $Na^+$, $K^+$, $Mg^{2+}$, $Ca^{2+}$ directly leaded to the decrease of anions binding with $NH_4^+$, there were less reaction dose between $SO_4^{2-}$ and $NH_4^+$ to form $(NH_4)_2SO_4$ or $NH_4HSO_4$, ultimately resulted in a decrease in simulated concentration of $NH_4^+$ when compared to the base case. Because in this case more anions such as $SO_4^{2-}$ were passively needed, according to the principle of chemical equilibrium mentioned above, the chemical conversion of $SO_2$ to $SO_4^{2-}$ was promoted, the simulated secondary $SO_4^{2-}$ increased, this could be proved by that the δ of $SO_2$ in Case DBP was negative (shown in Fig. 9, details of other monitoring stations were shown Table S24).

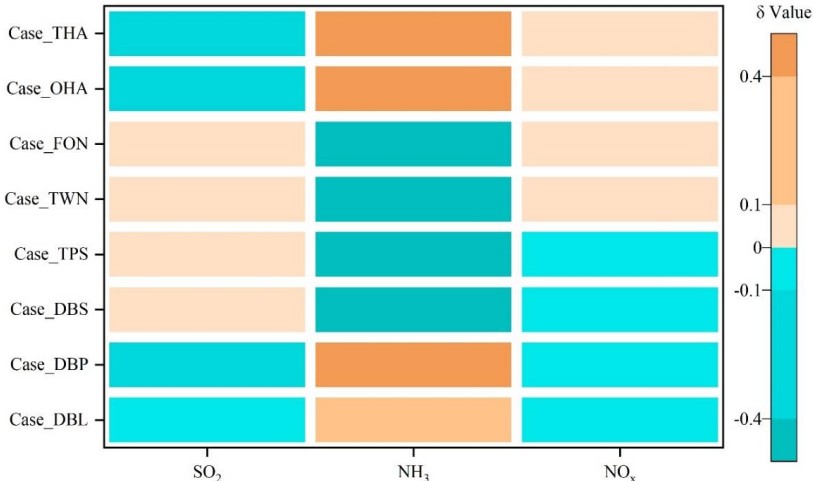

Fig.9 The sensitivity coefficients (δ) of simulated gas pollutants to the change of adopted source
profile in different cases.
Similarly, with the increase of metal ions in the system to bond with anions, the
number of anions which can bind to $NH_4^+$ decreased. The system needed less $NH_4^+$ and
weakened the need for conversion from $NH_3$ to $NH_4^+$, the simulated $NH_4^+$ concentration
decreased while the δ of $NH_3$ was positive and very sensitive. Different trends of
simulated concentration of gaseous pollutants mirrored the rules mentioned above from
another aspect. The δ of $SO_2$ and $NO_x$ was negative, $NH_3$ was positive. We could see
the same phenomena in DBL case (Fig. 9). When the percentages of Non-SNA in source
profile increased, they not only affected the simulated concentration of Non-SNA, but
also the secondary $SO_4^{2-}$, $NO_3^-$ and $NH_4^+$.
In $SO_4^{2-}$ perturbation cases (Case DBS and TPS), as the percentage of $SO_4^{2-}$ in
source profile increased, for the chemical reactions of sulfate radical consuming (as
shown in Table S22), the chemical equilibrium would move toward the products when
compared to the base case. While for the chemical reactions of sulfate radical formation
(The equations were shown in Table S23), meant the product was added in, the chemical
equilibrium would be pushed toward the reactants. The chemical reactions between
$SO_4^{2-}$ and $NH_4^+$ would shift to the direction of $(NH_4)_2SO_4$ generation, we could see the
simulated concentrations of $NH_4^+$ in DBS and TPS were both higher and $NH_3$ were
lower than those in the base case (SGL). In addition, when more $SO_4^{2-}$ was added in,





the conversion of $SO_2$ to $SO_4^{2-}$ was affected in some level and consumed less $SO_2$ than
the base case, simulated $SO_2$ showed insensitive but positive trend (Fig.9). And from
the potential solid phase species in ISORROPIA II under DBS and TPS cases (Table 3),
the solid phase species were mainly consisted of sulfate salts, so the simulated
concentration of $NO_3^-$ did not change apparently.

As the percentage of $NO_3^-$ in source profile increased (Case FON and TWN), the

associated chemical equilibrium shifted towards the consumption of $NO_3^-$, such as $NH_4^+$
$+ NO_3^- \rightarrow NH_4NO_3$, which would also consume more $NH_4^+$ and form more ammonium
salt, finally consumed more $NH_3$ because of $NH_3(gas) + H_2O(aq) \rightarrow NH_4^+(aq) + OH^-$
(aq). The simulation results also manifested that the concentration of $NH_4^+$ increased
while that of $NH_3$ decreased. Based on the assumption of ISORROPIA, the cations like
$Na^+$, $K^+$, $Mg^{2+}$, $Ca^{2+}$ and $NH_4^+$ preferentially to react with $SO_4^{2-}$, only if there were
cations left after neutralized $SO_4^{2-}$, could they react with $NO_3^-$ to form salts, so the
simulated concentration of $SO_4^{2-}$ was not obviously changed. Accordingly, the
simulated concentration of $NO_x$ and $SO_2$ almost unchanged (The $\delta$ of $NO_x$ and $SO_2$ was
insensitive).

In the cases of $NH_4^+$ perturbation (Case OHA and THA), when the percentage of

$NH_4^+$ in source profile increased, the related chemical equilibrium shifted towards the
direction of $NH_4^+$ consumption, such as in $2NH_4^+ + SO_4^{2-} \rightarrow (NH_4)_2SO_4$, more $SO_4^{2-}$
was consumed at the same time, which further promoted the conversion of $SO_2$ to $SO_4^{2-}$.
The increased $NH_4^+$ in OHA and THA also would inhibit the conversion of $NH_3$ to $NH_4^+$
when compared to the base case. This, in turn appeared as the increase of the simulated
secondary $SO_4^{2-}$ and $NH_3$, and the decrease of the simulated $SO_2$.

In summary, the effects of source profile variation on the simulation results of

different components were linked. When the percentages of Non-SNA, $SO_4^{2-}$, $NO_3^-$ and
$NH_4^+$ in the source profile changed, they not only affected the simulated concentration
of themselves, but also affected the simulation results of some other components. Both
the simulation results of primary components and secondary components were affected
by the change of source profile, the secondary $SO_4^{2-}$ and $NH_4^+$ were affected more than



the secondary $NO_3^-$.

**6    Conclusions**

Although the influence of source profile variation on the simulated concentration
of ambient $PM_{2.5}$ is not significant, its influence on the simulated chemical components
cannot be ignored. The variation of simulated components ranges from 8% to 167%
under selected different source profiles, and the simulation results of some components
are sensitive to the adopted $PM_{2.5}$ source profile in CTMs, e.g., both the simulated Non-
SNA and SNA are sensitive to the perturbation of Non-SNA in source profile, the
simulated $SO_4^{2-}$ and $NH_4^+$ are sensitive to the perturbation of $SO_4^{2-}$, simulated $NO_3^-$ and
$NH_4^+$ are sensitive to the perturbation of $NO_3^-$, $SO_4^{2-}$ and $NH_4^+$ are sensitive to the
perturbation of $NH_4^+$. These influences are not only specific to an individual component,
but also can be transmitted and linked among components, that is, the influence path is
connected to chemical mechanisms in the model since the variation of species allocation
in emission sources directly affect the thermodynamic equilibrium system
(ISORROPIA II, $SO_4^{2-}$-$NO_3^-$-$Cl^-$-$NH_4^+$-$Na^+$-$K^+$-$Mg^{2+}$-$Ca^{2+}$-$H_2O$ system).
Traditionally, the source profiles are regarded as a primary emission, but
interestingly, their variation could affect the simulation result of secondary components
as well in CTMs. We found the perturbation of $PM_{2.5}$ source profile caused the variation
of simulated gaseous pollutants, and related chemical reactions like gas-phase
chemistry of $SO_2$, $NO_x$ and $NH_3$, which mirrored that the perturbation of source profile
had an effect on the simulation of secondary $PM_{2.5}$ components. Overall, the emission
source profile used in CTMs is one of the important factors affecting the simulation
results of $PM_{2.5}$ chemical components. Additionally, organic species are one of the most
important components in $PM_{2.5}$ and gain much more attention on human health. While
the number of organic species in source profile is relatively scarce which brings a
challenge for simulation test designing, the variation of source profile adopted in CTMs
has an impact on the simulation of organic species is not taken into account in this study.
With the change of fuel and raw materials, the development of production



technology and the innovation of pollution treatment technology in recent years, some
components have changed significantly in the source profile. Given the important role
of air quality simulation in environment management and health risk assessment, the
representativeness and timeliness of the source profile should be considered.
**Data availability**
The input datasets for WRF simulation are available at
https://rda.ucar.edu/datasets/ds351.0/index.html (The National Center for Atmospheric
Research (NCAR)). The Multi-resolution Emission Inventory for China (MEICv1.3) is
available at http://meicmodel.org/?page_id=135. The $PM_{2.5}$ emission source profiles
from database of Source Profiles of Air Pollution (SPAP)
(http://www.nkspap.com:9091/, Nankai university), SPECIATE database
(https://www.epa.gov/air-emissions-modeling/speciate, U.S. Environmental Protection
Agency's (EPA)), Mendeley data repository (https://doi.org/10.17632/x8dfshjt9j.2, Bi
et al., 2019).
**Code availability**
The source code for CMAQ version 5.0.2 is available at
https://github.com/USEPA/CMAQ/tree/5.0.2 (last access: April 2014)
(https://doi.org/10.5281/zenodo.1079898, US EPA Office of Research and
Development, 2018). The source code for WRF version 3.7.1 is available at
https://github.com/NCAR/WRFV3 (last access: 14 August 2015, NCAR).
**Author contributions**
Zhongwei Luo: Data curation and collection, writing–original draft. Yan Han:
Modeling, writing–original draft. Kun Hua: Data collection. Yufen Zhang:
Supervision–Review & editing. Jianhui Wu: Supervision in source profile. Xiaohui Bi:
Supervision in source profile. Qili Dai: Resources. Baoshuang Liu: Resources. Yang
Chen: Modification and editing. Xin Long: Supervision in modeling. Yinchang Feng:
Supervision–Review & editing.



**Competing interests**

The authors declare that they have no known competing financial interests or personal relationships that could have appeared to influence the work reported in this paper.

**Disclaimer. Publisher's note**

Copernicus Publications remains neutral with regard to jurisdictional claims in published maps and institutional affiliations.

**Acknowledgements**

We would like to thank the National Natural Science Foundation of China (grant number 42177465) for providing funding for the project. We are grateful for the Inventory Spatial Allocate Tool (ISAT) provided by Kun Wang from Department of Air Pollution Control, Institute of Urban Safety and Environmental Science, Beijing Academy of Science and Technology.

**Financial support**

This study was financially supported by the National Natural Science Foundation of China (grant number 42177465).

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
