# Peer review of "The effect of emission source chemical profiles on simulated PM2.5 components"

_EGUsphere, 2022_

## Author Comment (AC2)

**Response to Anonymous Referee #1's comments on manuscript egusphere-2022-895**

We thank the Reviewer for the immensely helpful comments. In response, we have revised the main text to improve clarity and grammar throughout. We respond to each specific comment in detail below. The reviewer comments are shown in *black italics*. Our replies are shown in indented black text, and the modified text is shown in corresponding screenshots. The annotated line numbers refer to the revised copy of the manuscript.

*The manuscript attempts to explore the influence of adopted emission source profiles in CTMs on the simulated results of $PM_{2.5}$ components by sensitivity analysis. The extent of the influence for different components were quantitatively analyzed, the impact laws and pathway were identified. The topic is interesting and their findings highlight the importance of effective utilization of emission source profiles in CTMs. Although the description of experiments is complete to allow their reproduction by fellow researchers, some explanations and discussions are not clear. I recommend its publication subject to the following amendments.*

*Major concern 1:*

*What is the design basis for the perturbation of emission source profile in the sensitivity experiments?*

**Response:**

First, we analysis the source profile through the published literatures and existing source profile databases, we found that **the main components and their contents of different sources were significantly different**, for example, 1) In industry process, the percentages of Ca, Fe, OC and $SO_4^{2-}$ are relatively high, but the shares in different source profile database varied. In SPAPPC (database of Source Profiles of Air Pollution and published source profiles in China), these four components account for 16.4±14.9%, 10.4±14.4%, 6.9±6.1%, 6.2±6.4%, the proportions in SPECIATE (US EPA SPECIATE database) are 10.4±9.8%, 11.4±10.6%, 8.5±4.9%, 16.3±13.3%, respectively. 2) The transportation sector makes a dominant contribution of OC and EC, but still vary in wide range: In SPAPPC, the percentages of OC, EC are 40.8±15.0%, 23.1±13.8%, and in SPECIATE, the percentages are 40.6±16.4%, 36.1±21.5%, respectively. Besides that, **the variations of main components in the same category of emission sources are also obvious**, for example, the compositions of $PM_{2.5}$ emitted by coal-fired power plants with different flue gas desulfurization facilities, e.g. wet/dry limestone, ammonia and double-alkali flue gas desulfurization, have been proved to be very different. **So we take the variation range in the source profile as the range of the sensitivity experiment for each component.** The detail of this step is shown in section 2.2 of the manuscript.

Second, we divided the components into several groups according to the pre-experiment due to the large number and complex chemical composition of $PM_{2.5}$. **Through the pre-experiment**, we found that the results for SNA ($SO_4^{2-}$, $NO_3^-$, and $NH_4^+$) and Non-SNA were obviously different. Therefore, **we divided the components in the source profile into four groups (Non-SNA, $SO_4^{2-}$, $NO_3^-$, and $NH_4^+$)**. The second step could be found in section 4 of the manuscript.

Based on the two pieces of information mentioned above, 1) the perturbation on the percentage of each component in source profile must fell within the variation range of its measured value described in section 2.2. 2) The sum of the percentage of listed Non-SNA, SNA and Other components in $PM_{2.5}$ source profile was 100%; Finally, the sensitivity experiment of **perturbation on Non-SNA, perturbation on $SO_4^{2-}$, perturbation on $NO_3^-$, and perturbation on $NH_4^+$ were determined**. In the meantime, keeping the other modeling conditions unchanged except source profile.

In general, the perturbation on each component was fallen in the actual fluctuation percentage range of that component in source profile, and

grouped based on pre-experiment results   to design the sensitivity experiment. The design idea is shown in Figure RC1 as follows:

[Figure]

Figure RC1 The sketch of design idea

*Major concern 2:*

*The discussion of the results should be extended. The authors mentioned that emission source profile adopted in CTMs has a significant impact on the simulation results of PM$_{2.5}$ components, so how to select the appropriate source profiles in the simulation? In the section of conclusion*

, *the author concluded that "the representativeness and timeliness of the source profile should be considered". How to understand the "representativeness" and "timeliness" here?*

**Response:**

Source profile, a physicochemical point of view of which reveals the signatures of source emission, play an important role in the application of CTMs for converting total emissions from source into the speciated emission and calculating source-specific emission of individual compounds (Reff et al., 2009; Hsu et al., 2019). In the past few years, source profile of $PM_{2.5}$ from a variety of source types have been substantially developed all over the world, especially in US (Simon et al., 2010), Europe (Pernigotti et al., 2016) and East Asia (Liu et al., 2017; Bi et al., 2019). With the change of fuel and raw materials, the development of production technology and the innovation of pollution treatment technology in recent years, some components have changed significantly in the source profile. By comparing the source profile in exited databases and published literatures, we found that the components in $PM_{2.5}$ source profiles have the following characteristics:

**Firstly, the large variation of components content exists in source profiles.** We take coal-fired power plants (PP) as an example here (Coalfired power plants remain the main coal consumers in China (NBS, 2021), source profile data were from SPECIATE[1] and SAPPC[2]). The dominant components generally are similar such as $SO_4^{2-}$, $Cl^-$, Ca, OC, Al and Si in PP source profiles, however, there are large and small differences in their contributions. In SAPPC, the average weight percentage of main components are sorted by $SO_4^{2-}$, $Cl^-$, Ca, OC, Si, and their percentage range were 0.6%~47.4%, 0.1%~27.8%, 0.6%~24.1%, 0.3%~34%, 0.4%~28.3%, respectively. In SPECIATE, the main components in PP source profiles were $SO_4^{2-}$, Ca, OC, Al, Si, and their variation range were 0.4%~71.1%, 2.3%~24.8%, 0.7%~70.3%, 1.2%~19.7%, 1.9%~23.9%, separately. Our previous study also showed that the relatively large variation in the source profiles for industry emissions, vehicle emissions and residents coal combustion, it is called for the establishment of local profiles for these sources (due to their high uncertainties) through the uncertainty analysis (Bi et al., 2019).

**Secondly, the main components (or the tracer components) of emission sources have changed because of the changing standards.** On Jan. 1, 2012, China began to implement the new Emission Standards for Air Pollutants from Thermal Power Plants (GB13223-2011, https://www.mee.gov.cn/ywgz/fgbz/bz/bzwb/dqhjbh/dqgdwrywrwpfbz/201109/W020130125407916122018.pdf), which stipulates that $SO_2$

emissions from thermal power boilers in key areas shall be subject to the stricter standard. To meet new emission standards, the installation rate of desulphurization facilities in coal-fired power plants has greatly increased, which to some extent affects the composition of coal-fired sources in Chinese cities. It has been reported that the percentages of Ca, Mg, $SO_4^{2-}$ and $Cl^-$ in PP profiles increased after the limestone-gypsum method was used in coal-fired power plants (Zhang et al., 2020; Bi et al., 2019), Ammonia desulphurization will increase $NH_4^+$ and $SO_4^{2-}$ in particulate matter (Pan et al., 2016). Due to the changing standards of gasoline and diesel oil since the 1980s, Pb and Mn are no longer tracers of gasoline vehicle emissions (Bi et al., 2019). However, OC and EC still are the dominant species in vehicle emissions since the 1980s, despite the changing standards, this also could be seen from our manuscript. Especially China plans to achieve carbon neutral before 2060, more stringent standards will be introduced, the characteristics of source profiles' components will also change.

**Thirdly, with the development of advanced sampling and chemical analysis techniques, more valuable information has been explored to further know about the source profiles.** A number of recent studies found that, contrary to our previous belief, primary emission may be more important for some components, for example, sulfate (a major $PM_{2.5}$

component) was largely from primary emissions rather than secondary formation in ambient air in certain circumstances (Dai et al., 2019; Ding et al., 2021; Yan et al., 2020).

**Besides that, fuel, raw and auxiliary materials, process conditions, pollution removal facilities, source sampling methods and other factors have a significant impact on the source profile of PM$_{2.5}$.**

Therefore, The representativeness and timelessness of source profile, **from a macro perspective**, it needs to see whether it is a typical source profile, and whether it can represent the chemical composition of PM$_{2.5}$ emitted by sources in the region in the study stage; **From the micro-view**, it is to evaluate whether the components' characteristics in the source profiles can represent the chemical compositions of the vast majority of such sources in the actual environment, which is based on the general chemical composition law of the source profile. We should **consider the regional emission character** and **the characteristics of regional emission period** when selected the source profiles.

This paper preliminarily explored the impact of emission source profiles on the simulation of PM$_{2.5}$ components, the detail about how to select the source profile will be further studied in our future work, to provide some

new ideas for improving the uncertainty of model simulation.

1  SPECIATE- U.S. Environmental Protection Agency's (EPA) SPECIATE database, https://www.epa.gov/air-emissions-modeling/speciate.
2  SAPPC- SPAP database and published source profiles in China; SPAP-database of Source Profiles of Air Pollution, http://www.nkspap.com:9091/.

**Reference:**

Bi, X., Dai, Q., Wu, J., Zhang, Q., Zhang, W., Luo, R., Cheng, Y., Zhang, J., Wang, L., Yu, Z., Zhang, Y., Tian, Y., Feng, Y.: Characteristics of the main primary source profiles of particulate matter across China from 1987 to 2017, Atmos. Chem. Phys., 19, 3223-3243, https://doi.org/10.5194/acp-19-3223-2019, 2019.

Dai, Q., Bi, X., Song, W., Li, T., Liu, B., Ding, J., Xu, J., Song, C., Yang, N., Schulze, B. C., Zhang, Y., Feng, Y., Hopke, P. K.: Residential coal combustion as a source of primary sulfate in Xi'an, China, Atmos. Environ., 196, 66-76, https://doi.org/10.1016/j.atmosenv.2018.10.002, 2019.

Ding, X., Li, Q., Wu, D., Wang, X., Li, M., Wang, T., Chen, J.: Direct observation of sulfate explosive growth in wet plumes emitted from typical coal-fired stationary sources, Geophy. Res. Lett., 48, e2020GL092071, https://doi.org/10.1029/2020GL092071, 2021.

Hsu, Y., Divita, F., Dorn, J.: SPECIATE 5.0 - Speciation Database Development Documentation, Final Report, M. MENETREZ, Abt Associates Inc./Office of Research and Development/U.S. Environmental Protection Agency Research Triangle Park, NC27711, https://www.epa.gov/sites/default/files/2019-07/documents/speciate_5.0.pdf, 2019.

Liu, Y., Zhang, W., Bai, Z., Yang, W., Zhao, X., Han, B., Wang, X.: China Source Profile Shared Service (CSPSS): The Chinese $PM_{2.5}$ Database for Source Profiles, Aerosol Air Qual. Res., 17, 1501-1514, https://doi.org/10.4209/aaqr.2016.10.0469, 2017.

NBS: China Statistical Yearbook 2021, http://www.stats.gov.cn/tjsj/ndsj/2021/indexch.htm, last access: 2022.

Pan, D., Ran, Y., Bao, J., Wu, H., Huang, R., Yang, L.: Emission and Formation Characteristics of Aerosols from AmmoniaBased Wet Flue Gas Desulfurization, Energ. Fuel., 30, 666-673, https://doi.org/10.1021/acs.energyfuels.5b01677, 2016.

Pernigotti, D., Belis, C. A., Spano, L.: SPECIEUROPE: The European data base for PM source profiles, Atmos. Pollut. Res., 7, 307-314, https://doi.org/10.1016/j.apr.2015.10.007, 2016.

Reff, A., Bhave, P. V., Simon, H., Pace, T. G., Pouliot, G. A., Mobley, J. D., Houyoux, M.: Emissions Inventory of $PM_{2.5}$ Trace Elements across the United States, Environ. Sci. Technol., 43, 5790-5796, http://doi.org/10.1021/es802930x, 2009.

Simon, H., Beck, L., Bhave, P. V., Frank, D., Hsu, Y., Luecken, D., Mobley, J. D., Pouliot, G. A., Reff, A., Sarwar, G., Strum, M.: The development and uses of EPA's SPECIATE datebase, Atmos. Pollut. Res., 1, 196-206, https://doi.org/10.5094/APR.2010.026, 2010.

Yan, Q., Kong, S., Yan, Y., Liu, H., Wang, W., Chen, K., Yin, Y., Zheng, H., Wu, J., Yao, L., Zeng, X., Chen, Y., Zheng, S., Wu, F., Niu, Z., Zhang, Y., Zheng, M., Zhao, D., Liu, D., Qi, S.: Emission and simulation of primary fine andsubmicron particles and water-soluble ions from domestic coal combustion in China, Atmos. Environ., 224, 117308,

http://doi.org/10.1016/j.atmosenv.2020.117308, 2020.

Zhang, J., Wu, J., Lv, R., Song, D., Huang, F., Zhang, Y., Feng, Y.: Influence of Typical Desulfurization Process on Flue Gas Particulate Matter of Coal-fired Boilers (In Chinese), Environ. Sci., 41, 4455-4461, https://doi.org/10.13227/j.hjkx.202003193, 2020.

**Minor concern 1:**

*Line 21 and Line 27, there are two notes for CTM in one paragraph, which appear to be repetitive.*

**Response:**

We have deleted the duplicated notes for CTM in our manuscript.

> 20        The chemical transport model (CTM) is an essential tool for air quality prediction
> 21    and management, widely used in air pollution control and health risk assessment.
> 22    However, the current models do not perform very well in simulating $PM_{2.5}$ components.
> 23    Studies suggested that the uncertainties of model chemical mechanism, source emission
> 24    inventory and meteorological field can cause inaccurate simulation results. Still, the
> 25    emission source profile of $PM_{2.5}$ has not been fully taken into account in current
> 26    numerical simulation. This study aims to answer (1) Whether the variation of source
> 27    profile adopted in  (CTMs has an impact on the simulation

**Minor concern 2:**

*Line 57-59, the references are verbose.*

**Response:**

We have removed redundant references in our manuscript.

| 57 | adverse impact on human health (Shi et al., 2018) and ecosystem  |
| 58 | , such as acid rain in southwest China (Han et al., 2019), food security |
| 59 | (Zhou et al., 2018), etc. ↵ |

**Minor concern 3:**

*Line 111-113, It is not clearly explained the role of source profiles in CTMs.*

**Response:**

We have added extra explain in our manuscript and cited the source.

| 111 | In particular, the emission source profile of PM$_{2.5}$ (Hereinafter referred to as |
| 112 | "source profile"), creating speciated emission inventories for CTMs (Hsu et al., 2019), |
| 113 | has not been fully taken into account in the current numerical simulation . In |

Reference: Hsu, Y., Divita, F., Dorn, J.: SPECIATE 5.0 - Speciation Database Development Documentation, Final Report, M. MENETREZ, Abt Associates Inc./Office of Research and Development/U.S. Environmental Protection Agency Research Triangle Park, NC27711, https://www.epa.gov/sites/default/files/2019-07/documents/speciate_5.0.pdf, 2019.

**Minor concern 4 and 5:**

*Line 257: "The detailed information on" should be "The information of…"*

*Line 259: "Coefficient Divergence (CD)" would be appropriate.*

**Response:**

We have replaced the sentence with the correct expression in our manuscript. New line is in 261-263.

261  CMAQ_SPA and CMAQ_SPE. The detailed information  source profiles is shown

262  in Figure S1. To determine the similarity between the two groups of source profiles,

263  Coefficient  Divergence (CD) is calculated using the following formula

264  (Wongphatarakul et al., 1998):

**Minor concern 6:**

*In the supplementary material, Fig. S1, the author selected code 91041, 900162.5, 91155, 91022 and 91162 as SPECIATE source profiles for simulation. Detailed information of these source profiles need be provided by authors.*

**Response:**

We have added a table (Table S26) in supplementary material to show the detail information of these source profiles.

Table S26    The selected information of source profile in SPECIATE and SPAPPC database

| Code | Profile Name | Controls | Profile Date | Profile Notes | Keywords |
|---|---|---|---|---|---|
| 91041[a] | Draft Sub-Bituminous Combustion - Composite | Mixture of Baghouse, None, Electrostatic Precipitator, Wet Scrubber, Mechanical Collectors, Dry Lime Scrubber, Ammonia Injection | 2006-5-24 | Replaced by Profile 91110. Median of Profiles 3191, 3192, 3690, 3694, and 3700. | Sub-Bituminous Coal Combustion; PM Composite |

| | | | | | |
|---|---|---|---|---|---|
| 900162.5[b] | Industrial Manufacturing - Average | Not Applicable | 1989-1-5 | Average profile developed from original profiles representing the source category group 3xxxxxxx. | INDUSTRIAL |
| 91155[c] | Residential Coal Combustion - Composite | Uncontrolled | 2009-7-12 | Median of Profiles 3761, 432012.5 | Residential Coal Combustion; Inventory speciation |
| 91022[a] | Draft On-road Gasoline Exhaust - Composite | Mixture of Catalytic converter and Not available | 2006-5-24 | Replaced by Profile 91122. Median of Profiles 311072.5, 3517, 3884, 3892, 3904, 3947, 3951, 3955, 3959, and 4558. | On-road Gasoline Exhaust; PM Composite |
| 91162[c] | LDDV Exhaust - Composite | Mixture of Catalytic converter and Not available | 2009-7-12 | Median of Profiles 321042.5, 3912, 3963, 4675 | LDDV Exhaust; Inventory speciation |
| Local | PP | Mixture of Baghouse, None, Electrostatic Precipitator, Wet Scrubber, Mechanical Collectors, Dry Lime Scrubber, | | Average of profiles power and heating power plant | |
| Local | IN | Wet Scrubber, Dry Lime Scrubber, | | Average of profiles steel, metallurgy, cement, glass, | |

| | | | |
|---|---|---|---|
| | | | industrial boiler |
| Local | TR | Mixture of Catalytic converter | Average of profiles gasoline, diesel, gasoline-diesel exhaust |
| Local | RE | | Average of profiles civil boiler |

a, Hsu, Ying, Randy Strait, Stephen Roe, David Holoman. 2006. 'SPECIATE 4.0 Speciation database development document - Final Report', Prepared for US EPA, RTP, NC, EPA Contract Nos. EP-D-06-001, Work Assignment Numbers 0-03 and 68-D-02-063, WA 4-04 and WA 5-05, by E.H. Pechan & Associates, Incorporation, Durham, NC. https://www.epa.gov/sites/production/files/2015-10/documents/speciatedoc_1206.pdf.

b, Shareef, G. S. Engineering Judgement, Radian Corporation. August 1987.

c, Reff, Adam, Prakash V Bhave, Heather Simon, Thompson G Pace, George A Pouliot, J David Mobley, and Marc Houyoux. 2009. 'Emissions Inventory of $PM_{2.5}$ Trace Elements across the United States', Environmental Science & Technology, 43, no. 15: 5790-96. DOI: 10.1021/es802930x.

We appreciate it very much for these good suggestions and have done it according to your ideas. We also uploaded *the revised manuscript and supplementary materia*l in the attached document.

---

## Author Comment (AC3)

**Response to Anonymous Referee #2's comments on manuscript egusphere-2022-895**

We thank the reviewer for the immensely helpful comments. In response, we have carefully addressed the referee's concerns with this work. Please see point-by-point response to the comments and the revised manuscript for details. The reviewer's comments are shown in black *italics*. Our replies are shown in indented black text.

*The manuscript investigates the sensitivity of simulated $PM_{2.5}$ and its components' concentrations to the uncertainties in the component-specified $PM_{2.5}$ source emission inventories using the CMAQ chemical transport model. The relatively-complete chemical components, including Al, Ca, Cl, EC, Fe, K, Mg, Mn, Na, OC Si, $NH_4^+$, $NO_3^-$, $SO_4^{2-}$, and others, are taken into account in the emission inventory used. The authors showed that the influence of the relative contributions of different components to the total $PM_{2.5}$ emission (denoted as source profile changes in the manuscript) on simulated $PM_{2.5}$ concentration was insignificant, but its impact on $PM_{2.5}$ components could not be ignored. They also showed that these source profile changes caused the variations in simulated gaseous pollutants' concentrations. While such kind of model experiment should be a welcome addition to the literature on air quality model simulation, I do have concerns that the data and methodology used in this study would be*

*sensible (or well introduced) and the conclusions applicable to the simulations done by other chemical transport models with different chemical and physical modules. Therefore, I cannot recommend publication the current version of this manuscript in GMD.*

*The major issues are follows:*

*1. What is the grid resolution of the MEIC emission inventory that was used for the model simulation in this study? Is the resolution sufficiently fine for the Dom3 (4 km× 4km) simulation? What does the area marked in green in Fig. 1 refer to? No information on the regional distributions of either PM$_{2.5}$ emission sources or their simulated concentrations is provided in the manuscript. Are all the 10 monitoring sites located in the cities of Dom3? Is there any site that is located near the desert area? Were the mineral dust emissions taken into account in the simulation?*

**Response:**

Thank you for your reminder. More description of simulation area is placed in Fig. 1 and emission information in Fig. S2 of the revised manuscript. To address the reviewer's comment, additional interpretation has been made.

The grid resolution of the MEIC emission inventory was 0.25°×0.25°; We extracted the emissions from the original national

emission inventory and reprocessed the emissions into 36km×36km, 12km×12km, and 4km×4km grids for Domains 1, 2, and 3, respectively. The Inventory Spatial Allocate Tool (ISAT) was used to provide grided $PM_{2.5}$ emission inventory for the simulations. Considering the purpose of this paper is to explore how much the source profile changes will affect the simulation results, the resolution of the emission inventory is enough. For different scenarios, other modeling conditions remain the same except for the component-specified $PM_{2.5}$ source emission inventories changed.

The area marked in green in Fig. 1 is Tianjin city in the third domain (Dom 3). The third domain with a horizontal resolution of 4 km×4 km mainly focuses on Tianjin region which is marked in Fig. RF1(b) as follow (In the revised manuscript, we have replaced Fig. 1 with Fig.RF1 below to make it more clearly).

[Figure]

Fig. RF1 Modeling domains of the CMAQ model. (a) The three nested domains in CMAQ model; (b) Land use and observation sites of Dom3.

Data source of Land use: GLOBELAND30, www.globeland30.org, National Geomatics Center of China.

[Figure]

Fig. RF2 The regional distribution of PM$_{2.5}$ emission sources. (a) coal-fired power plant; (b) industry process; (c) transportation sector; (d) residential coal combustion.

The information of regional distribution of PM$_{2.5}$ emission sources are shown in Fig. RF2. In the revised manuscript, we have also provided the regional distributions of PM$_{2.5}$ emission sources (Fig. S2) in the supplementary material.

All the monitoring sites locate in the third domain which is shown in Fig. RF1(b). No sites are located in desert areas and the dust emissions are not taken into account in our simulation as the study region is far away from the deserts. The land use type of Dom3 is shown in Fig. RF1(b).

**To fully address the reviewer's comment, additional interpretation has been made as follows:**

In chemical transport models such as CMAQ, GEOS-Chem, CAMx, the $PM_{2.5}$ emission inventory is speciated in the chemical-composition dimension (Reff et al., 2009). Some commonly used emission inventories are listed in Table RF1. Different CTMs and their aerosol module have different regulations on $PM_{2.5}$ species types. Pollutants or species in emission inventory, especially for PM and VOCs, need to be speciated into chemical components for CTMs to match chemical mechanism. Taking CMAQ as an example, the aerosol module (AERO6) expands the definition of the PM Other species in earlier versions to include more detailed PM species (Chapel Hill, 2012); There are 18 $PM_{2.5}$ species in AERO6: OC, EC, $SO_4^{2-}$, $NO_3^-$, $NH_4^+$, $H_2O$, Na, Cl, NCOM, Al, Ca, Fe, Si, Ti, Mg, K, Mn, and Other. Other CTMs also have similar regulation, the classification of $PM_{2.5}$ species in mainstream CTMs are shown in Table RF2.

Table RF1 The air pollutants in emission inventory

| Scale | Name | Air pollutants |
|---|---|---|
| Global | EDGAR[1] | CO, $NO_x$, NMVOC, $CH_4$; $NH_3$, $NO_x$, $SO_2$; $PM_{10}$, $PM_{2.5}$, BC, OC |
| Global | EDGAR-HTAP[2] | $SO_2$, $NO_x$, CO, NMVOC, $PM_{10}$, $PM_{2.5}$, BC, OC, $NH_3$ |
| Global | GAINS[3] | $SO_2$, $NO_X$, VOC, PM, $NH_3$, $CO_2$, $CH_4$, $N_2O$ and the F-gases |
| Reginal | MIX, MEIC[4] | $SO_2$, $NO_x$, CO, NMVOC, $PM_{10}$, $PM_{2.5}$, BC, OC, $NH_3$, and $CO_2$ |
| Reginal | NEI[5] | CO, NOx, $PM_{10}$, $PM_{2.5}$, $SO_2$, VOC, $NH_3$ |
| Reginal | REAS[6] | $SO_2$, $NO_x$, CO, NMVOC, $PM_{10}$, $PM_{2.5}$, BC, OC, $NH_3$, and $CO_2$ |

| Reginal | EMEP[7] | $SO_2$, $NO_x$, NMVOCs, $PM_{2.5}$, $NH_3$ |
|---|---|---|

Note:

1, Emissions Database for Global Atmospheric Research (EDGAR) (1970-). https://edgar.jrc.ec.europa.eu/dataset_ap61

2, The Task Force Hemispheric Transport of Air Pollution (HTAP) (2000-2010). https://jeodpp.jrc.ec.europa.eu/ftp/jrc-opendata/EDGAR/datasets/htap_v2_2/ALL/

3, Greenhouse Gas and Air Pollution Interactions and Synergies (GAINS) (1990-).https://gains.iiasa.ac.at/gains/download/GAINS-tutorial.pdf.

4, A new Asian anthropogenic emission inventory (MIX) (2008, 2010); Multi-resolution Emission Inventory for China (MEIC) (2008-). http://meicmodel.org/

5, National emission inventory (NEI) (1970-), https://www.epa.gov/air-emissions-inventories/national-emissions-inventory-nei

6, Regional Emission inventory in Asia (REAS) (1950-2015). https://www.nies.go.jp/REAS/index.html#REASv3.2.1

7, European Monitoring and Evaluation Programme (EMEP) (1990-), https://www.eea.europa.eu/data-and-maps/dashboards/national-air-pollutant-emissions-data

Table RF2 The speciated allocation for $PM_{2.5}$ in mainstream CTMs

| CTMs | Aerosol module | $PM_{2.5}$ species |
|---|---|---|
| CMAQ[1] | AERO6 | Al, Ca, Cl, EC, Fe, K, Mg, Mn, Na, OC, Si, Ti, $NH_4^+$, $NO_3^-$, $SO_4^{2-}$, NCOM, Other, $H_2O$ |
| | AERO5 | OC, EC, $NO_3^-$, $SO_4^{2-}$, Other |
| GEOS-Chem[2] | aerosol.mod | Al, Ca, Cl, EC, Fe, K, Mg, Mn, Na, OC, Si, Ti, $NH_4^+$, $NO_3^-$, $SO_4^{2-}$, Other |
| WRF-Chem[3] | MADE, MOSAIC, MAM | OC, EC, $NO_3^-$, $SO_4^{2-}$, Ca, Na, Cl, $H_2O$, Other |
| CAMx[4] | CF | OC, EC, $NO_3^-$, $SO_4^{2-}$, $NH_4^+$, Cl, Na, Other |

Note:

1, Particulate matter (aerosols): PM using three lognormal sub-distributions, or modes, two interacting modes (Aitken and accumulation) represent $PM_{2.5}$

https://www.airqualitymodeling.org/index.php/CMAQ_version_5.0_(February_2010_release)_OGD#Aerosol_Module.

2, Particulate matter in GEOS-Chem: $PM_{2.5}$ = ( NH4 + NIT + SO4 ) * 1.10 + BCPI + BCPO + ( OCPO + ( OCPI * 1.05 ) ) * (OM/OC ratio) + DST1 + DST2 * 0.30 + SALA * 1.86 + SOA * 1.05. (NIT-NO3; BCPI and BCPO-EC; OCPO and OCPI-OC, NCOM; DST1-SO4, NH4, NO3, Cl, Na, K, Ca, Fe, Al, Si, Ti, Mn, Other, OC, NCOM; DST2-SO4, Cl, ASOL; SALA-SO4, Cl, Na, Mg, K, Ca.

http://wiki.seas.harvard.edu/geos-chem/index.php/GEOS-Chem_to_CMAQv5.0) http://wiki.seas.harvard.edu/geos-chem/index.php/Particulate_matter_in_GEOS-Chem.

As total $PM_{2.5}$ need to be speciated into its chemical components to match the chemical mechanism in CTMs , emission source profiles, which can provide "species" and "split factor" for $PM_{2.5}$, are key inputs for creating chemically-resolved emission inventories for CTMs. However, the actual emission source profile of $PM_{2.5}$ and the sensitivity of simulated components' concentrations to the variation in $PM_{2.5}$ source profiles are currently not well considered. In some studies, the $PM_{2.5}$ emission inventory is speciated using "None" or "simplified profiles" in the chemical-composition dimension (Reff et al., 2009). The corresponding literature-based data is presented in Table RF3 as bellow, we only selected the main components of $PM_{2.5}$ ($SO_4^{2-}$, $NO_3^-$, $NH_4^+$, OC and EC) as example here. The species allocation coefficients of $PM_{2.5}$ emission sources are commonly treated in the following ways: (1) allocated $PM_{2.5}$ components of source emissions by referring to source profile data in published literature or database like the US SPECIATE; (2) chemical profiles came from local measurement. With the development of production technology and the innovation of pollution treatment technology in recent years, some

source profiles have changed dramatically. The timeliness of $PM_{2.5}$ species allocation coefficients in current CTMs also need to be considered.

Although the number of $PM_{2.5}$ species and calculation method in different CTMs are different, no matter what kinds of CTMs, as long as it involves chemical components simulation for $PM_{2.5}$, the influence of source emission profiles should be considered. It remains unclear **whether the variations of adopted emission source profiles of $PM_{2.5}$ had influence on the CTMs' performance and how much the influence would be and how it works**. The purpose of this paper is to explore how much the $PM_{2.5}$ emission source profile changes will affect the simulation results. Taking CMAQ (one of the most widely used CTMs) and MEIC (a high-resolution inventory of anthropogenic air pollutants in China) as the carrier, we tested the sensitivity of the simulated chemical components to the variation of source profiles. The same kind of experiment is also applicable to other CTMs and emission inventories (e.g. NEI, EEI, REAS, HATP, etc.).

Table RF3 The adopted source profile and simulation result for different CTMs from published literatures

| The component proportion in source profile | PM$_{2.5}$ components | Model | NMB | R | Study area | Period | Reference |
|---|---|---|---|---|---|---|---|
| 9% | SO$_4^{2-}$ | CMAQv4.7.1 | -45% | 0.73 | Eastern China | 2010 | (Cheng et al., 2015) |
| 1% | NO$_3^-$ | | 29% | 0.82 | | | |
| Not explicitly Specified | SO$_4^{2-}$ | CMAQv4.7.1 | -4.5% | 0.87 | Qing Dao | Jan. 2016 | (Zhang et al., 2017) |
| | NO$_3^-$ | | 10% | 0.87 | | | |
| | NH$_4^+$ | | -6% | 0.9 | | | |
| Not explicitly Specified | SO$_4^{2-}$ | CMAQv5.0.1 | -54% | 0.6 | Northern China | 2013 | (Zheng et al., 2015) |
| | NO$_3^-$ | | -40% | 0.8 | | | |
| | NH$_4^+$ | | -58% | 0.7 | | | |
| | OC | | -25% | 0.8 | | | |
| | EC | | 196% | 0.6 | | | |
| | SO$_4^{2-}$ | Revised CMAQ | 6% | 0.7 | | | |
| | NO$_3^-$ | | 6% | 0.8 | | | |
| | NH$_4^+$ | | -4% | 0.8 | | | |
| | OC | | -28% | 0.7 | | | |
| | EC | | 183% | 0.6 | | | |
| Not explicitly Specified | SO$_4^{2-}$ | WRF-Chem3.6.1 | -84% | 0.31 | Nanjing | Jan. 2017 | (Sha et al., 2019) |
| | | | -71% | 0.26 | | Apr. 2017 | |
| | NO$_3^-$ | | 45% | 0.51 | | Jan. 2017 | |
| | | | 67% | 0.32 | | Apr. 2017 | |
| | NH$_4^+$ | | -34% | 0.27 | | Jan. 2017 | |
| | | | -13% | 0.31 | | Apr. 2017 | |

| | | | | | | | |
|---|---|---|---|---|---|---|---|
| Not explicitly Specified | $SO_4^{2-}$ | CMAQv5.0.2 | -41% | 0.82 | Qing Dao | Dec. 2015 ~ Jan. 2016 | (Gao et al., 2020) |
| | $NO_3^-$ | | 41% | 0.83 | | | |
| | $NH_4^+$ | | -5% | 0.83 | | | |
| Not explicitly Specified | $SO_4^{2-}$ | RAQMS | -4% | 0.83 | Beijing | Feb. to Mar. 2014 | (Li et al., 2020) |
| | $NO_3^-$ | | -4% | 0.77 | | | |
| | $NH_4^+$ | | 4% | 0.81 | | | |
| | OC | | -39% | 0.92 | | | |
| | EC | | -9% | 0.81 | | | |
| Not explicitly Specified | $SO_4^{2-}$ | CMAQv5.0.1 | -56%~-29% | - | China | 2013 | (Shi et al., 2017) |
| | $NO_3^-$ | | -47%~19% | | | | |
| | $NH_4^+$ | | -44%~1 | | | | |
| Not explicitly Specified | $SO_4^{2-}$ | CMAQv4.7 | -16% and -6% | - | USA | Jan. 2006 | (Foley et al., 2010) |
| | | | -19%~-0.2% | | | Aug. 2006 | |
| | $NO_3^-$ | | -5% and 1% | | | Jan. 2006 | |
| | $NH_4^+$ | | 13% and 14% | | | Jan. 2006 | |
| | | | 15% and -6% | | | Aug. 2006 | |
| | OC | | -20% | | | Jan. 2006 | |
| | | | -49% | | | Aug. 2006 | |
| | EC | | -25% | | | Jan. 2006 | |
| | | | -32% | | | Aug. 2006 | |
| 9% | $SO_4^{2-}$ | CMAQv4.5.1 | -34%~7% | - | USA | Jan. 2002 | (Liu et al., 2010) |
| | | | -18%~-37% | | | Jul. 2002 | |
| 1% | $NO_3^-$ | | 16%~118% | | | Jan. 2002 | |
| | | | -69%~88% | | | Jul. 2002 | |
| 0% | $NH_4^+$ | | -0.5%~61% | | | Jan. 2002 | |

| | | | | | | | |
|---|---|---|---|---|---|---|---|
| | | | -43%~53% | | | Jul. 2002 | |
| 30% | OC | | -4%~13% | | | Jan. 2002 | |
| | | | -71%~-64% | | | Jul. 2002 | |
| 24% | EC | | -16%~18% | | | Jan. 2002 | |
| | | | -39%~38% | | | Jul. 2002 | |
| 9% | $SO_4^{2-}$ | CMAQv4.5.1 | 5% | 0.7 | South Eastern USA | Jan. 2002 | (Zhang et al., 2013) |
| | | CAMx-4.4.2 | 33% | 0.6 | | | |
| | | CMAQv4.5.1 | -39% | 0.5 | | Jul. 2002 | |
| | | CAMx-4.4.2 | -9% | 0.6 | | | |
| 1% | $NO_3^-$ | CMAQv4.5.1 | 46% | 0.8 | | Jan. 2002 | |
| | | CAMx-4.4.2 | -21% | 0.8 | | | |
| | | CMAQv4.5.1 | -62% | 0.2 | | Jul. 2002 | |
| | | CAMx-4.4.2 | -80% | 0.2 | | | |
| 0% | $NH_4^+$ | CMAQv4.5.1 | -7% | 0.8 | | Jan. 2002 | |
| | | CAMx-4.4.2 | -8% | 0.7 | | | |
| | | CMAQv4.5.1 | -52% | 0.7 | | Jul. 2002 | |
| | | CAMx-4.4.2 | -45% | 0.7 | | | |
| 30% | OC | CMAQv4.5.1 | -15% | 0.8 | | Jan. 2002 | |
| | | CAMx-4.4.2 | -18% | 0.8 | | | |
| | | CMAQv4.5.1 | -73% | 0.7 | | Jul. 2002 | |
| | | CAMx-4.4.2 | -47% | 0.7 | | | |
| 24% | EC | CMAQv4.5.1 | -9% | 0.7 | | Jan. 2002 | |
| | | CAMx-4.4.2 | 5% | 0.7 | | | |
| | | CMAQv4.5.1 | -47% | 0.4 | | Jul. 2002 | |
| | | CAMx-4.4.2 | -33% | 0.4 | | | |

| | | | | | | | |
|---|---|---|---|---|---|---|---|
| 9% | $SO_4^{2-}$ | CMAQv5.0 | 0.7% and -31% | 0.85 | USA | 1990-2010 | (Xing et al., 2015) |
| | | | -2% | 0.61 | Europe | | |
| 1% | $NO_3^-$ | | 56%~59% | 0.66 | USA | | |
| | | | -6% | 0.70 | Europe | | |
| 0% | $NH_4^+$ | | -13% | 0.52 | USA | | |
| | | | 34% | 0.62 | Europe | | |
| Not explicitly Specified | $SO_4^{2-}$ | CMAQv4.5 | -16% | 0.82 | USA | 2002~2008 | (Friberg et al., 2016) |
| | $NO_3^-$ | | 72% | 0.64 | | | |
| | $NH_4^+$ | | 13% | 0.68 | | | |
| | OC | | -30% | 0.39 | | | |
| | EC | | -22% | 0.5 | | | |
| Not explicitly Specified | $SO_4^{2-}$ | CMAQv5.0.2 | -50%~29% | - | California | 2013 | (Chen et al., 2020) |
| | $NO_3^-$ | | -27%~48% | | | | |
| | $NH_4^+$ | | -32%~130% | | | | |
| | OC | | -35%~13% | | | | |
| | EC | | 0~43% | | | | |
| The emission inventories for $SO_4^{2-}$, $NO_3^-$ and $NH_4^+$ emitted from residential coal combustion were established | $SO_4^{2-}$ | GEOS-Chem v11-01 | Quite different | | China | 2015 | (Yan et al., 2020) |
| | $NO_3^-$ | | | | | | |
| | $NH_4^+$ | | | | | | |
| Not explicitly Specified | $SO_4^{2-}$ | WRF-Chem | MB=5μg/m$^3$ | RMSE=12.5μg/m$^3$ | BTH, China | 2014 | (Li et al., 2018) |
| | $NO_3^-$ | | MB=-0.3μg/m$^3$ | RMSE=14.3μg/m$^3$ | | | |

| | NH₄⁺ | | MB=-0.4μg/m³ | RMSE=8.2μg/m³ | | | |
|---|---|---|---|---|---|---|---|
| Local source profile | SO₄²⁻ | CAMx | | 0.32 | Tianjin | 2017-2018 | (Ma et al., 2022) |
| | NO₃⁻ | | | 0.59 | | | |
| | OC | | | 0.27 | | | |
| | EC | | | 0.47 | | | |

*2. At the beginning of Sect. 2.2 it is stated that in addition to SPA and SPE, the PM$_{2.5}$ emission source profile database from published literature was used. Where and what are the final, merged emission source profiles used in this study? The simulated PM$_{2.5}$ and its components' concentrations using CMAQ_SPA are compared with those using CMAQ_SPE. However, no comparison with observed PM$_{2.5}$ components' concentrations at the monitoring sites has been made to show the advantage of the SPA over the SPE.*

**Response:**

More descriptions of source profiles are shown in Fig. S1 and Table S26 of our revised supplementary material. In addition, to address the reviewer's comment, we added an extra explanation as follows:

In this study, for SPE, the selected source profile of each source category group was the average/median profile developed from original profiles in SPECIATE database. The source profile codes for power plant (PP), industrial process (IN), residential coal combustion (RE), and transportation sector (TR) are 900162.5, 91155, 91022 and 91162, respectively. Please see Table RF4 for details. For SPA, the selected source profiles were from database of Source Profiles of Air Pollution, they are also available in our previous paper (Bi et al., 2019). The detailed information of source profiles as shown in the following Fig. RF3 and

Table RF4. They have also been updated in the revised supplementary material (Fig. S1 and Table S26).

[Figure]

Fig. RF3    The selected speciation profile of PM$_{2.5}$ for case CMAQ_SPE and CMAQ_SPA
In SPE, the selected source profiles were average profile developed from original profiles of the source category group in SPECIATE database, the power plant (PP) source profile code was 91041, industrial process (IN) was 900162.5, Residential coal combustion (RE) was 91155, Transportation sector (TR) was 91022 and 91162. In SPA, the selected source profiles were from SPAPPC database which were measured from local emission sources.

Table RF4    The selected information of source profile in SPECIATE and SPAPPC database

| Code | Profile Name | Controls | Profile Date | Profile Notes | Keywords |
|---|---|---|---|---|---|
| 91041[a] | Draft Sub-Bituminous Combustion - Composite | Mixture of Baghouse, None, Electrostatic Precipitator, Wet Scrubber, Mechanical Collectors, Dry Lime Scrubber, | 2006-5-24 | Replaced by Profile 91110. Median of Profiles 3191, 3192, 3690, 3694, and 3700. | Sub-Bituminous Coal Combustion; PM Composite |

| | | Ammonia Injection | | | |
|---|---|---|---|---|---|
| 900162.5[b] | Industrial Manufacturing - Average | Not Applicable | 1989-1-5 | Average profile developed from original profiles representing the source category group 3xxxxxxx. | INDUSTRIAL |
| 91155[c] | Residential Coal Combustion - Composite | Uncontrolled | 2009-7-12 | Median of Profiles 3761, 432012.5 | Residential Coal Combustion; Inventory speciation |
| 91022[a] | Draft On-road Gasoline Exhaust - Composite | Mixture of Catalytic converter and Not available | 2006-5-24 | Replaced by Profile 91122. Median of Profiles 311072.5, 3517, 3884, 3892, 3904, 3947, 3951, 3955, 3959, and 4558. | On-road Gasoline Exhaust; PM Composite |
| 91162[c] | LDDV Exhaust - Composite | Mixture of Catalytic converter and Not available | 2009-7-12 | Median of Profiles 321042.5, 3912, 3963, 4675 | LDDV Exhaust; Inventory speciation |
| Local | PP | Mixture of Baghouse, None, Electrostatic Precipitator, Wet Scrubber, Mechanical Collectors, Dry Lime Scrubber, | | Average of profiles power and heating power plant | |

| | | | |
|---|---|---|---|
| Local | IN | Wet Scrubber, Dry Lime Scrubber, | Average of profiles steel, metallurgy, cement, glass, industrial boiler |
| Local | TR | Mixture of Catalytic converter | Average of profiles gasoline, diesel, gasoline-diesel exhaust |
| Local | RE | | Average of profiles civil boiler |

a, Hsu, Ying, Randy Strait, Stephen Roe, David Holoman. 2006. 'SPECIATE 4.0 Speciation database development document - Final Report', Prepared for US EPA, RTP, NC, EPA Contract Nos. EP-D-06-001, Work Assignment Numbers 0-03 and 68-D-02-063, WA 4-04 and WA 5-05, by E.H. Pechan & Associates, Incorporation, Durham, NC. https://www.epa.gov/sites/production/files/2015-10/documents/speciatedoc_1206.pdf.
b, Shareef, G. S. Engineering Judgement, Radian Corporation. August 1987.
c, Reff, Adam, Prakash V Bhave, Heather Simon, Thompson G Pace, George A Pouliot, J David Mobley, and Marc Houyoux. 2009. 'Emissions Inventory of PM$_{2.5}$ Trace Elements across the United States', Environmental Science & Technology, 43, no. 15: 5790-96. DOI: 10.1021/es802930x.

We agree with that it's necessary to compare with observed values as to discuss the model performance, but here we mainly tried to answer (1) Whether the variation of source profile adopted in CTMs has an impact on the simulation of PM$_{2.5}$ chemical components? (2) How much does it impact? (3) How does the impact work? so we didn't do comparison with observed PM$_{2.5}$ components' concentrations at the monitoring sites. By comparing SPA and SPE source profiles, our purpose is to show that the source profile of same source category can vary greatly. Different simulation scenarios were designed and the sensitivity of components

simulation results to PM$_{2.5}$ sources profile was explored through with chemical components of source profiles perturbation. We found that the sensitivity of simulation results to source profile changes should be considered in numerical simulation. In fact, the emission inventory, and the selection of simulated area here only are the carrier to conduct this study. Any two different groups of source profiles could be used for designing comparative experiments.

Thanks for the reviewer's valued reminder. There are several factors will influence model performance like the emission, model mechanism, meteorological modeling performance. By providing accurate and time-sensitive source profile to make the model inputs more accurate and the interference from sources on the uncertainty of simulation results is eliminated somehow. In the next work, we will use different source profile for simulation and compare the simulation results with local measured PM$_{2.5}$ components.

*3. While the MEIC inventory includes four categories, i.e. power plants (PP), industrial processes (IN), residential emission (RE) and transport sector (TR), the SPA and SPE are shown to have different categories (perhaps more than the MEIC does). How were these chemical PM$_{2.5}$ emission source profiles combined to match the MEIC categories? For instance, the residential emission should include not only coal burning but*

*also straw burning, and the latter was seemly not considered in the simulations. Also, the chemical profiles for gasoline and diesel oil in the transport sector might be different.*

**Response:**

Thank you for your valued advices. More descriptions have been added in Section 6 (Lines 557-560 in the revised manuscript).

556       Our study tentatively discussed the impact mechanism of emission source profiles

557    on $PM_{2.5}$ components simulation results in CTMs. In the next work, we will use

558    different source profile for simulation, compare the simulation results with local

559    measured $PM_{2.5}$ components and discuss the influence of sub-source profiles variation

560    on the simulation results. In addition, the size distribution, mixing state, aging and

Just as you mentioned, in the database of Source Profiles of Air Pollution (SPAP) and U.S. Environmental Protection Agency's (EPA) SPECIATE database, these four source categories (coal-fired power plant, industry process, transportation sector and residential coal combustion) contain a series of sub-categories. But unfortunately, the MEIC inventory does not include the corresponding sub-categories. So we take the average values of all source profiles in each source category as representing source profile, the details could be seen in our previous work (Bi et al., 2019); Then multiply inventory emissions by profile fraction to get emissions of specific chemical compounds. The general step for speciation is shown in Fig. RF4.

[Figure]

Fig. RF4 Speciation in general step

Source: International Emissions Inventory Conference. SPECIATE and using the Speciation Tool to prepare VOC and PM chemical speciation profiles for air quality modeling, p31. https://www.epa.gov/sites/default/files/2017-10/documents/speciate_speciationtool_training.pdf.

In our study, we found that the simulated concentration of $PM_{2.5}$ components, not only primary components but also secondary components, indeed varied with the source profiles. The representativeness and timeliness of source profile should be considered due to the underappreciated impact of emission source profiles on the simulation of $PM_{2.5}$ components. Thank you for your valuable comments, we will deeply discuss the influence of sub-source profiles on the simulation results in the follow-up study.

*4. How are the dynamic, microphysical and chemical processes of aerosols treated in the CMAQ model used for this study? Are the size distribution, mixing state, aging and solubility taken into account for different aerosol components? By which molecular form are the chemical components (Al, Ca, Cl, EC, Fe, K, Mg, Mn, Na, OC Si, $NH_4^+$, $NO_3^-$, and $SO_4^{2-}$) emitted from the sources? Taking elemental Ca as an example, it should be emitted*

*by CaO, CaCO₃, CaSO₄, or other compound, rather than merely by the cation $Ca^{2+}$. The similar principle applies for anions ($NO_3^-$ and $SO_4^{2-}$). The difference in the exiting form of these emitted aerosol components might have large impacts on the thermodynamic equilibrium of ions in liquid aerosols and clouds.*

**Response:**

Thank you for the reviewer's questions, more descriptions have been added in Section 6 (Lines 560-563 in the revised manuscript).

556       Our study tentatively discussed the impact mechanism of emission source profiles
557    on PM₂.₅ components simulation results in CTMs. In the next work, we will use
558    different source profile for simulation, compare the simulation results with local
559    measured PM₂.₅ components and discuss the influence of sub-source profiles variation
560    on the simulation results. In addition, the size distribution, mixing state, aging and
561    solubility for different aerosol components might have something to do with source
562    profile, how much the influence of source profile changes on these physical and
563    chemical process, is deserved to do in the future.

Please see the point-by-point response as follows:

**For:** *How are the dynamic, microphysical and chemical processes of aerosols treated in the CMAQ model used for this study?*

The key scientific algorithms simulating aerosol processes for the CCTM in CMAQ are: (1) aerosol removal by size-dependent dry deposition; (2) aerosol-cloud droplet interaction and removal by

precipitation; (3) new particle formation by binary homogeneous nucleation in a sulfuric acid/water vapor system; (4) the production of an organic aerosol component from gas-phase precursors; and (5) particle coagulation and condensation growth (Byun and Young, 1999).

The particle dynamics of aerosol distribution using three interacting lognormal distributions, or modes. Two modes (Aitken and accumulation) are generally less than 2.5µm in diameter while the coarse mode contains significant amounts of mass above 2.5µm. The equation of lognormal distribution is as follow:

$$n(\ln D) = \frac{N}{\sqrt{2\pi}\ln\sigma_g}\exp\left[-0.5\left(\frac{\ln\frac{D}{D_g}}{\ln\sigma_g}\right)^2\right] \dots\dots (1)$$

Where $N$ is the particle number concentration within the mode suspended in a unit volume of air, $D$ is the particle diameter, $D_g$ is the geometric mean diameter, $\sigma_g$ is the geometric standard deviation of modal distribution. A brief summary is described by (Binkowski and Roselle, 2003) and fully described by (Whitby and McMurry, 1997). The aerosol species of PM$_{2.5}$ in CMAQ are listed in Table RF5.

Table RF5 Aerosol species of PM$_{2.5}$ in CMAQ

| CMAQ species | Description |
|---|---|
| AECI, AECJ | Aitken (I) and accumulation (J) mode EC mass |
| APOCI, APOCJ | Aitken (I) and accumulation (J) mode OC mass |
| APNCOMI, APNCOMJ | Aitken (I) and accumulation (J) mode primary non-carbon |

| | |
|---|---|
| | organic matter mass |
| ASO4J | Accumulation (J) mode sulfate mass |
| ANO3J | Accumulation (J) mode nitrate mass |
| ACLJ | Accumulation (J) mode particulate chloride mass |
| ANH4J | Accumulation (J) mode particulate ammonium mass |
| ANAJ | Accumulation (J) mode sodium mass |
| AKJ | Accumulation (J) mode potassium mass |
| AMGJ | Accumulation (J) mode magnesium mass |
| ACAJ | Accumulation (J) mode calcium mass |
| AFEJ | Accumulation (J) mode iron mass |
| AMNJ | Accumulation (J) mode manganese mass |
| AALJ | Accumulation (J) mode aluminum mass |
| ASIJ | Accumulation (J) mode silicon mass |
| ATIJ | Accumulation (J) mode titanium mass |
| AH2OJ | Accumulation (J) mode particulate water mass |
| AOTHRJ | Accumulation (J) mode remaining unspeciated fine mode primary PM mass |

The aerosol microphysics i.e. coagulation, condensation, new particle formation, deposition, etc.) are considered in CMAQ using aero_subs.F, aero_depv.F, coags.f, in CCTM module correspondingly. The microphysical process and the related numerical simulation in subroutines called by the CMAQ driver are covered in more detail in the literatures (Binkowski and Roselle, 2003; Byun and Young, 1999).

The aerosol chemical species are listed in Table RF4. ISORROPIA v2.2 in the reverse mode are used to calculate the condensation/evaporation of volatile inorganic gases to/from the gas-phase concentrations of coarse particle surfaces. ISORROPIA v2.2 is also used in the forward mode to calculate instantaneous thermodynamic equilibrium between the gas and fine-particle modes. The equilibria and the associated constants are shown in Table RF6.

**Table RF6 Equilibrium relations and Constants**

| Number | Reaction | $K^0$ (298.15K) |
|--------|----------|---------|
| I1 | $Ca(NO_3)_{2(s)} \leftrightarrow Ca^{2+}_{(aq)} + 2NO^-_{3(aq)}$ | $6.067 \times 10^5$ |
| I2 | $Ca(Cl)_{2(s)} \leftrightarrow Ca^{2+}_{(aq)} + 2Cl^-_{(aq)}$ | $7.974 \times 10^{11}$ |
| I3 | $CaSO_4 \cdot 2H_2O_{(s)} \leftrightarrow Ca^{2+}_{(aq)} + SO^{2-}_{4\ (aq)} + 2H_2O$ | $4.319 \times 10^{-5}$ |
| I4 | $K_2SO_{4(s)} \leftrightarrow 2K^+_{(aq)} + SO^{2-}_{4\ (aq)}$ | $1.569 \times 10^{-2}$ |
| I5 | $KHSO_{4(s)} \leftrightarrow K^+_{(aq)} + HSO^-_{4(aq)}$ | 24.016 |
| I6 | $KNO_{3(s)} \leftrightarrow K^+_{(aq)} + NO^-_{3(aq)}$ | 0.872 |
| I7 | $KCl_{(s)} \leftrightarrow K^+_{(aq)} + Cl^-_{(aq)}$ | 8.680 |
| I8 | $MgSO_{4(s)} \leftrightarrow Mg^{2+}_{(aq)} + SO^{2-}_{4\ (aq)}$ | $1.079 \times 10^5$ |
| I9 | $Mg(NO_3)_{2(s)} \leftrightarrow Mg^{2+}_{(aq)} + 2NO^-_{3(aq)}$ | $2.507 \times 10^{15}$ |
| I10 | $Mg(Cl)_{2(s)} \leftrightarrow Mg^{2+}_{(aq)} + 2Cl^-_{(aq)}$ | $9.557 \times 10^{21}$ |
| I11 | $HSO^-_{4(aq)} \leftrightarrow H^+_{(aq)} + SO^{2-}_{4\ (aq)}$ | $1.015 \times 10^{-2}$ |
| I12 | $NH_{3(g)} \leftrightarrow NH_{3(aq)}$ | 57.64 |
| I13 | $NH_{3(aq)} + H_2O_{(aq)} \leftrightarrow NH^+_{4(aq)} + OH^-_{(aq)}$ | $1.805 \times 10^{-5}$ |
| I14 | $HNO_{3(g)} \leftrightarrow H^+_{(aq)} + NO^-_{3(aq)}$ | $2.511 \times 10^6$ |
| I15 | $HNO_{3(g)} \leftrightarrow HNO_{3(aq)}$ | $2.1 \times 10^5$ |
| I16 | $HCl_{(g)} \leftrightarrow H^+_{(aq)} + Cl^-_{(aq)}$ | $1.971 \times 10^6$ |
| I17 | $HCl_{(g)} \leftrightarrow HCl_{(aq)}$ | $2.5 \times 10^3$ |
| I18 | $H_2O_{(aq)} \leftrightarrow H^+_{(aq)} + OH^-_{(aq)}$ | $1.010 \times 10^{-14}$ |
| I19 | $Na_2SO_{4(s)} \leftrightarrow 2Na^+_{(aq)} + SO^{2-}_{4\ (aq)}$ | 0.4799 |
| I20 | $(NH_4)_2SO_{4(s)} \leftrightarrow 2NH^+_{4(aq)} + SO^{2-}_{4\ (aq)}$ | 1.817 |

| | | |
|---|---|---|
| I21 | $NH_4Cl_{(s)} \leftrightarrow NH_{3(g)} + HCl_{(g)}$ | $1.086 \times 10^{-16}$ |
| I22 | $NaNO_{3(s)} \leftrightarrow Na^+_{(aq)} + NO^-_{3(aq)}$ | 11.97 |
| I23 | $NaCl_{(s)} \leftrightarrow Na^+_{(aq)} + Cl^-_{(aq)}$ | 37.66 |
| I24 | $NaHSO_{4(s)} \leftrightarrow Na^+_{(aq)} + HSO^-_{4(aq)}$ | $2.413 \times 10^4$ |
| I25 | $NH_4NO_{3(s)} \leftrightarrow NH_{3(g)} + HNO_{3(g)}$ | $4.199 \times 10^{-17}$ |
| I26 | $NH_4HSO_{4(s)} \leftrightarrow NH^+_{4(aq)} + HSO^-_{4(aq)}$ | 1.383 |
| I27 | $(NH_4)_3H(SO_4)_{2(s)} \leftrightarrow 3NH^+_{4(aq)} + HSO^-_{4(aq)} + SO^{2-}_{4\ (aq)}$ | 29.72 |

Source: (Fountoukis and Nenes, 2007)

Besides that, for a higher computational efficiency, a VBS-style approach (four surrogate species with specific vapor pressures) is widely used in models; For the nonvolatile POA configuration, mass is tracked separately in terms of its carbon (OC) and non-carbon (NCOM) content. With this approach in AERO6, mass can be added to the non-carbon species to simulate the aging of POA in response to atmospheric oxidants. Details are shown in CMAQ users guide (chapter 6, https://github.com/USEPA/CMAQ/blob/main/DOCS/Users_Guide/CMAQ_UG_ch06_model_configuration_options.md#6.11_Aerosol_Dynamics) and the literature (Binkowski and Roselle, 2003).

**For**: *Are the size distribution, mixing state, aging and solubility taken into account for different aerosol components?*

Yes, they are all taken into account for different aerosol components. For size distribution, taking $PM_{2.5}$ as an example, except for a very small fraction of OC, EC and non-carbon organic matter are allocated in the Aitken mode, the rest are allocated in the accumulation mode. The size distribution of different $PM_{2.5}$ components are shown in Table RF7.

Table RF7 The size distribution of different $PM_{2.5}$ components

| Name | Aitken (I) | Accumulation (J) | Coarse (K) |
|------|-----------|------------------|------------|
| EC | 0.001 | 0.999 | 0 |
| OC | 0.001 | 0.999 | 0 |
| NCOM | 0.001 | 0.999 | 0 |
| $SO_4^{2-}$ | 0 | 1 | 0 |
| $NO_3^-$ | 0 | 1 | 0 |
| $Cl^-$ | 0 | 1 | 0 |
| $NH_4^+$ | 0 | 1 | 0 |
| Na | 0 | 1 | 0 |
| K | 0 | 1 | 0 |
| Mg | 0 | 1 | 0 |
| Ca | 0 | 1 | 0 |
| Fe | 0 | 1 | 0 |
| Mn | 0 | 1 | 0 |
| Al | 0 | 1 | 0 |
| Si | 0 | 1 | 0 |
| Ti | 0 | 1 | 0 |
| $H_2O$ | 0 | 1 | 0 |
| Other | 0 | 1 | 0 |

Source: AERO_EMIS.F in CCTM module

As regards as mixing state (hetchem.f in CMAQ), the empirical equations developed by Martin et al (Martin et al., 2003) are applied to determine the crystallization relative humidity (CRH) for a given mixture of sulfate, nitrate, and ammonium. Though those equations are validated only at 293K, they are applied at all ambient temperatures because

insufficient data exist to estimate the temperature dependence of the CRH of mixed sulfate-nitrate-ammonium particles.

As to aerosol aging, the subroutine of poaage.F in AERO module calculates oxidative aging of POA using the following reaction (Table RF8):

Table RF8 Oxidative aging of POA

| |
|---|
| POCRm ---> PNCOM (rate constant = koheff*[OH]) |
|         - POCRm = reduced primary organic carbon (molar concentration) |
| POMOC = (POC + NCOM)/POC |
|         - in other words: pimary OM/OC = (POC + PNCOM)/POC |
| PHOrat = (44/12 - POMOC)/(POMOC - 14/12) |
| Omoles = NCOM/(16 + PHOrat) if POMOC is between 14/12 and 44/12 |
| Omoles = NCOM/16 for POMOC larger than 44/12 |
|         - if OM/OC > 3.667, then POC is fully oxidized and all    NCOM is oxygen |
| Omoles = 0 for POMOC smaller than 14/12 |
|         - if OM/OC < 1.167, then POC is fully reduced and all NCOM is hydrogen |
| POCRm = POC/12 - Omoles |
| NOTE: POC was divided by 12 b/c we want moles of carbon atoms not moles of POC (since each carbon atom w\in the molecule is allowed to react) |

For solubility, the system modeled by ISORROPIA II consists of the following potential components: Gas phase: $NH_3(g)$, $HNO_3(g)$, $HCl(g)$, $H_2O(g)$; Liquid phase: $NH_4^+(aq)$, $Na^+(aq)$, $H^+(aq)$, $Cl^-(aq)$, $NO_3^-(aq)$, $SO_4^{2-}(aq)$, $HNO_3(aq)$, $NH_3(aq)$, $HCl(aq)$, $HSO_4^-(aq)$, $OH^-(aq)$, $H_2O(aq)$, $Ca^{2+}(aq)$, $K^+(aq)$, $Mg^{2+}(aq)$; Solid phase: $(NH_4)_2SO_4(s)$, $NH_4HSO_4(s)$, $(NH_4)_3H(SO_4)_2(s)$, $NH_4NO_3(s)$, $NH_4Cl(s)$, $NaCl(s)$, $NaNO_3(s)$, $NaHSO_4(s)$, $Na_2SO_4(s)$, $CaSO_4(s)$, $Ca(NO_3)_2(s)$, $CaCl_2(s)$, $K_2SO_4(s)$, $KHSO_4(s)$, $KNO_3(s)$, $KCl(s)$, $MgSO_4(s)$, $Mg(NO_3)_2(s)$, $MgCl_2(s)$;

where the subscripts (g), (aq), (s) denote gas, aqueous and solid, respectively.

**For**: *By which molecular form are the chemical components (Al, Ca, Cl, EC, Fe, K, Mg, Mn, Na, OC Si, $NH_4^+$, $NO_3^-$, and $SO_4^{2-}$) emitted from the sources? Taking elemental Ca as an example, it should be emitted by CaO, $CaCO_3$, $CaSO_4$, or other compound, rather than merely by the cation $Ca^{2+}$. The similar principle applies for anions ($NO_3^-$ and $SO_4^{2-}$). The difference in the exiting form of these emitted aerosol components might have large impacts on the thermodynamic equilibrium of ions in liquid aerosols and clouds.*

Generally, the PM samples emitted from the sources are collected on Teflon and quartz fiber filters and then sent for chemical component analysis. Elements analysis uses Teflon filters, common chemical analysis instruments are: inductively coupled plasma optical emission spectrometer (ICP-OES), inductively coupled plasma atomic emission spectrometer (ICP-AES), inductively coupled plasma mass spectrometer (ICP-MS) instruments and X-ray fluorescence. The total carbon (TC) mass in the samples are typically determined using thermal or thermal–optical methods. There are two widely utilized approaches to dividing OC and EC from TC, known as IMPROVE_A (from the Desert Research Institute– DRI) and NIOSH (method 5040; from the National Institute for Occupational Safety and Health – NIOSH), which are operationally defined by the time–

temperature protocols, and the OC–EC split point is determined by optical reflectance/transmittance (Ho et al., 2003; Bi et al., 2019). PM samples collected on the quartz fiber filters are normally used for the determination of water-soluble inorganic ions via different types of ion chromatography (IC) with high-capacity cation-exchange and anion-exchange columns. In addition, the molecular form of particulate matter emitted by pollution sources is difficult to measure. Hence, data form in emission source profiles are chemical components NOT chemical compounds.

The emission input files for CTMs are generated from data provided by emission inventories, only the species that are specifically defined in the chemical mechanism will be included in model inputs and outputs. PM need to be speciated into chemical components for CTMs to match chemical mechanism, and the emission source profiles can provide "species" and "split factor"(Detail is shown in Fig. RF3). **The species for PM$_{2.5}$ in mainstream CTMs** are listed in Table RF2. The process of modeling speciation requires components **rather than chemical compounds.**

Thank you for your valuable comments. Our study tentatively discussed the impact mechanism of emission source profiles on PM$_{2.5}$ components simulation results in CTMs. We found the influences are connected to model chemical mechanisms since the variation of species allocations in emission sources directly affected the thermodynamic

equilibrium system. We will continue exploring the influence of source profile changes on aerosol microphysical and chemical processes in a follow-up study.

Table RF2 The speciated allocation for $PM_{2.5}$ in mainstream CTMs

| CTMs | Aerosol module | $PM_{2.5}$ species |
|---|---|---|
| CMAQ[1] | AERO6 | Al, Ca, Cl, EC, Fe, K, Mg, Mn, Na, OC, Si, Ti, $NH_4^+$, $NO_3^-$, $SO_4^{2-}$, NCOM, Other, $H_2O$ |
| | AERO5 | OC, EC, $NO_3^-$, $SO_4^{2-}$, Other |
| GEOS-Chem[2] | aerosol.mod | Al, Ca, Cl, EC, Fe, K, Mg, Mn, Na, OC, Si, Ti, $NH_4^+$, $NO_3^-$, $SO_4^{2-}$, Other |
| WRF-Chem[3] | MADE, MOSAIC, MAM | OC, EC, $NO_3^-$, $SO_4^{2-}$, Ca, Na, Cl, $H_2O$, Other |
| CAMx[4] | CF | OC, EC, $NO_3^-$, $SO_4^{2-}$, $NH_4^+$, Cl, Na, Other |

Note:

1, Particulate matter (aerosols): PM using three lognormal sub-distributions, or modes, two interacting modes (Aitken and accumulation) represent $PM_{2.5}$

https://www.airqualitymodeling.org/index.php/CMAQ_version_5.0_(February_2010_release)_OGD#Aerosol_Module.

2, Particulate matter in GEOS-Chem: $PM_{2.5}$ = ( NH4 + NIT + SO4 ) * 1.10 + BCPI + BCPO + ( OCPO + ( OCPI * 1.05 ) ) * (OM/OC ratio) + DST1 + DST2 * 0.30 + SALA * 1.86 + SOA * 1.05. (NIT-NO3; BCPI and BCPO-EC; OCPO and OCPI-OC, NCOM; DST1-SO4, NH4, NO3, Cl, Na, K, Ca, Fe, Al, Si, Ti, Mn, Other, OC, NCOM; DST2-SO4, Cl, ASOL; SALA-SO4, Cl, Na, Mg, K, Ca.

http://wiki.seas.harvard.edu/geos-chem/index.php/GEOS-Chem_to_CMAQv5.0) http://wiki.seas.harvard.edu/geos-chem/index.php/Particulate_matter_in_GEOS-Chem.

3, Aerosols in WRF-Chem: PM using 3 or 7 log-normal modes, two interacting modes (Aitken and accumulation) represent $PM_{2.5}$.

https://ruc.noaa.gov/wrf/wrf-chem/wrf_tutorial_2018/Aerosols.pdf.

4, Aerosol Chemistry: $PM_{2.5}$ = PSO4 + PNO3 + PNH4 + PEC + NA + PCL + POA + SOA1 + SOA2 + SOA3 + SOA4 + SOPA + SOPB + FPRM + FCRS + (PFE + PMN + PK + PCA + PMG + PAL + PSI + PTI) (Fe, Mn, K, Ca, Mg, Al, Si and Ti are Optional Species).

https://camx-wp.azurewebsites.net/Files/CAMxUsersGuide_v7.20.pdf.

**Model species definitions**

| species name | species description | AE5 | AE6 |
|---|---|---|---|
| POC | organic carbon | Y | Y |
| PEC | elemental carbon | Y | Y |
| PSO4 | sulfate | Y | Y |
| PNO3 | nitrate | Y | Y |
| PMFINE | unspeciated PM2.5 | Y | N |
| PNH4 | ammonium | N | Y |
| PNCOM | non-carbon organic matter | N | Y |
| PFE | iron | N | Y |
| PAL | aluminum | N | Y |
| PSI | silica | N | Y |
| PTI | titanium | N | Y |
| PCA | calcium | N | Y |
| PMG | magnesium | N | Y |
| PK | potassium | N | Y |
| PMN | manganese | N | Y |
| PNA | sodium | N | Y |
| PCL | chloride | N | Y |
| PH2O | water | N | Y |
| PMOTHR | unspeciated PM2.5 | N | Y |

**Example modeling speciation profile – AE6**

**Prescribed Burning – Composite (91109)**

| pollutant | species | massfrac |
|---|---|---|
| PM2_5 | POC | 0.5019 |
| PM2_5 | PEC | 0.1093 |
| PM2_5 | PSO4 | 0.0033 |
| PM2_5 | PNO3 | 0.0107 |
| PM2_5 | PNH4 | 0.0034 |
| PM2_5 | PAL | 0.0005 |
| PM2_5 | PCA | 0.0007 |
| PM2_5 | PCL | 0.0024 |
| PM2_5 | PFE | 0.0004 |
| PM2_5 | PK | 0.0014 |
| PM2_5 | PMN | 0.0001 |
| PM2_5 | PMOTHR | 0.0125 |
| PM2_5 | PNA | 0.0014 |
| PM2_5 | PNCOM | 0.3513 |
| PM2_5 | PSI | 0.0001 |
| PM2_5 | PTI | 0.0007 |

Fig. RF3 PM$_{2.5}$ speciation- Modeling profile example

Source: International Emissions Inventory Conference. SPECIATE and using the Speciation Tool to prepare VOC and PM chemical speciation profiles for air quality modeling, p31. https://www.epa.gov/sites/default/files/2017-10/documents/speciate_speciationtool_training.pdf.

5. In Sect. 1 and Table S1, the deviations of PM$_{2.5}$ components simulated by CMAQ are presented. All these components (NH$_4^+$, NO$_3^-$, SO$_4^{2-}$, and part of OC), except for EC and part of OC, are second aerosols, and their loadings in the atmosphere are controlled primarily by the emissions of gaseous precursors, instead of the emission of aerosols. The presentation here and associated arguments seems to be misleading as the effect of uncertainties in the gaseous emissions is not considered in this study.

**Response:**

Thank you for your advices. One of the important sources of these atmospheric components ($NH_4^+$, $NO_3^-$, $SO_4^{2-}$, and OC) is formed by chemical conversion of gaseous precursors, which *are second aerosols*. But they still have some primary sources, a number of recent studies found that, primary emission may be also important. These components ($NH_4^+$, $NO_3^-$, $SO_4^{2-}$, and OC) exist in primary emission sources such as coal-fired power plant, industry process, transportation sector and residential coal combustion, the detail is shown in Fig. RF4 (Fig. 2~ 5 in manuscript); For example, sulfate (a major $PM_{2.5}$ component) is largely from primary emissions rather than secondary formation in ambient air in certain circumstances (Chen et al., 2017; Dai et al., 2019; Ding et al., 2021; Ding et al., 2019; Li et al., 2017; Yang et al., 2020; Yan et al., 2020), its weight percentage variation range is 0.7~71% in coal-fired power plant , 0.03%~40% in industry process, 0.02~40% in transportation sector, 1~40% in residential coal combustion, respectively.

[Figure]

Fig. RF4 Chemical profiles for PM₂.₅ emitted from coal-fired power plant (PP), industry processes (IN), transportation sector (TR), residential coal combustion (RE).

In our study, we found source profile variation could affect the simulation result of secondary components, **they could lever the whole aerosol equilibrium system.** The effects of source profile variation on the simulation results of different components were linked. When the percentages of Non-SNA, $SO_4^{2-}$, $NO_3^-$ and $NH_4^+$ in the source profile changed, they not only affected the simulated concentration of themselves, but also affected the simulation results of some other components through the thermodynamic equilibrium system (ISORROPIA II, $SO_4^{2-}$-$NO_3^-$-$Cl^-$-$NH_4^+$-$Na^+$-$K^+$-$Mg^{2+}$-$Ca^{2+}$-$H_2O$ system). Section 5 in our manuscript

focused on these performances: in the sensitivity tests, when we only perturb the PM$_{2.5}$ source profile (primary emission) but not the emission inventory of gaseous precursors, the simulated result of secondary PM$_{2.5}$ components also changed, this side-fact indicates the crucial role of primary PM$_{2.5}$ components on the simulation of second components formation in CTMs.

**Reference:**

Bi, X., Dai, Q., Wu, J., Zhang, Q., Zhang, W., Luo, R., Cheng, Y., Zhang, J., Wang, L., Yu, Z., Zhang, Y., Tian, Y., Feng, Y.: Characteristics of the main primary source profiles of particulate matter across China from 1987 to 2017, Atmos. Chem. Phys., 19, 3223-3243, https://doi.org/10.5194/acp-19-3223-2019, 2019.

Binkowski, F., Roselle, S. J.: Models-3 Community Multiscale Air Quality (CMAQ) model aerosol component 1. Model description, Journal of Geophysical Research: Atmospheres, 108, 4183, http://doi.org/10.1029/2001JD001409, 2003.

Byun, D., Young, J.: Governing Equations and Computational Structure of the Community Multiscale Air Quality (CMAQ) Chemical Transport Model, https://www.cmascenter.org/cmaq/science_documentation/pdf/ch06.pdf

Chapel Hill, N.: Operational Guidance for the Community Multiscale Air Quality (CMAQ) Modeling System Version 5.0, https://www.airqualitymodeling.org/index.php/CMAQ_version_5.0_(February_2010_release)_OGD#Aerosol_Module, last access: February 2012.

Chen, J., Yin, D., Zhao, Z., Kaduwela, A. P., Avise, J. C., Damassa, J. A., Beyersdorf, A., Burton, S., Ferrare, R., Herman, J. R., Kim, H., Neuman, A., Nowak, J. B., Parworth, C., Scarino, A. J., Wisthaler, A., Young, D. E., Zhang, Q.: Modeling air quality in the San Joaquin valley of California during the 2013 Discover-AQ field campaign, Atmos. Environ., 5, 100067, https://doi.org/10.1016/j.aeaoa.2020.100067, 2020.

Chen, S., Xu, L., Zhang, Y., Chen, B., Wang, X., Zhang, X., Zheng, M., Chen, J., Wang, W., Sun, Y., Fu, P., Wang, Z., Li, W.: Direct observations of organic aerosols in common wintertime hazes in North China: insights into direct emissions from Chinese residential stoves, Atoms. Chem. Phys., 17, 1259-1270, https://doi.org/10.5194/acp-17-1259-2017, 2017.

Cheng, N. L., Meng, F., Wang, J. K., Chen, Y. B., Wei, X., Han, H.: Numerical simulation of the spatial distribution and deposition of PM$_{2.5}$ in East China coastal area in 2010 (In Chinese), Journ. Safety Environ., 15, 305-310, https://doi.org/10.13637/j.issn.1009-6094.2015.06.063, 2015.

Dai, Q., Bi, X., Song, W., Li, T., Liu, B., Ding, J., Xu, J., Song, C., Yang, N., Schulze, B. C., Zhang, Y., Feng, Y., Hopke, P. K.: Residential coal combustion as a source of primary sulfate in Xi'an, China, Atmos. Environ., 196, 66-76, https://doi.org/10.1016/j.atmosenv.2018.10.002, 2019.

Ding, X., Li, Q., Wu, D., Liang, Y., Xu, X., Xie, G., Wei, Y., Sun, H., Zhu, C., Fu, H., Chen, J.: Unexpectedly Increased Particle Emissions from the Steel Industry Determined by Wet/Semidry/Dry Flue Gas Desulfurization Technologies, Environ. Sci. Technol., 53, 10361-10370, https://doi.org/10.1021/acs.est.9b03081, 2019.

Ding, X., Li, Q., Wu, D., Wang, X., Li, M., Wang, T., Chen, J.: Direct observation of sulfate explosive growth in wet plumes emitted from typical coal-fired stationary sources, Geophy. Res. Lett., 48, e2020GL092071, https://doi.org/10.1029/2020GL092071, 2021.

Foley, K. M., Roselle, S. J., Appel, K. W., Bhave, P. V., Pleim, J., Otte, T., Mathur, R., Sarwar, G., Young, J. O., Gilliam, R.: Incremental testing of the community multiscale air quality (CMAQ) modeling system version 4.7, Geosci. Model Dev., 3, 205-226, https://doi.org/10.5194/gmd-3-205-2010, 2010.

Fountoukis, C., Nenes, A.: ISORROPIA II: a computationally efficient thermodynamic equilibrium model for $K^+–Ca^{2+}–Mg^{2+}–NH_4^+–Na^+–SO_4^{2-}–NO_3^-–Cl^-–H_2O$ aerosols, Atmos. Chem. Phys., 7, 4639-4659, https://doi.org/10.5194/acp-7-4639-2007, 2007.

Friberg, M. D., Zhai, X., Holmes, H. A., Chang, H. H., Strickland, M. J., Sarnat, S. E., Tolbert, P. E., Russell, A. G., Mulholland, J. A.: Method for Fusing Observational Data and Chemical Transport Model Simulations To Estimate Spatiotemporally Resolved Ambient Air Pollution, Environ. Sci. Technol., 50, 3695-3705, https://doi.org/10.1021/acs.est.5b05134, 2016.

Gao, Y., Shan, H., Zhang, S., Sheng, L., Li, J., Zhang, J., Ma, M., Meng, H., Luo, K., Gao, H., Yao, X.: Characteristics and sources of $PM_{2.5}$ with focus on two severe pollution events in a coastal city of Qingdao, China, Chemosphere, 247, 125861, https://doi.org/10.1016/j.chemosphere.2020.125861, 2020.

Ho, K. F., Lee, S. C., Chow, J. C., Watson, J. G.: Characterization of $PM_{10}$ and $PM_{2.5}$ source profiles for fugitive dust in Hong Kong, Atmos. Environ., 37, 1023-1032, https://doi.org/10.1016/s1352-2310(02)01028-2, 2003.

Li, J., Han, Z., Li, J., Liu, R., Wu, Y., Liang, L., Zhang, R.: The formation and evolution of secondary organic aerosol during haze events in Beijing in wintertime, Sci. Total Environ., 703, 134937, https://doi.org/10.1016/j.scitotenv.2019.134937, 2020.

Li, X., Wu, J., Elser, M., Cao, J., Feng, T., El-Haddad, I., Huang, R., Tie, X., Prevot, A. S. H., Li, G.: Contributions of residential coal combustion to the air quality in Beijing-Tianjin-Hebei (BTH), China: A case study, Atmos. Chem. Phy. Discuss., http://doi.org/10.5194/acp-2017-1237, 2018.

Li, Z., Jiang, J., Ma, Z., Fajardo, O., Deng, J., Duan, L.: Influence of flue gas desulfurization (FGD) installations on emission characteristics of PM2.5 from coal-fired power plants equipped with selective catalytic reduction (SCR), Environ. Pollut., 230, 655-662, https://doi.org/10.1016/j.envpol.2017.06.103, 2017.

Liu, X. H., Zhang, Y., Olsen, K. M., Wang, W. X., Do, B. A., Bridgers, G. M.: Responses of future air quality to emission controls over North Carolina, Part I: Model evaluation for current-year simulations, Atmos. Environ., 44, 2443-2456, https://doi.org/10.1016/j.atmosenv.2010.04.002, 2010.

Ma, S., Shao, M., Zhang, Y., Dai, Q., Wang, L., Wu, J., Tian, Y., Bi, X., Feng, Y.: Evaluating the performance of chemical transport models for $PM_{2.5}$ source apportionment: An integrated application of spectral analysis and grey incidence analysis, Sci. Total Environ., 837, 155781, https://doi.org/10.1016/j.scitotenv.2022.155781, 2022.

Martin, S., Schlenker, J., Malinowski, A., Hung, H., Rudich, Y.: Crystallization of atmospheric sulfate-nitrate-ammonium particles, Geophy. Res. Lett., 30, 2102, https://doi.org/10.1029/2003GL017930,

2003.

Reff, A., Bhave, P. V., Simon, H., Pace, T. G., Pouliot, G. A., Mobley, J. D., Houyoux, M.: Emissions Inventory of PM$_{2.5}$ Trace Elements across the United States, Environ. Sci. Technol., 43, 5790-5796, http://doi.org/10.1021/es802930x, 2009.

Sha, T., Ma, X., Jia, H., Tian, R., Chang, Y., Cao, F., Zhang, Y.: Aerosol chemical component: Simulations with WRF-Chem and comparison with observations in Nanjing, Atmos. Environ., 218, 1-14, https://doi.org/10.1016/j.atmosenv.2019.116982, 2019.

Shi, Z., Li, J., Huang, L., Wang, P., Wu, L., Ying, Q., Zhang, H., Lu, L., Liu, X., Liao, H., Hu, J.: Source apportionment of fine particulate matter in China in 2013 using a source-oriented chemical transport model, Sci. Total Environ., 601-602, 1476-1487, https://doi.org/10.1016/j.scitotenv.2017.06.019, 2017.

Whitby, E., Mcmurry, P. H.: Modal aerosol dynamics modeling, Aerosol. Sci. Technol., 27, 673-688, https://www.researchgate.net/profile/Uma-Shankar-3/publication/236373518_Modal_Aerosol_Dynamics_Modeling/links/0deec52ebc352c6bc6000000/Modal-Aerosol-Dynamics-Modeling.pdf, 1997.

Xing, J., Mathur, R., Pleim, J., Hogrefe, C., Gan, C. M., Wong, D. C., Wei, C., Gilliam, R., Pouliot, G.: Observations and modeling of air quality trends over 1990–2010 across the Northern Hemisphere: China, the United States and Europe, Atmos. Chem. Phy., 15, 2723-2747, https://doi.org/10.5194/acp-15-2723-2015, 2015.

Yan, Q., Kong, S., Yan, Y., Liu, H., Wang, W., Chen, K., Yin, Y., Zheng, H., Wu, J., Yao, L., Zeng, X., Chen, Y., Zheng, S., Wu, F., Niu, Z., Zhang, Y., Zheng, M., Zhao, D., Liu, D., Qi, S.: Emission and simulation of primary fine andsubmicron particles and water-soluble ions from domestic coal combustion in China, Atmos. Environ., 224, 117308, http://doi.org/10.1016/j.atmosenv.2020.117308, 2020.

Yang, F., Liu, H., Feng, P., Li, Z., Tan, H.: Effects of wet flue gas desulfurization and wet electrostatic precipitators on emission characteristics of particulate matter and its ionic compositions from four 300 MW level ultralow coal-fired power plants, Energ. Fuel., 34, 16423-16432, https://doi.org/10.1021/acs.energyfuels.0c03222, 2020.

Zhang, Q., Xue, D., Wang, S., Wang, L., Wang, J., Ma, Y., Liu, X.: Analysis on the evolution of PM$_{2.5}$ heavy air pollution process in Qingdao (In Chinese), China Environ. Sci., 37, 3623-3635, https://doi.org/10.3969/j.issn.1000-6923.2017.10.003, 2017.

Zhang, Y., Olsen, K. M., Wang, K.: Fine Scale Modeling of Agricultural Air Quality over the Southeastern United States Using Two Air Quality Models. Part I. Application and Evaluation, Aerosol Air Qual. Res., 13, 1231-1252, https://doi.org/10.4209/aaqr.2012.12.0346, 2013.

Zheng, B., Zhang, Q., Zhang, Y., He, K. B., Wang, K., Zheng, G. J., Duan, F. K., Ma, Y. L., Kimoto, T.: Heterogeneous chemistry: a mechanism missing in current models to explain secondary inorganic aerosol formation during the January 2013 haze episode in North China, Atmos. Chem. Phys., 15, 2031-2049, https://doi.org/10.5194/acp-15-2031-2015, 2015.

---

## Referee Report (RR1)

**Review of the paper entitled "The underappreciated impact of emission source profiles on the simulation of PM2.5 components: New evidence from sensitivity analysis" by Zhongwei Luo et al.**

This article investigates the influence of source profile changes used in the chemical transport model on the simulation of PM2.5 chemical composition. The research results are convincing and have significant implications for improving the simulation effect of chemical transport models. I recommend the acceptance for publication after minor revisions. Several editorial comments for improving the information content and presentation of the paper are listed as follows.

Comments:

1. Abstract:

   The sentences in the abstract part are almost exactly the same as those in the conclusion part of the text. Please try to avoid this situation and make appropriate modifications.

   L26-29: It is unnecessary to have these sentences regarding the aims of this paper in the abstract. Please remove them.

   L32: it should be "……PM2.5 concentrations".  There are many English errors in the text part. Please correct all of them before publication.

2. Model configuration:

   You used the MEICv1.3 source emission inventory. This type of emission inventory can vary greatly from year to year, which can have a significant impact on the simulation results. Please provide additional information on which year's emission inventory was used and explain the reasons.

   L138-140:  Regarding the CMAQ, more references are needed such as (1) Eder, B., and S. Yu, 2006. A performance evaluation of the 2004 release of Models-3 CMAQ. Atmospheric Environment, 40: 4811-4824. (2) Yu, et al., 2014. Aerosol indirect effect on the grid-scale clouds in the two-way coupled WRF-CMAQ: model description, development, evaluation and regional analysis. Atmos. Chem. Phys. 14, 11247–11285, doi:10.5194/acp-14-1-2014.

3. Fig.3

   Why are there Chinese characters in the picture?

4. There are some English grammar errors such as

   L262: It should be "…p is the…".

   L266: It should be "…the value is close…".

   L337: It should be "Evaluation index for simulation results".

   L356: It should be "…the simulated results of…".

   Please correct other English grammar errors in the text.

5. Part 3 (L274~277,P13):

Please provide additional information on the specific location of the site you have chosen and explain the reason for your choice.

---

## Author Response (AR2)

**Point-by-point reply to the comments**

Dear reviewer and top editor:

We would like to thank you for the time and effort spent in reviewing the manuscript. In response, we have carefully addressed your concerns with this work. Please see point-by-point response to the comments and the revised manuscript for details. The reviewer's and top editor's comments are shown in black *italics*. Our replies are shown in indented black text.

Sincerely

Yinchang Feng and co-authors

**Content**

**Response to one anonymous Referee**

**RE1**

*As one reviewer points out, there is still need for clarification regarding the chemical equilibrium of ions in the model: "In the $PM_{2.5}$ source profiles (Sect 2.2), the chemical components (Ca, K, Mg, Mn, Na) are expressed in the form of element. In the aerosol thermodynamic process (Sect. 5), cations ($Na^+$, $Ca^{2+}$, $K^+$, $Mg^{2+}$) are used. Are all those elemental components (Ca, K, Mg, Mn, Na) from source emissions assumed to take part in the thermodynamic process in the form of cations ($Na^+$, $Ca^{2+}$, $K^+$, $Mg^{2+}$)? If so, are there any other anions used to make an ion balance with them? Are the cations and anions in equilibrium in the source profile as well as in the ISORROPIA simulation?"*

We thank the reviewer for pointing this out. To address the reviewer's comment, we added further explanation as follows:

Source profile, the physical and chemical characterization of primary sources, characterizes specific sources from a physicochemical point of view which reveals the signatures of source emissions (Bi et al., 2019). Generally, the $PM_{2.5}$ samples emitted from the sources are collected on Teflon and quartz fiber filters and then sent for chemical component analysis. Elements analysis uses Teflon filters, common chemical analysis instruments are: inductively coupled plasma optical emission spectrometer (ICP-OES), inductively coupled plasma atomic emission spectrometer (ICP-AES), inductively coupled plasma mass spectrometer (ICP-MS) instruments and X-ray fluorescence. The total carbon (TC) mass in the samples are typically determined using thermal or thermal–optical methods. There are two widely utilized approaches to dividing OC and EC from TC, known as IMPROVE_A (from the Desert Research Institute–DRI) and NIOSH (method 5040; from the National Institute for Occupational Safety and Health – NIOSH), which are operationally defined by the time–temperature protocols, and the OC–EC split point is determined by optical reflectance/transmittance

(Ho et al., 2003; Bi et al., 2019). PM samples collected on the quartz fiber filters are normally used for the determination of water-soluble inorganic ions via different types of ion chromatography (IC) with high-capacity cation-exchange and anion-exchange columns. Taking the two databases of source profiles mentioned in this paper as examples, in SPAP Database, the $PM_{2.5}$ experiment analytical items contain 20 inorganic elements, 9 water-soluble ions, OC and EC (Details could be seen in Table TE1 as follows); And in SPECIATE database, it includes bulk species ($SO_4^{2-}$, $NO_3^-$, EC, OC, $NH_4^+$, **NCOM, MO, $H_2O$, PMO**) and 37 trace elements (Na, Mg, Al, Si, P, S, Cl, K, Ca, Ti, V, Cr, Mn, Fe, Co, Ni, Cu, Zn, Ga, As, Se, Br, Rb, Sr, Zr, Mo, Pd, Ag, Cd, In, Sn, Sb, Ba, La, Ce, Hg, Pb) (Reff et al., 2009). **Source profile has been used extensively to determine the emission source by fingerprinting the traced chemical components not compounds**. **Ion equilibrium is not well considered in $PM_{2.5}$ source profile,** as some ions which are not tested or not included due to technical limits.

Table TE1 Chemical components analysis of $PM_{2.5}$ in SPAP

| Items | Analysis method | Instrument |
|---|---|---|
| Na, Mg, Al, Si, K, Ca, Ti, V, Cr, Mn, Fe, Ni, Cu, Zn, Pb, As, Cd, Co, Hg, S | ICP | Thermo iCAP 7000 |
| $SO_4^{2-}$, $NO_3^-$, $F^-$, $Cl^-$, $K^+$, $Ca^{2+}$, $Na^+$, $Mg^{2+}$, $NH_4^+$ | ICS | Thermo ICS900 |
| OC, EC | IMPROVE_A | DRI 2001A |

Source: SPAP-Database of source profiles of air pollution, State Environmental Protection Key Laboratory of Urban Ambient Air Particulate Matter Pollution Prevention and Control & Tianjin Key Laboratory of Urban, Nankai University.

In CMAQ, the aerosol module (AERO6) expands the definition of the PM Other species in earlier versions to include more detailed PM species (Chapel Hill, 2012); There are 18 $PM_{2.5}$ species in AERO6: OC, EC, $SO_4^{2-}$, $NO_3^-$, $NH_4^+$, $H_2O$, Na, Cl, NCOM, Al, Ca, Fe, Si, Ti, Mg, K, Mn, and Other; Among them, for example, Na, K, Ca, Mg, $NO_3^-$, Cl, and $SO_4^{2-}$ participate in thermodynamic process (Calculate by ISORROPIA II, a thermodynamic equilibrium model); OC participate in gas phase chemistry and POA aging; Part of Fe and Mn take part in aqueous sulfur related

reactions; Si, Al, Ti and part of Fe represent crustal matter, undergo the microphysical processes and their deposition rates are determined within the aerosol module (Chapel Hill, 2012; Appel et al., 2013).

The generic solution procedure of ISORROPIA II (thermodynamic equilibrium model) is shown in the following Fig. TE1. **Inputs needed by ISORROPIA II are the total concentrations of Na, K, Ca, Mg, NH₃, HNO₃, HCl, and H₂SO₄** together with the ambient relative humidity and temperature (Nenes et al., 1998; Fountoukis and Nenes, 2007), **not all elemental components in source profile participate in thermodynamic process**. The elemental components (Ca, K, Mg, Mn, Na) from source emissions assumed to take part in the thermodynamic process, anions (like $SO_4^{2-}$, $NO_3^-$, $Cl^-$, etc.) are used to balance with cations (Detail equilibrium relations are shown in Table TE2). The number of species and equilibrium reactions is determined by the relative abundance of each aerosol precursor (NH₃, Na, Ca, K, Mg, HNO₃, HCl, H₂SO₄) and the ambient relative humidity and temperature. The major species potentially present are determined by the value of $R_1$, $R_2$ and $R_3$. $R_1$, $R_2$ and $R_3$ are termed "total sulfate ratio", "crustal species and sodium ratio" and "crustal species ratio" respectively; $R_1$'s value is determined by molar concentration of $NH_4^+$, $Ca^{2+}$, $K^+$, $Mg^{2+}$, $Na^+$ and $SO_4^{2-}$, $R_2$ is controlled by $Ca^{2+}$, $K^+$, $Mg^{2+}$, $Na^+$ and $SO_4^{2-}$, $R_3$ is influenced by $Ca^{2+}$, $K^+$, $Mg^{2+}$ and $SO_4^{2-}$. Based on their values, aerosol composition regimes are defined. In ISORROPIA simulation, when the INPUT cations are changed, there must be some anions add in the system to balance with cations.

Our sensitivity experiment found that when the INPUT source profile (i.e. species allocation in emission sources) changed, for example, when we perturb an individual component in source profile, the influences are not only specific to this individual component, but also can be transmitted and linked among components, that is, the influence path is connected to chemical mechanisms in the model since the variation of species allocation in emission sources directly affect the thermodynamic equilibrium system (ISORROPIA II, $SO_4^{2-}$-$NO_3^-$-$Cl^-$-$NH_4^+$-$Na^+$-$K^+$-$Mg^{2+}$-$Ca^{2+}$-$H_2O$ system).

[Figure]

**Input**: RH, T, Concentrations of $NH_3$, $H_2SO_4$, Na, HCl, $HNO_3$, Ca, K, Mg

Calculate $R_1$, $R_2$, $R_3$

Determine possible major species and RH subdomain

Calculate equilibrium reaction constants for "Forward" or "Reverse" problem

Stable solution?          Metastable solution?

Solid + Liquid Aerosol          Liquid Aerosol (No solids present)

Mass conservation, activity coefficient and electroneutrality computations

Major species concentration

Calculate minor species concentration

**Output**: Equilibrium concentration of species in gas, solid and liquid phase

Fig. TE1 Generic solution procedure of ISORROPIA II

Table TE2 Equilibrium relations and $K$ used in ISORROPIA II

| Number | Reaction | $K^0$ (298.15K) |
|--------|----------|-----------------|
| I1 | $Ca(NO_3)_{2(s)} \leftrightarrow Ca^{2+}_{(aq)} + 2NO^-_{3(aq)}$ | $6.067 \times 10^5$ |
| I2 | $Ca(Cl)_{2(s)} \leftrightarrow Ca^{2+}_{(aq)} + 2Cl^-_{(aq)}$ | $7.974 \times 10^{11}$ |
| I3 | $CaSO_4 \cdot 2H_2O_{(s)} \leftrightarrow Ca^{2+}_{(aq)} + SO^{2-}_{4\,(aq)} + 2H_2O$ | $4.319 \times 10^{-5}$ |
| I4 | $K_2SO_{4(s)} \leftrightarrow 2K^+_{(aq)} + SO^{2-}_{4\,(aq)}$ | $1.569 \times 10^{-2}$ |
| I5 | $KHSO_{4(s)} \leftrightarrow K^+_{(aq)} + HSO^-_{4(aq)}$ | 24.016 |
| I6 | $KNO_{3(s)} \leftrightarrow K^+_{(aq)} + NO^-_{3(aq)}$ | 0.872 |
| I7 | $KCl_{(s)} \leftrightarrow K^+_{(aq)} + Cl^-_{(aq)}$ | 8.680 |
| I8 | $MgSO_{4(s)} \leftrightarrow Mg^{2+}_{(aq)} + SO^{2-}_{4\,(aq)}$ | $1.079 \times 10^5$ |
| I9 | $Mg(NO_3)_{2(s)} \leftrightarrow Mg^{2+}_{(aq)} + 2NO^-_{3(aq)}$ | $2.507 \times 10^{15}$ |
| I10 | $Mg(Cl)_{2(s)} \leftrightarrow Mg^{2+}_{(aq)} + 2Cl^-_{(aq)}$ | $9.557 \times 10^{21}$ |
| I11 | $HSO^-_{4(aq)} \leftrightarrow H^+_{(aq)} + SO^{2-}_{4\,(aq)}$ | $1.015 \times 10^{-2}$ |
| I12 | $NH_{3(g)} \leftrightarrow NH_{3(aq)}$ | 57.64 |

| | | |
|---|---|---|
| I13 | $NH_{3(aq)} + H_2O_{(aq)} \leftrightarrow NH_{4(aq)}^+ + OH_{(aq)}^-$ | $1.805 \times 10^{-5}$ |
| I14 | $HNO_{3(g)} \leftrightarrow H_{(aq)}^+ + NO_{3(aq)}^-$ | $2.511 \times 10^6$ |
| I15 | $HNO_{3(g)} \leftrightarrow HNO_{3(aq)}$ | $2.1 \times 10^5$ |
| I16 | $HCl_{(g)} \leftrightarrow H_{(aq)}^+ + Cl_{(aq)}^-$ | $1.971 \times 10^6$ |
| I17 | $HCl_{(g)} \leftrightarrow HCl_{(aq)}$ | $2.5 \times 10^3$ |
| I18 | $H_2O_{(aq)} \leftrightarrow H_{(aq)}^+ + OH_{(aq)}^-$ | $1.010 \times 10^{-14}$ |
| I19 | $Na_2SO_{4(s)} \leftrightarrow 2Na_{(aq)}^+ + SO_{4\ (aq)}^{2-}$ | 0.4799 |
| I20 | $(NH_4)_2SO_{4(s)} \leftrightarrow 2NH_{4(aq)}^+ + SO_{4\ (aq)}^{2-}$ | 1.817 |
| I21 | $NH_4Cl_{(s)} \leftrightarrow NH_{3(g)} + HCl_{(g)}$ | $1.086 \times 10^{-16}$ |
| I22 | $NaNO_{3(s)} \leftrightarrow Na_{(aq)}^+ + NO_{3(aq)}^-$ | 11.97 |
| I23 | $NaCl_{(s)} \leftrightarrow Na_{(aq)}^+ + Cl_{(aq)}^-$ | 37.66 |
| I24 | $NaHSO_{4(s)} \leftrightarrow Na_{(aq)}^+ + HSO_{4(aq)}^-$ | $2.413 \times 10^4$ |
| I25 | $NH_4NO_{3(s)} \leftrightarrow NH_{3(g)} + HNO_{3(g)}$ | $4.199 \times 10^{-17}$ |
| I26 | $NH_4HSO_{4(s)} \leftrightarrow NH_{4(aq)}^+ + HSO_{4(aq)}^-$ | 1.383 |
| I27 | $(NH_4)_3H(SO_4)_{2(s)} \leftrightarrow 3NH_{4(aq)}^+ + HSO_{4(aq)}^- + SO_{4\ (aq)}^{2-}$ | 29.72 |

Source: (Fountoukis and Nenes, 2007)

**Response to Topical editor Klaus Klingmüller**

TE1

*Title: It seems inappropriate to speak of an "underappreciated" impact of the source profiles. While a realistic representation of emission sources may be challenging, the importance of source profile data is certainly appreciated. The meaning of "profile" in the context of this study should be clarified in the abstract and also in the title. A possible title might be "The effect of emission source chemical profiles on simulated PM$_{2.5}$ components: sensitivity analysis with CMAQ 5.0.2".*

Thank you for your advice. We have revised the title as "The effect of emission

source chemical profiles on simulated PM$_{2.5}$ components: sensitivity analysis with CMAQ 5.0.2"

We also add the meaning of "profile" in the abstract. "Still, the emission source profiles (used to create speciated emission inventories for CTMs) of PM$_{2.5}$ has not been fully taken into account in current numerical simulation."

Detail shows as following screenshots 1-2:

**Screenshot 1:**

1
2 The effect of emission
3 source chemical profiles on simulated PM$_{2.5}$ components: sensitivity analysis
4 with CMAQ 5.0.2

**Screenshot 2:**

28 cause inaccurate simulation results. Still, the emission source profile (used to create
29 speciated emission inventories for CTMs) of PM$_{2.5}$ has not been fully taken into account
30 in current numerical simulation. This study aims to answer (1) Whether the variation of
31 source profile adopted in CTMs has an impact on the simulation of PM$_{2.5}$ chemical
32 components? (2) How much does it impact? (3) How does the impact work? Based on

**TE2**

*Line 22: The claim that "current models do not perform very well in simulating PM$_{2.5}$ components" is too general and does not reflect the literature.*

Thank you for your suggestion. we have revised the original sentence as "current models do not perform very well in reproducing the observations of some major chemical components, for example, sulfate, nitrate, ammonium and organic carbon". To address your comment, we add an extra explanation as follows:

Based on our summary of published relevant literatures, the normalized mean bias (NMB) of sulfate, nitrate, ammonium and organic carbon are highly variable and inconsistent between the simulated and the observed values to some extent (SO$_4^{2-}$:

84%~98%; $NO_3^-$: -80%~118%; $NH_4^+$: -58%~130%; OC: -73%~43%; As Fig. TE2 shows below). We have also collected some published literatures to further elaborate this conclusion, and the references are listed in Table TE3 (Table S1 in supplementary material). Detailed information has been supplemented in the introduction of the revised manuscript.

[Figure]

Fig. TE2  The normalized mean bias (NMB) between the simulated and the observed values in some literatures

Table TE3 The simulation error of CTMs on the components of $PM_{2.5}$ in different studies

| $PM_{2.5}$ components | Model | NMB | R | Study area | Period | Reference |
|---|---|---|---|---|---|---|
| $SO_4^{2-}$ | CMAQv4.7.1 | -45% | 0.73 | Eastern China | 2010 | (Cheng et al., 2015) |
| $NO_3^-$ | | 29% | 0.82 | | | |
| $SO_4^{2-}$ | CMAQv4.7.1 | -4.5% | 0.87 | Qing Dao | Jan. 2016 | (Zhang et al., 2017) |
| $NO_3^-$ | | 10% | 0.87 | | | |
| $NH_4^+$ | | -6% | 0.9 | | | |
| $SO_4^{2-}$ | CMAQv5.0.1 | -54% | 0.6 | Northern China | 2013 | (Zheng et al., 2015) |
| $NO_3^-$ | | -40% | 0.8 | | | |
| $NH_4^+$ | | -58% | 0.7 | | | |
| OC | | -25% | 0.8 | | | |
| EC | | 196% | 0.6 | | | |
| $SO_4^{2-}$ | Revised CMAQ | 6% | 0.7 | | | |
| $NO_3^-$ | | 6% | 0.8 | | | |
| $NH_4^+$ | | -4% | 0.8 | | | |
| OC | | -28% | 0.7 | | | |
| EC | | 183% | 0.6 | | | |
| $SO_4^{2-}$ | WRF-Chem3.6.1 | -84% | 0.31 | Nanjing | Jan. 2017 | (Sha et al., 2019) |
| | | -71% | 0.26 | | Apr. 2017 | |
| $NO_3^-$ | | 45% | 0.51 | | Jan. 2017 | |
| | | 67% | 0.32 | | Apr. 2017 | |

| | | | | | | |
|---|---|---|---|---|---|---|
| $NH_4^+$ | | -34% | 0.27 | | Jan. 2017 | |
| | | -13% | 0.31 | | Apr. 2017 | |
| $SO_4^{2-}$ | CMAQv5.0.2 | -41% | 0.82 | Qing Dao | Dec. 2015 ~ Jan. 2016 | (Gao et al., 2020) |
| $NO_3^-$ | | 41% | 0.83 | | | |
| $NH_4^+$ | | -5% | 0.83 | | | |
| $SO_4^{2-}$ | RAQMS | -4% | 0.83 | Beijing | Feb. to Mar. 2014 | (Li et al., 2020) |
| $NO_3^-$ | | -4% | 0.77 | | | |
| $NH_4^+$ | | 4% | 0.81 | | | |
| OC | | -39% | 0.92 | | | |
| EC | | -9% | 0.81 | | | |
| $SO_4^{2-}$ | CMAQv5.0.1 | -56%~-29% | - | China | 2013 | (Shi et al., 2017) |
| $NO_3^-$ | | -47%~19% | | | | |
| $NH_4^+$ | | -44%~1 | | | | |
| $SO_4^{2-}$ | CMAQv4.7 | -16% and -6% | - | USA | Jan. 2006 | (Foley et al., 2010) |
| | | -19%~-0.2% | | | Aug. 2006 | |
| $NO_3^-$ | | -5% and 1% | | | Jan. 2006 | |
| $NH_4^+$ | | 13% and 14% | | | Jan. 2006 | |
| | | 15% and -6% | | | Aug. 2006 | |
| OC | | -20% | | | Jan. 2006 | |
| | | -49% | | | Aug. 2006 | |
| EC | | -25% | | | Jan. 2006 | |
| | | -32% | | | Aug. 2006 | |

| | | | | | | |
|---|---|---|---|---|---|---|
| SO$_4^{2-}$ | CMAQv4.5.1 | -34%~7% | - | USA | Jan. 2002 | (Liu et al., 2010) |
| | | -18%~-37% | | | Jul. 2002 | |
| NO$_3^-$ | | 16%~118% | | | Jan. 2002 | |
| | | -69%~88% | | | Jul. 2002 | |
| NH$_4^+$ | | -0.5%~61% | | | Jan. 2002 | |
| | | -43%~53% | | | Jul. 2002 | |
| OC | | -4%~13% | | | Jan. 2002 | |
| | | -71%~-64% | | | Jul. 2002 | |
| EC | | -16%~18% | | | Jan. 2002 | |
| | | -39%~38% | | | Jul. 2002 | |
| SO$_4^{2-}$ | CMAQv4.5.1 | 5% | 0.7 | South Eastern USA | Jan. 2002 | (Zhang et al., 2013) |
| | CAMx-4.4.2 | 33% | 0.6 | | | |
| | CMAQv4.5.1 | -39% | 0.5 | | Jul. 2002 | |
| | CAMx-4.4.2 | -9% | 0.6 | | | |
| NO$_3^-$ | CMAQv4.5.1 | 46% | 0.8 | | Jan. 2002 | |
| | CAMx-4.4.2 | -21% | 0.8 | | | |
| | CMAQv4.5.1 | -62% | 0.2 | | Jul. 2002 | |
| | CAMx-4.4.2 | -80% | 0.2 | | | |
| NH$_4^+$ | CMAQv4.5.1 | -7% | 0.8 | | Jan. 2002 | |
| | CAMx-4.4.2 | -8% | 0.7 | | | |
| | CMAQv4.5.1 | -52% | 0.7 | | Jul. 2002 | |
| | CAMx-4.4.2 | -45% | 0.7 | | | |

| | | | | | | |
|---|---|---|---|---|---|---|
| OC | CMAQv4.5.1 | -15% | 0.8 | | Jan. 2002 | |
| | CAMx-4.4.2 | -18% | 0.8 | | | |
| | CMAQv4.5.1 | -73% | 0.7 | | Jul. 2002 | |
| | CAMx-4.4.2 | -47% | 0.7 | | | |
| EC | CMAQv4.5.1 | -9% | 0.7 | | Jan. 2002 | |
| | CAMx-4.4.2 | 5% | 0.7 | | | |
| | CMAQv4.5.1 | -47% | 0.4 | | Jul. 2002 | |
| | CAMx-4.4.2 | -33% | 0.4 | | | |
| $SO_4^{2-}$ | CMAQv5.0 | 0.7% and -31% | 0.85 | USA | 1990-2010 | (Xing et al., 2015) |
| | | -2% | 0.61 | Europe | | |
| $NO_3^-$ | | 56%~59% | 0.66 | USA | | |
| | | -6% | 0.70 | Europe | | |
| $NH_4^+$ | | -13% | 0.52 | USA | | |
| | | 34% | 0.62 | Europe | | |
| $SO_4^{2-}$ | CMAQv4.5 | -16% | 0.82 | USA | 2002~2008 | (Friberg et al., 2016) |
| $NO_3^-$ | | 72% | 0.64 | | | |
| $NH_4^+$ | | 13% | 0.68 | | | |
| OC | | -30% | 0.39 | | | |
| EC | | -22% | 0.5 | | | |
| $SO_4^{2-}$ | CMAQv5.0.2 | -50%~29% | - | California | 2013 | (Chen et al., 2020) |
| $NO_3^-$ | | -27%~48% | | | | |
| $NH_4^+$ | | -32%~130% | | | | |

| | | | | | | |
|---|---|---|---|---|---|---|
| OC | | -35%~13% | | | | |
| EC | | 0~43% | | | | |

**TE3**

*Line 33: You highlight that the effect of changes in the source profile on the simulated PM$_{2.5}$ components cannot be ignored as a major result. However, the composition of the emissions obviously affects the composition of the pollution (it is the exact relationship which is less obvious due to chemistry). **In addition, the percentages given in the abstract are of limited relevance as they only apply to a single site.***

Thank you for your questions.

The chemical composition of ambient PM$_{2.5}$ is influenced by both source emissions and atmospheric chemical reactions during transport. Source profile, species allocation in emission sources, is used to create speciated emission inventories for CTMs. In the reported literatures, PM$_{2.5}$ species allocation coefficients of emission sources for CTMs are commonly treated by referring to source profile data in published literature or database like the US SPECIATE. However, with the development of production technology and the innovation of pollution treatment technology in recent years, the chemical composition of PM$_{2.5}$ source emissions has changed. It is worth exploring whether the variation of source profile adopted in CTMs has a significant impact on the simulation results.

To our knowledge, this is the first study to date in response to the above issues. In our study, we separately selected source profiles from SPAPPC and SPECIATE databases and used them to create speciated emission inventories for CTMs. By designing a series of sensitivity experiments based on variations in source profile, we found the influence of source profile variation on the simulation of chemical components in PM$_{2.5}$ could not be ignored. The simulation results of some components are sensitive to the adopted source profile in CTMs. In addition, there is a linkage effect, the variation of some components in source profile would bring changes to the simulated results of other components, since the variation of species allocation in emission sources directly affect the thermodynamic equilibrium system in CTMs.

In this study, we used CMAQ (one of the most widely used CTMs), MEIC (a highresolution inventory of anthropogenic air pollutants in China), meteorological field, simulation domain and motoring sites as carriers to explore the influence of source profile changes on the simulation results. The same kind of experiment is also applicable to other CTMs, other emission source profiles, and other simulation domain.

We have rewritten this part in the abstract to make it more clearly expressed (The modified text is shown in corresponding screenshot 3 below).

**Screenshot 3:**

| | |
|---|---|
| 30 | in current numerical simulation. This study aims to answer (1) Whether the variation of |
| 31 | source profile adopted in CTMs has an impact on the simulation of PM$_{2.5}$ chemical |
| 32 | components? (2) How much does it impact? (3) How does the impact work? Based on |
| 33 | the characteristics and variation rules of chemical components in typical PM$_{2.5}$ sources, |
| 34 | different simulation scenarios were designed and the sensitivity of simulated PM$_{2.5}$ |
| 35 | components to  sources chemical profile was |
| 36 | explored. Our findings showed that the influence of source profile changes on  |
| 37 |  simulated PM$_{2.5}$ components |
| 38 | could not be ignored  |
| 39 |  Simulation results of some |
| 40 | components were sensitive to the adopted  source profile in CTMs. Moreover, |
| 41 | there was a linkage effect, the variation of some components in the source profile would |
| 42 | bring changes to the simulated results of other components. These influences are |
| 43 | connected to  chemical mechanisms of the model since the variation of species |
| 44 | allocations in emission sources  can affect potential composition |
| 45 | and phase state of aerosols, chemical reaction priority and multicomponent chemical |
| 46 | balance in thermodynamic equilibrium system. We also found that the perturbation of |

**TE4**

*The article still lacks information on how the MEIC emissions are combined with the source profiles.*

Thank you for your comments. More descriptions have been added in Table S26 of our supplementary material. Please see the detail explanation as follows:

In the database of Source Profiles of Air Pollution (SPAP) and U.S. Environmental Protection Agency's (EPA) SPECIATE database, these four source categories (coalfired power plant, industry process, transportation sector and residential coal combustion) contain a series of sub-categories. But the MEIC emission inventory does not include the corresponding sub-categories. So we take the average values of source profiles in each source category as representing source profile (Table TE4), the details could also be seen in our previous work (Bi et al., 2019); Then multiply inventory emissions by profile fraction to get emissions of specific chemical components. The general step for speciation is shown in Fig. TE3.

[Figure]

Fig. TE3 Speciation in general step

Source: International Emissions Inventory Conference. SPECIATE and using the Speciation Tool to prepare VOC and PM chemical speciation profiles for air quality modeling, p31. https://www.epa.gov/sites/default/files/2017-10/documents/speciate_speciationtool_training.pdf.

The modified text is shown in corresponding screenshots 4 below:

**Screenshot 4:**

185 other sources, collected from more than 40 cities in China since 2001. In addition to
186 inorganic elements, water-soluble ions, OC, EC and other conventional components,
187 some source profiles also encompass a series of tracer information, such as organic
188 markers, isotopes, single particle mass spectrometry, VOCs and other gaseous
189 precursors. Based on species in the aerosol chemical mechanism (AERO6) of CMAQ
190 (Appel et al., 2013; Chapel Hill, 2012), we selected 15 components in $PM_{2.5}$ source
191 profiles including Al, Ca, Cl, EC, Fe, K, Mg, Mn, Na, OC, Si, Ti, $NH_4^+$, $NO_3^-$ and $SO_4^{2-}$,
192 the remaining components are classified as "other".
193
194
195  In the database of Source Profiles of Air Pollution (SPAP) and U.S.
196 Environmental Protection Agency's (EPA) SPECIATE database, these four source
197 categories (coal-fired power plant, industry process, transportation sector and
198 residential coal combustion) contain a series of sub-categories. But the MEIC emission
199 inventory does not include the corresponding sub-categories. So we take the average
200 values of source profiles in each source category as representing source profile, the
201 details could also be seen in our previous work (Bi et al., 2019); Then multiply
202 inventory emissions by profile fraction to get emissions of specific chemical
203 components.

Table TE4   The selected information of source profile in SPECIATE and SPAPPC database

| Code | Profile Name | Controls | Profile Date | Profile Notes | Keywords | Match MEIC source[e] |
|---|---|---|---|---|---|---|
| 91041[a] | Draft Sub-Bituminous Combustion - Composite | Mixture of Baghouse, None, Electrostatic Precipitator, Wet Scrubber, Mechanical Collectors, Dry Lime Scrubber, Ammonia Injection | 2006-5-24 | Replaced by Profile 91110.  Median of Profiles 3191, 3192, 3690, 3694, and 3700. | Sub-Bituminous Coal Combustion; PM Composite | PP |
| 900162.5[b] | Industrial Manufacturing - Average | Not Applicable | 1989-1-5 | Average profile developed from original profiles representing the source category group 3xxxxxxx. | INDUSTRIAL | IN |
| 91155[c] | Residential Coal Combustion - Composite | Uncontrolled | 2009-7-12 | Median of Profiles 3761, 432012.5 | Residential Coal Combustion; Inventory speciation | RE |
| 91022[a] | Draft On-road Gasoline Exhaust - Composite | Mixture of Catalytic converter and Not available | 2006-5-24 | Replaced by Profile 91122.  Median of Profiles 311072.5, 3517, 3884, 3892, | On-road Gasoline Exhaust; PM Composite | TR |

| | | | | | |
|---|---|---|---|---|---|
| 91162[c] | LDDV Exhaust - Composite | Mixture of Catalytic converter and Not available | 2009-7-12 | 3904, 3947, 3951, 3955, 3959, and 4558. Median of Profiles 321042.5, 3912, 3963, 4675 | LDDV Exhaust; Inventory speciation |
| Local[d] | Coal combustion by power plants | Mixture of Baghouse, None, Electrostatic Precipitator, Wet Scrubber, Mechanical Collectors, Dry Lime Scrubber, | | Average of profiles power and heating power plant | PP |
| Local[d] | Industrial processes | Wet Scrubber, Dry Lime Scrubber, | | Average of profiles steel, metallurgy, cement, glass, industrial boiler | IN |
| Local[d] | Transportation sector | Mixture of Catalytic converter | | Average of profiles gasoline, diesel, gasoline-diesel exhaust | TR |
| Local[d] | Residential emission | | | Average of profiles civil boiler | RE |

a, Hsu, Ying, Randy Strait, Stephen Roe, David Holoman. 2006. 'SPECIATE 4.0 Speciation database development document - Final Report', Prepared for US EPA, RTP, NC, EPA Contract Nos. EP-D-06-001, Work Assignment Numbers 0-03 and 68-D-02-063, WA 4-04 and WA 5-05, by E.H. Pechan & Associates, Incorporation, Durham, NC. https://www.epa.gov/sites/production/files/2015-10/documents/speciatedoc_1206.pdf.

b, Shareef, G. S. Engineering Judgement, Radian Corporation. August 1987.

c, Reff, Adam, Prakash V Bhave, Heather Simon, Thompson G Pace, George A Pouliot, J David Mobley, and Marc Houyoux. 2009. 'Emissions Inventory of $PM_{2.5}$ Trace Elements across the United States', Environmental Science & Technology, 43, no. 15: 5790-96. DOI: 10.1021/es802930x.

d, Database of Source Profiles of Air Pollution (SPAP), measured by State Environmental Protection Key Laboratory of Urban Ambient Air Particulate Matter Pollution Prevention and Control & Tianjin Key Laboratory of Urban, Nankai University.

e, Coal combustion by power plants (PP), industrial processes (IN), residential emission (RE) and transportation sector (TR).

**TE5**

*Figs. 2 to 5: Please clarify that the figures present statistics of profiles from different data sources. It would be helpful to include all profiles considered - including the SPAP profiles - in Tables S3 to S11. Figs. 2 to 5 could be combined into one figure with four panels.*

Thank you for your advice. We have clarified the figures present statistics of profiles from different data source and added the SPAP profiles data in our supplementary material (Table S3 to S11), Table TE5~TE13 below. Figs. 2 to 5 are combined into one figure (Fig. TE4) in the revised manuscript. Details could also be seen as follows:

[Figure]

Fig. TE4 Chemical profiles for $PM_{2.5}$ emitted from (a) coal-fired power plant (PP), (b) industry processes (IN), (c) transportation sector (TR), (d) residential coal combustion (RE). Data obtained from SPAPPC (SPAP database and published source profiles in China) and SPECIATE (U.S. EPA SPECIATE database)

Table TE5    Power plant source profiles from published literatures in China, SPAP and SPECIATE database

| Year | $SO_4^{2-}$ | $NO_3^-$ | $Cl^-$ | $NH_4^+$ | Na | K | Mg | Ca | OC | EC | Fe | Mn | Al | Si | Ti | Other | City/Region | Data source |
|---|---|---|---|---|---|---|---|---|---|---|---|---|---|---|---|---|---|---|
| 2005~2006 | 2.9 | 0.6 | 3.4 | 0.1 | 1 | 0.3 | 0.5 | 2.5 | 34 | 4 | 2 | 0.05 | 1.7 | | 0.1 | 46.9 | Southern China | (Liu, 2007) |
| 2006 | 23 | | 0.7 | 4 | | | 0.7 | 5.5 | 2 | 0.3 | 2 | | 4.2 | | | 57.6 | Shang Hai | .(Zheng et al., 2013) |
| 2009~2013 | 0.8 | 0.2 | 0.1 | | 0.3 | 0.8 | | 1.7 | 0.3 | 1.8 | 2 | | 15.1 | 20.3 | 0.6 | 56 | Shijiazhuang | (Qi et al., 2015) |
| 2012 | 5.8 | 1.5 | 1.8 | 0.6 | 1 | 2.6 | 2 | 3 | 13 | 0.4 | 0.9 | 0.03 | 2.3 | 12.8 | 0.1 | 52.2 | Beijing | (Ma et al., 2015) |
| 2013 | 2.4 | 0.2 | | 0.03 | 2.2 | 0.2 | 0.9 | 8.8 | 4 | 0.8 | 0.8 | 0.04 | 5.7 | 7.5 | | 66.4 | Changzhou | (Teng et al., 2015) |
| 2015~2016 | 8.7 | 0.9 | 16.5 | 4.9 | 2.1 | 3.9 | 1.1 | 9.6 | 4.2 | 0.4 | 2.3 | 0.1 | 1.3 | 2.4 | 0.1 | 41.7 | Tianjin | (Bi et al., 2019) |
| 2017 | 7.8 | | 9.3 | | 0.2 | 0.1 | 0.2 | 3.6 | 1.4 | 0.1 | 1.6 | | 2.2 | 15.2 | | 58.3 | Yantai | (Wen et al., 2019) |
| | 3.2 | | 9.2 | 3.0 | 0.5 | 0.2 | 1.1 | 19.3 | 6.7 | 0.7 | 2.6 | 0.1 | 3.0 | 3.0 | 0.1 | 47 | | SPAP, Bi et al. 2019 |
| | 10.1 | 0.1 | 32.8 | 14.0 | 0.2 | 0.1 | 0.8 | 0.9 | 1.9 | 0.3 | 0.2 | 0.01 | 0.2 | 2.1 | 0.01 | 36 | | SPAP, Bi et al. 2019 |
| | 0.03 | 0.3 | 0.1 | 0.5 | 0.03 | 0.1 | 0.04 | 1.1 | 0.1 | 0.05 | 0.2 | 0.003 | 0.8 | 0.6 | 0.03 | 96 | | SPAP, Bi et al. 2019 |
| | 9.4 | 2.1 | 3.9 | 1.5 | 0.9 | 0.7 | 3.2 | 33.4 | 0.1 | 0.1 | 2.7 | 0.1 | 4.2 | 0.8 | 0.04 | 37 | | SPAP, Bi et al. 2019 |
| | 30.7 | 3.8 | 1.1 | 15.4 | 0.8 | 0.4 | 1.6 | 3.4 | 27.2 | 4.3 | 3.0 | 0.1 | 9.7 | 3.8 | 0.3 | | | SPAP, Bi et al. 2019 |
| | 26.4 | 3.7 | 1.0 | 13.6 | 1.4 | 0.6 | 3.1 | 6.4 | 23.2 | 3.5 | 3.6 | 0.1 | 15.7 | 12.1 | 0.4 | | | SPAP, Bi et al. 2019 |
| | 0.5 | 0.1 | 0.04 | 2.4 | | 0.3 | 0.3 | 1.7 | 1.1 | 1.4 | 2.9 | 0.04 | 4.8 | 4.8 | 0.3 | 79 | | SPAP, Bi et al. 2019 |
| | 0.3 | 0.02 | 0.1 | 0.05 | | 0.6 | 0.5 | 3.2 | 1.9 | 4.0 | 3.5 | 0.03 | 8.0 | 11.3 | 0.5 | 66 | | SPAP, Bi et al. 2019 |
| | 0.8 | 0.1 | 0.4 | 0.03 | | 0.5 | 0.4 | 4.8 | 2.1 | 3.8 | 3.8 | 0.1 | 6.9 | 6.2 | 0.4 | 70 | | SPAP, Bi et al. 2019 |
| | 1.7 | 0.1 | 1.5 | 0.1 | 0.1 | 0.9 | 0.8 | 6.8 | 1.9 | 3.4 | 5.8 | 0.2 | 13.8 | 11.9 | 0.7 | 50 | | SPAP, Bi et al. 2019 |
| | 0.9 | 0.04 | 0.3 | 0.03 | 0.2 | 1.1 | 1.0 | 8.0 | 2.6 | 3.3 | 7.6 | 0.2 | 14.9 | 10.1 | 0.7 | 49 | | SPAP, Bi et al. 2019 |
| | | | 0.03 | 0.2 | 0.4 | 0.6 | 0.5 | 5.3 | 7.9 | 5.1 | 2.2 | 0.03 | 7.1 | 20.3 | 0.3 | 50 | | SPAP, Bi et al. 2019 |
| | 2.7 | 0.3 | 0.4 | 0.7 | 0.6 | 0.6 | 0.6 | 5.0 | 6.1 | 10.5 | 2.9 | 0.03 | 5.1 | 14.2 | 0.3 | 50 | | SPAP, Bi et al. 2019 |
| | | | 0.03 | 0.1 | 0.7 | 0.6 | 0.9 | 12.3 | 5.5 | 8.5 | 2.6 | 0.04 | 6.5 | 13.6 | 0.3 | 48 | | SPAP, Bi et al. 2019 |
| | 1.8 | 0.3 | 0.2 | 0.2 | 0.3 | 0.4 | 0.4 | 3.4 | 2.0 | 5.8 | 2.6 | 0.04 | 7.3 | 20.0 | 0.4 | 55 | | SPAP, Bi et al. 2019 |
| | 0.1 | | 0.0 | 0.2 | 0.4 | 0.4 | 0.4 | 3.6 | 2.6 | 3.6 | 2.2 | 0.03 | 6.3 | 20.0 | 0.3 | 60 | | SPAP, Bi et al. 2019 |

| Year | | | | | | | | | | | | | | | | | Data source | Code |
|------|---|---|---|---|---|---|---|---|---|---|---|---|---|---|---|---|------|------|
| 1987 | 10.2 | | 0.1 | 0.3 | | 0.5 | | 3.5 | | 4.3 | 2.9 | 0.03 | 6 | 9.0 | 0.4 | 62.9 | Colorado | 3190 |
| 1987 | 2.1 | | 0.1 | 0.3 | | 0.5 | | 2.6 | 4.4 | 6.7 | 2.7 | 0.03 | 6.4 | 9.1 | 0.4 | 64.7 | Colorado | 3191 |
| 1987 | 18.2 | 0.1 | 0.1 | 0.4 | | 0.4 | | 4.3 | 1.9 | 1.9 | 3.1 | 0.02 | 5.5 | 8.9 | 0.5 | 54.6 | Colorado | 3192 |
| 1987 | 2.4 | | 0.1 | 0.3 | | 0.8 | | 7.2 | 2.9 | 1.2 | 4.7 | 0.06 | 9 | 12.0 | 0.5 | 58.9 | Colorado | 3194 |
| 1995 | 27.3 | 0.2 | 0.2 | 2.0 | 2.1 | 0.8 | 2.4 | 3.8 | 3.2 | 2.2 | 3.3 | 0.12 | 4.2 | 7.8 | 0.2 | 40.1 | Colorado | 3687 |
| 1995 | 15.4 | 0.4 | 1.7 | 0.2 | 0.3 | 0.1 | 0.3 | 10.0 | 1.9 | 2.5 | 0.7 | 0.01 | 1.3 | 2.3 | 0.01 | 62.9 | Colorado | 3691 |
| 1995 | 7.7 | 0.2 | 1.6 | 6.6 | 0.1 | 0.4 | 0.5 | 2.3 | 11.7 | 1.7 | 1.9 | 0.01 | 5.4 | 9.0 | 0.4 | 50.5 | Colorado | 3700 |
| 1997 | 1.5 | | 0.3 | | 0.6 | 2.0 | 1.9 | 4.0 | 8.7 | 0.4 | 1.9 | 0.03 | 19.7 | 23.9 | | 35.2 | South Africa | 3987 |
| 1999 | 10.2 | 1.6 | 3.8 | 0.3 | 3.8 | 0.6 | 1.2 | 4.3 | 70.3 | 0.01 | 0.8 | 0.03 | 1.2 | 1.9 | 0.1 | 0.03 | Texas | 4290 |
| 1999 | 71.1 | | 0.2 | 5.5 | 0.1 | 0.3 | | 5.4 | 1.0 | 0.2 | 2.3 | 0.08 | 2.8 | 8.5 | 0.5 | 2.2 | Texas | 4307 |
| 1999 | 41.5 | 0.1 | | 0.8 | 0.2 | 0.6 | | 24.8 | 3.6 | 0.9 | 4.4 | 0.3 | 2.1 | 12.5 | 1.4 | 6.7 | Texas | 4310 |
| 1999 | 4.3 | 1.3 | | 0.1 | 0.2 | 0.1 | 1.7 | 21.8 | 0.7 | | 3.7 | 0.1 | 7.4 | 7.6 | 1.0 | 50.0 | Texas | 4315 |
| 2002 | 5.7 | 1.0 | 0.3 | 0.5 | 0.4 | 0.2 | 1.3 | 16.1 | 55.7 | 2.4 | 2.9 | 0.03 | 5.6 | 6.1 | 0.6 | 1.2 | Texas | 4368 |
| 2002 | 46.2 | 0.1 | 1.1 | 5.1 | 0.1 | 0.5 | 0.1 | 11.1 | 10.3 | 0.1 | 3.7 | 0.2 | 6.5 | 13.9 | 0.8 | 0.2 | Texas | 4371 |
| 2002 | 6.4 | 0.7 | 0.0 | 0.1 | 0.2 | 0.3 | 1.5 | 18.8 | 1.5 | 1.4 | 3.5 | 0.1 | 6.8 | 9.1 | 1.0 | 48.6 | Texas | 4317 |
| 2006 | 12.7 | 0.2 | 0.07 | 0.4 | 0.1 | 0.4 | 0.5 | 3.7 | 2.6 | 1.9 | 2.7 | 0.02 | 5.4 | 8.9 | 0.4 | 60.0 | | 91041 |
| 2010 | 0.4 | | | | 3.9 | 0.3 | 0.0 | 8.5 | | | 2.0 | | 9.5 | 11.2 | 0.6 | 63.7 | Canada | 95518 |

Note:

1. The values under different components are the weight percentage in $PM_{2.5}$, %; the number under data source represents the code number in speciate_5.0_0;

2. SPAP data used in this table were deposited to the Mendeley data repository and can be freely downloaded from 
[revised manuscript text omitted]

**TE6**

*Article structure: Eq. (1) and its discussion should be part of Section 2.2. Please consider shortening the titles of Sections 3 to 5.*

Thank you for your advice. We have rewritten the section 2.2, added discussion on the Coefficient Divergence (CD) values between different source profiles, and shorten the titles of Section 3 to 5. The detail description are as follows:

The CD values of coal-fired power plant (PP), industrial process (IN), transportation sector (TR), residential coal combustion (RE) source profile between SPAPPC and SPECIATE database are 0.64±0.10 (0.34~0.92), 0.72±0.09 (0.45~0.94), 0.69±0.09 (0.33~0.86), 0.75±0.10 (0.58~0.91), respectively; The CD values between different sources are 0.78±0.10 (0.32~1.00), which show obvious differences among PM$_{2.5}$ source profiles in source category. Detailed information is shown in Fig. TE5~TE8 (Fig. S2~S6 in our supplementary material).

[Figure]

Fig. TE5 The coefficient divergence values for PP source profiles

Note: Power plant source profiles from published literatures in China (PP_L), SPAP (PP_SPA) and

SPECIATE database (PP_SPE). Numbers represent source profile sequence number.

[Figure]

Fig. TE6 The coefficient divergence values for IN source profiles.

Note: Industrial process (sintering) source profiles from published literatures (IN_Si_L) in China, SPAP (IN_Si_SPA) and SPECIATE database (IN_Si_SPE); Industrial process (iron-making) source profiles from published literatures in China (IN_Ir_L), SPAP (IN_Ir_SPA) and SPECIATE database(IN_Ir_SPE); Industrial process (steelmaking) source profiles from published literatures in China (IN_St_L), SPAP (IN_St_SPA) and SPECIATE (IN_St_SPE) database; Industrial process (Cement) source profiles from published literatures in China (IN_Ce_L), SPAP (IN_Ce_SPA) and SPECIATE database (IN_Ce_SPE). Numbers represent source profile sequence number.

[Figure]

Fig. TE7 The coefficient divergence values for TR source profiles

Note: Transportation sector (Heavy duty gasoline) source profiles from published literatures in China (TR_HG_L), SPAP (TR_HG_SPA) and SPECIATE (TR_HG_SPE) database. Transportation sector (Light diesel) source profiles from published literatures in China (TR_LD_L), SPAP (TR_LD_SPA) and SPECIATE (TR_LD_SPE) database. Transportation sector (Light duty gasoline) source profiles from published literatures in China (TR_LG_L), SPAP (TR_LG_SPA) and SPECIATE (TR_LG_SPE) database. Numbers represent source profile sequence number.

[Figure]

Fig. TE8 The coefficient divergence values for RE source profiles

Note: Residential coal combustion source profiles from published literatures in China (RE_L), SPAP (RE_SPA) and SPECIATE (RE_SPE) database. Numbers represent source profile sequence number.

We insert eq.(1) in section 2.2, which is shown in the screenshot 5 below.

**Screenshot 5:**

204     To determine the similarity between the two groups of source profiles, Coefficient

205 Divergence (CD) is calculated using the following formula (Wongphatarakul et al.,

206 1998):

207
$$CD_{jk} = \sqrt{\frac{1}{p}\sum_{i-1}^{p}\left(\frac{x_{ij} - x_{ik}}{x_{ij} + x_{ik}}\right)^2} \quad \dots\dots\dots\dots\dots\dots\dots\dots\dots \quad (1)$$

208 Where $CD_{jk}$ is the coefficient of divergence of source profile $j$ and $k$, $p$ is the

209 number of chemical components in source profile, $x_{ij}$ is the weight percentage for

210 chemical component $i$ in source profile $j$, $x_{ik}$ is the weight percentage for $i$ in source

211 profile $k$ (%). The CD value is in the range of 0 to 1, if the two source profiles are

212 similar, the value of CD is close to 0; if the two are very different, the value is close

213 to 1.

The titles of Section 3-5 are shorted as:

Section 3- Is there an impact of variation of source profile on the simulation results?

Section 4- How much does it impact?

Section 5- How does the impact work?

**TE7**

*Line 281: Specify which station location is used.*

Thank you for your comments. We have revised the original sentence as "We selected one air quality monitoring station (Site 8 as the selected station here and any site could be available) to explore the effect of emission source chemical profiles on simulated $PM_{2.5}$ components, then used the left 9 sites to further illustrate the conclusions suggested." The modified text is shown in screenshot 6 below:

**Screenshot 6:**

325  We selected one air quality monitoring station (Site 8, as the selected station
326 here and  site could be available) to explore the effect of emission source
327 chemical profiles on simulated PM$_{2.5}$ components
328
329  then used the left 9 sites to further illustrate the conclusions
330 suggested. ↵

**TE8**

*Please clearly define in the main document what the numerical values represent, i.e., where appropriate, mention that you are discussing mean values and indicate the averaging period (e.g., Fig. 6, Eq. (2) etc.).*

Thank you for your comments. The numerical values of Fig.6 and Eq. (2) are the mean values from Oct. 1 to Oct. 30 in 2018. We have clearly defined what the numerical values represent in the revised manuscript. The details are shown as following screenshots 7-9.

**Screenshot 7:**

164 data archived at National Center for Atmospheric Research (NCAR). In addition,
165 surface and upper air observations obtained from NCAR were used to further refine the
166 analysis data. The modeling was conducted from Oct. 1 to Oct.30 in 2018,  and
167 major configurations we used in CMAQ were illuminated as follows: Gas-phase

**Screenshot 8:**

331   The simulation results for PM$_{2.5}$ species under CMAQ_SPA and CMAQ_SPE

332   cases also showed big differences (as shown in Fig.  3 and Table S13

333   The largest difference in average simulated concentration was EC with CAMQ_SPE

334   giving higher by 167% than CMAQ_SPA; For OC and Mn, higher values were also

335   given by CMAQ_SPE than by CMAQ_SPA (45% and 126% on average, respectively);

336   For the  other components of concern, the simulated concentration by

337   CMAQ_SPE was lower than CMAQ_SPA with Ti (58%), Na (55%), Mg (53%), Ca

338   (51%), Al (33%), Cl (30%), K (29%), Si (22%), Fe (16%), NH$_4^+$ (9%), SO$_4^{2-}$ (9%), NO$_3^-$

339   (8%), separately. While the simulated PM$_{2.5}$ concentrations under the two cases were

340   quite close. The influence of source profile variation on the simulated PM$_{2.5}$

341   concentration was not significant, but the influence on the simulation of chemical

342   components in PM$_{2.5}$ could not be ignored.

**Screenshot 9:**

[Figure]

344

345    Fig. 6 3 The  relative concentration difference of average simulated  result

346    (PM₂.₅ and its components) between CMAQ_SPE and CAMQ_SPA (relative to CAMQ_SPA)

347    during simulation period; PM₂.₅ source profiles from SPAPPC and SPECIATE database were

348     used to create speciated

349    emission inventories for CMAQ, corresponding to case CMAQ_SPA and CMAQ_SPE, respectively.

**TE9**

*Fig. 6: Clarify the y-axis label (e.g., "Relative concentration difference")*

Thank you for your advice. We have revised Fig. 6 in the manuscript, as shown below (Fig. TE9).

[Figure]

Fig. TE9 The relative concentration difference of average simulated results (PM$_{2.5}$ and its components) between CMAQ_SPE and CAMQ_SPA (relative to CAMQ_SPA) during simulation period; PM$_{2.5}$ source profiles from SPAPPC and SPECIATE database were used to create speciated emission inventories for CMAQ, corresponding to case CMAQ_SPA and CMAQ_SPE, respectively.

**TE10**

*Fig. 7 and Table 1: Please define SNA in the captions.*

We have defined SNA in the captions, details as follows (highlighted in yellow):

[Figure]

Fig. TE10 The general roadmap of sensitivity tests (The histogram in each case were the speciation profile in CTMs; SNA represent $SO_4^{2-}$, $NO_3^-$, and $NH_4^+$, Non-SNA represent other components in $PM_{2.5}$).

Table TE14 The content of sensitivity experiment cases

| Cases | Description |
|---|---|
| Case DBL: add perturbation to Non-SNA and SNA | The percentage of all the listed components in the source profile of base case (SGL) were doubled, and the proportion of unlisted components (Other) decreased to 9%. |
| Case DBP: add perturbation to Non-SNA | The percentages of non-SNA were doubled and SNA( $SO_4^{2-}$, $NO_3^-$, $NH_4^+$) species stayed the same with that in SGL (the cumulative percentage of listed species was 85.3%), the proportion of unlisted components decreased to 14.7%. |
| Case DBS and TPS: add perturbation to $SO_4^{2-}$ | The percentage of $SO_4^{2-}$ was doubled (11%, DBS, represented Double Sulfate), tripled (16.5%, TPS, represented Triple Sulfate) and the other listed 14 species stayed the same with that in SGL (the cumulative percentage of listed species was 51% and 57%, respectively), the proportion of unlisted components decreased to 49% and 43%. |
| Case TWN and FON: add perturbation to $NO_3^-$ | The $NO_3^-$ content was raised up to 20 times (3.3%, TWN) and 40 times (6.6%, FON) of that in SGL (0.16%), the other 14 |

| | species stayed the same with SGL (the cumulative percentage of listed species was 48.6% and 51.9%, respectively), the proportion of unlisted components decreased to 51.4% and 48.1%. |
|---|---|
| Case OHA and THA: add perturbation to $NH_4^+$ | The $NH_4^+$ content was raised up to 100 times (2.2%, OHA), 200 times (4.4%, THA) of that in SGL (0.02%), the other 14 species stayed the same with SGL (the cumulative percentage of listed species was 47.7% and 49.9%, respectively), the proportion of unlisted components decreased to 52.3% and 50.1%. |

Note:

1. SNA represent $SO_4^{2-}$, $NO_3^-$, and $NH_4^+$, Non-SNA represent other components in $PM_{2.5}$.

2. The listed components contain Al, Ca, Cl, EC, Fe, K, Mg, Mn, Na, OC, Si, Ti, $NH_4^+$, $NO_3^-$ and $SO_4^{2-}$, unlisted components are classified as Other.

3. The source profiles in all cases listed in the table were calculated based on the base case SGL. In the design of simulation cases, the reason why the disturbance amplitude of $NH_4^+$ and $NO_3^-$ were significantly higher than that of other components such as $SO_4^{2-}$ and Non-SNA, was because the percentages of $NH_4^+$ and $NO_3^-$ in the base source profile (SGL, based on the chemical composition of code 000002.5 in the EPA Speciate_5.0_0 database ) were very low, while the percentage of $NH_4^+$ and $NO_3^-$ in SPAPPC exhibited in section 2.2 were orders of magnitude higher than those in SGL.

**TE11**

*Table 2: The labels "Case S1" to "Case S4" are not used elsewhere, please reconsider the labelling. The choice of the factors used to enhance the components should be discussed and motivated in the main text. **Note that to qualify as a model experiment description paper the article "should include the discussion of why particular choices were made in the experiment design"** (cf. https://www.geoscientific-model-development.net/about/manuscript_types.html).*

Thank you for your advice. We relabeled the sensitivity experiment cases to ensure consistency. To address the editor's comment, please see the point-by-point response as follows:

In Table 2, Case S1 represents add perturbation to Non-SNA (components other than $SO_4^{2-}$, $NO_3^-$, and $NH_4^+$ in $PM_{2.5}$ emission profiles), Case S2 stands for add

perturbation to $SO_4^{2-}$, Case S3-perturbation to $NO_3^-$, Case S4-add perturbation to $NH_4^+$. In order to ensure consistency, we relabeled "Case S1" as Case DBP, "Case S2" is subdivided into Case DBS and TPS, "Case S3" as Case TWN and FON, "Case S4" as OHA and THA.

The first column ( "Cases" column) in Table 2 represents the simulation cases (base case and sensitivity experiments group). The column "$R_1$", "$R_2$" and "$R_3$" represent the "total sulfate ratio", "crustal species and sodium ratio" and "crustal species ratio" respectively. $R_1$'s value is determined by molar concentration of $NH_4^+$, $Ca^{2+}$, $K^+$, $Mg^{2+}$, $Na^+$ and $SO_4^{2-}$, $R_2$ is controlled by $Ca^{2+}$, $K^+$, $Mg^{2+}$, $Na^+$ and $SO_4^{2-}$, and $R_3$ is influenced by $Ca^{2+}$, $K^+$, $Mg^{2+}$ and $SO_4^{2-}$. The last column ("Solid phase species") is the aerosol composition. In CMAQ model, the aerosol thermodynamic equilibrium process is carried out according to ISORROPIA II (thermodynamic equilibrium model), including a $SO_4^{2-}$-$NO_3^-$-$Cl^-$-$NH_4^+$-$Na^+$-$K^+$-$Mg^{2+}$-$Ca^{2+}$-$H_2O$ system. The number of species and equilibrium reactions is determined by the relative abundance of $NH_3$, Na, Ca, K, Mg, $HNO_3$, HCl, $H_2SO_4$, as well as the ambient relative humidity and temperature. Guided by the value of $R_1$, $R_2$ and $R_3$, 5 aerosol composition regimes in ISORROPIA are defined, detailed rules are shown in the following Table TE15.

Table TE15 Five aerosol types in ISORROPIA and corresponding R value

| $R_1$ | $R_2$ | $R_3$ | Aerosol type | Solid phase |
|---|---|---|---|---|
| $R_1 < 1$ | any value | any value | Sulfate Rich (free acid) | $NaHSO_4$, $NH_4HSO_4$, $KHSO_4$, $CaSO_4$ |
| $1 \leq R_1 \leq 2$ | any value | any value | Sulfate Rich | $NaHSO_4$, $NH_4HSO_4$, $Na_2SO_4$, $(NH_4)_2SO_4$, $(NH_4)_3H(SO_4)_2$, $CaSO_4$, $KHSO_4$, $K_2SO_4$, $MgSO_4$ |
| $R_1 \geq 2$ | $R_2 < 2$ | any value | Sulfate Poor, Crustal & Sodium Poor | $Na_2SO_4$, $(NH_4)_2SO_4$, $NH_4NO_3$, $NH_4Cl$, $CaSO_4$, $K_2SO_4$, $MgSO_4$ |
| $R_1 \geq 2$ | $R_2 \geq 2$ | $R_3 < 2$ | Sulfate Poor, Crustal & Sodium Rich, | $Na_2SO_4$, $NaNO_4$, $NaCl$, $NH_4NO_3$, $NH_4Cl$, $CaSO4$, $K_2SO4$, $MgSO_4$ |

| | | | Sulfate Poor, Crustal & Sodium Rich, Crustal Rich | NaNO4, NaCl, NH$_4$NO$_3$, NH$_4$Cl, CaSO$_4$, K$_2$SO$_4$, MgSO$_4$, Ca(NO$_3$)$_2$, CaCl$_2$, Mg(NO$_3$)$_2$, MgCl$_2$, KNO$_3$, KCl |
|---|---|---|---|---|
| R$_1 \geq 2$ | R$_2 \geq 2$ | R$_3 > 2$ | | |

Source: Fountoukis and Nenes, 2007

By summarizing the source profile through the published literatures and existing source profile databases, we found that the main components and their contents of different PM$_{2.5}$ sources were significantly different. Source profile, i.e. species allocation in emission sources, is used to create speciated emission inventories for CTMs. In order to explore the sensitivity of simulated PM$_{2.5}$ components to changes in source profile, different simulation scenarios were designed.

**Step1: Provide perturbation range for experiment cases based on the variation range of components' measured values**

The perturbation rules must be followed: a) perturbation on the percentage of each component in source profile fell within the variation range of its measured value described in section 2.2; b) The sum of the percentage of Non-SNA, SNA and Other components in PM$_{2.5}$ source profile was 100%. The design idea is shown in Figure TE11.

**Step2: Classify the experiment cases**

Through the pre-experiment, we found the impact pattern for SNA (SO$_4^{2-}$, NO$_3^-$, and NH$_4^+$) and Non-SNA were obviously different: When we perturb the percentage of all the components except "other" in the source profile, the simulated concentrations of Non-SNA were equal proportion change (Linear), while the simulated concentration of NO$_3^-$, SO$_4^{2-}$ and NH$_4^+$ were not equal proportion change (Non-linear). Therefore, we divided the components in the source profile into four groups (Non-SNA, SO$_4^{2-}$, NO$_3^-$, and NH$_4^+$), then sensitivity experiment of perturbation on Non-SNA, perturbation on SO$_4^{2-}$, perturbation on NO$_3^-$, and perturbation on NH$_4^+$ were determined.

**Step 3: Assess the impact and identify the influence pathway**

We try to answer (1) How much does the variation of source profile impact on the

simulation of $PM_{2.5}$ chemical components? (2) How does the impact work? We propose the sensitivity coefficient as evaluation index to quantify the impact in each sensitivity experiment. And calculate each R's value in different cases; Base on their values, the prior composition and phase state of species are determined (The major species potentially present are determined by the value of $R_1$, $R_2$ and $R_3$, Table TE15 above). Then base on the perspective of potential composition and phase state of aerosols, chemical reaction priority and multicomponent chemical balance in thermodynamic equilibrium system to explore how does the variation of source profile impact on the simulated chemical components.

[Figure]

Fig. TE11 The sketch of sensitivity experiment design idea

In order to illustrate this issue more clearly, we have made the following revision in main text and supplementary material (screenshots 10-12 below):

**Screenshot 10:**

363

Table 1 The content of sensitivity experiment cases

[revised manuscript text omitted]

523     shown in Table S27).

**TE12**

*Line 374 and Fig. 8: The negative sensitivity coefficient of NH$_4$$^+$ needs further*

*explanation. For all other stations the corresponding value is positive (Table S14). Moreover, according to lines 326, $NH_4^+$ increased while $PM_{2.5}$ did not change much, so a negative delta is surprising.*

We are extremely sorry for this **typo error**. The values of $NH_4^+$ in DBL case are negative. Line 374 and Table S14 have been corrected, and original simulation results are provided for supplementary illustration (TE16 as follow).

Table TE16 The sensitivity coefficients (δ) of simulated components in case DBL at different monitoring sites

| Components | $\delta_1$ | $\delta_2$ | $\delta_3$ | $\delta_4$ | $\delta_5$ | $\delta_6$ | $\delta_7$ | $\delta_9$ | $\delta_{10}$ |
|---|---|---|---|---|---|---|---|---|---|
| Al | 0.20 | 0.32 | 0.39 | 0.33 | 0.45 | 0.44 | 0.54 | 0.35 | 0.21 |
| Ca | 0.18 | 0.29 | 0.36 | 0.30 | 0.42 | 0.41 | 0.51 | 0.32 | 0.18 |
| Cl | 0.12 | 0.28 | 0.33 | 0.28 | 0.38 | 0.36 | 0.47 | 0.26 | 0.10 |
| EC | 0.18 | 0.29 | 0.36 | 0.30 | 0.41 | 0.41 | 0.51 | 0.32 | 0.18 |
| Fe | 0.20 | 0.32 | 0.39 | 0.33 | 0.45 | 0.44 | 0.54 | 0.35 | 0.21 |
| K | 0.18 | 0.29 | 0.36 | 0.30 | 0.42 | 0.41 | 0.51 | 0.32 | 0.18 |
| Mg | 0.18 | 0.29 | 0.36 | 0.30 | 0.42 | 0.41 | 0.51 | 0.32 | 0.18 |
| Mn | 0.20 | 0.32 | 0.39 | 0.33 | 0.45 | 0.44 | 0.54 | 0.35 | 0.21 |
| Na | 0.18 | 0.29 | 0.36 | 0.30 | 0.42 | 0.41 | 0.51 | 0.32 | 0.18 |
| OC | 0.18 | 0.30 | 0.36 | 0.30 | 0.42 | 0.41 | 0.51 | 0.32 | 0.18 |
| Si | 0.20 | 0.32 | 0.39 | 0.33 | 0.45 | 0.44 | 0.54 | 0.35 | 0.21 |
| Ti | 0.20 | 0.32 | 0.39 | 0.33 | 0.45 | 0.44 | 0.54 | 0.35 | 0.21 |
| $NH_4^+$ | -12.28 | -16.94 | -19.80 | -16.35 | -20.68 | -20.35 | -23.34 | -17.89 | -8.93 |
| $NO_3^-$ | 1.23 | 1.63 | 1.54 | 1.36 | 1.35 | 1.50 | 1.14 | 1.66 | 1.96 |
| $SO_4^{2-}$ | 0.21 | 0.32 | 0.40 | 0.33 | 0.48 | 0.48 | 0.62 | 0.39 | 0.19 |
| Other | 0.18 | 0.29 | 0.36 | 0.30 | 0.42 | 0.41 | 0.51 | 0.33 | 0.18 |

$\delta_i$ represent the sensitivity coefficients (δ) of simulated components in case DBL at monitoring site i.

**TE13**

*Fig. 8: The cyan colours are difficult to distinguish.*

We have modified the cyan color scheme to distinguish between them (Fig. TE12 below).

[Figure]

Fig. TE12 The sensitivity coefficients (δ) of simulated components to the perturbation of adopted source profile in different cases. Note: Each small color box in the figure represented the sensitivity level (indicated by the legend on the right) of PM$_{2.5}$ components (the x-coordinate) in different cases (y-coordinate). The blank grids in DBP case indicated no perturbation to SNA in PM$_{2.5}$ source profile under this case.

**TE14**

*Line 450: Please insert the Eqs. (3) to (5) after "parameters" and adjust the following sentences accordingly.*

Thank you for your advice. We have inserted the meaning of each parameter as follow:

$$R_1 = \frac{\left[ NH_4^+ \right] + \left[ Ca^{2+} \right] + \left[ K^+ \right] + \left[ Mg^{2+} \right] + \left[ Na^+ \right]}{\left[ SO_4^{2-} \right]} \quad \ldots\ldots\ldots\ldots\ldots\ldots (1)$$

$$R_2 = \frac{\left[ Ca^{2+} \right] + \left[ K^+ \right] + \left[ Mg^{2+} \right] + \left[ Na^+ \right]}{\left[ SO_4^{2-} \right]} \quad \ldots\ldots\ldots\ldots\ldots\ldots\ldots\ldots (2)$$

$$R_3 = \frac{\left[ Ca^{2+} \right] + \left[ K^+ \right] + \left[ Mg^{2+} \right]}{\left[ SO_4^{2-} \right]} \quad \ldots\ldots\ldots\ldots\ldots\ldots\ldots\ldots\ldots\ldots (3)$$

Where [X] denotes the molar concentration of component (mol m$^{-3}$), R$_1$, R$_2$ and

R$_3$ are termed "total sulfate ratio", "crustal species and sodium ratio" and "crustal species ratio" respectively; Based on their values, some aerosol composition regimes are defined (Detailed rules are defined in Table TE17. It has also been added in Table S27 of our supplementary material).

Table TE17 Five aerosol types in ISORROPIA and corresponding R value

| R$_1$ | R$_2$ | R$_3$ | Aerosol type | Solid phase |
|---|---|---|---|---|
| R$_1$<1 | any value | any value | Sulfate Rich (free acid) | NaHSO$_4$, NH$_4$HSO$_4$, KHSO$_4$, CaSO$_4$ |
| 1≤R$_1$≤2 | any value | any value | Sulfate Rich | NaHSO$_4$, NH$_4$HSO$_4$, Na$_2$SO$_4$, (NH$_4$)$_2$SO$_4$, (NH$_4$)$_3$H(SO$_4$)$_2$, CaSO$_4$, KHSO$_4$, K$_2$SO$_4$, MgSO$_4$ |
| R$_1$≥2 | R$_2$<2 | any value | Sulfate Poor, Crustal & Sodium Poor | Na$_2$SO$_4$, (NH$_4$)$_2$SO$_4$, NH$_4$NO$_3$, NH$_4$Cl, CaSO$_4$, K$_2$SO$_4$, MgSO$_4$ |
| R$_1$≥2 | R$_2$≥2 | R$_3$<2 | Sulfate Poor, Crustal & Sodium Rich, Crustal Poor | Na$_2$SO$_4$, NaNO$_4$, NaCl, NH$_4$NO$_3$, NH$_4$Cl, CaSO4, K$_2$SO4, MgSO$_4$ |
| R$_1$≥2 | R$_2$≥2 | R$_3$>2 | Sulfate Poor, Crustal & Sodium Rich, Crustal Rich | NaNO4, NaCl, NH$_4$NO$_3$, NH$_4$Cl, CaSO$_4$, K$_2$SO$_4$, MgSO$_4$, Ca(NO$_3$)$_2$, CaCl$_2$, Mg(NO$_3$)$_2$, MgCl$_2$, KNO$_3$, KCl |

Source: Fountoukis and Nenes, 2007

**TE15**

*Table 2: Please specify if these are averages. Given the variability of the simulations, it is not clear how representative these values are.*

Thank you for your advice.

R$_1$, R$_2$ and R$_3$ represent the "total sulfate ratio", "crustal species and sodium ratio" and "crustal species ratio" respectively. R$_1$'s value is determined by molar concentration of NH$_4^+$, Ca$^{2+}$, K$^+$, Mg$^{2+}$, Na$^+$ and SO$_4^{2-}$, R$_2$ is controlled by Ca$^{2+}$, K$^+$, Mg$^{2+}$, Na$^+$ and SO$_4^{2-}$, R$_3$ is influenced by Ca$^{2+}$, K$^+$, Mg$^{2+}$ and SO$_4^{2-}$ (Equation 1~3 below).

$$R_1 = \frac{\left[NH_4^+\right]+\left[Ca^{2+}\right]+\left[K^+\right]+\left[Mg^{2+}\right]+\left[Na^+\right]}{\left[SO_4^{2-}\right]} \quad \dots\dots\dots\dots\dots\dots (1)$$

$$R_2 = \frac{\left[ Ca^{2+} \right] + \left[ K^+ \right] + \left[ Mg^{2+} \right] + \left[ Na^+ \right]}{\left[ SO_4^{2-} \right]} \quad\text{................................} \quad (2)$$

$$R_3 = \frac{\left[ Ca^{2+} \right] + \left[ K^+ \right] + \left[ Mg^{2+} \right]}{\left[ SO_4^{2-} \right]} \quad\text{..............................} \quad (3)$$

Where [X] denotes molar concentration of component (mol·m$^{-3}$)

The values of $R_1$, $R_2$ and $R_3$ in Table 2 are monthly average values during Oct.1~Oct. 30 in 2018. Based on their values, the aerosol composition regimes in the model are defined. The model introduced the values of R to define the simulation subsystems and potential aerosol species, then discuss the influence pathway of source profile perturbation on simulated PM$_{2.5}$ components and linkage mechanism among components.

We also add Table TE18 below to illustrate the content of sensitivity experiment, potential aerosol species in ISORROPIA II under different cases, then to make this question clearly.

Table TE18 The content of sensitivity experiment, potential aerosol species in ISORROPIA II under different cases

| Experiment Cases | Description[3] | $R_1$ | $R_2$ | $R_3$ | Solid phase species |
|---|---|---|---|---|---|
| Case SGL | Base case | 2.53 | 2.52 | 1.9 | $CaSO_4$, $MgSO_4$, $K_2SO_4$, $Na_2SO_4$, NaCl, $NaNO_3$, $NH_4Cl$, $NH_4NO_3$ |
| Case DBL: add perturbation to Non-SNA and SNA[1] | The percentage of all the listed components in the source profile of base case (SGL) were doubled, and the proportion of unlisted components (Other)[2] decreased to 9%. | 2.53 | 2.52 | 1.9 | $CaSO_4$, $MgSO_4$, $K_2SO_4$, $Na_2SO_4$, NaCl, $NaNO_3$, $NH_4Cl$, $NH_4NO_3$ |
| Case DBP: add perturbation to Non-SNA | The percentages of non-SNA were doubled and SNA( $SO_4^{2-}$, $NO_3^-$, $NH_4^+$) species stayed the same with that in SGL (the cumulative percentage of listed species was 85.3%), the proportion of unlisted components decreased to 14.7%. | 5.04 | 5.03 | 3.79 | $CaSO_4$, $MgSO_4$, $K_2SO_4$, $CaCl_2$, $Ca(NO_3)_2$, $MgCl_2$, $Mg(NO_3)_2$, KCl, $KNO_3$, NaCl, $NaNO_3$, $NH_4Cl$, $NH_4NO_3$ |
| Case DBS: add perturbation to $SO_4^{2-}$ | The percentage of $SO_4^{2-}$ was doubled (11%, DBS, represented Double Sulfate) and the other listed 14 species stayed the same with that in SGL (the cumulative percentage of listed species was 51%), the proportion of unlisted components decreased to 49%. | 1.26 | 1.26 | 0.95 | $CaSO_4$, $MgSO_4$, $K_2SO_4$, $KHSO_4$, $Na_2SO_4$, $NaHSO_4$, $(NH_4)_2SO_4$, $NH_4HSO_4$, $(NH_4)_3H(SO_4)_2$ |
| Case TPS: add perturbation to $SO_4^{2-}$ | The percentage of $SO_4^{2-}$ was tripled (16.5%, TPS, represented Triple Sulfate) and the other listed 14 species stayed the same with that in SGL (the cumulative percentage of listed species was 57%), the proportion of unlisted components decreased to 43%. | 0.84 | 0.84 | 0.63 | $CaSO_4$, $KHSO_4$, $NaHSO_4$, $NH_4HSO_4$ |

| | | | | | |
|---|---|---|---|---|---|
| Case TWN: add perturbation to $NO_3^-$ | The $NO_3^-$ content was raised up to 20 times (3.3%, TWN) of that in SGL (0.16%), the other 14 species stayed the same with SGL (the cumulative percentage of listed species was 48.6%), the proportion of unlisted components decreased to 51.4%. | 2.53 | 2.52 | 1.9 | $CaSO_4$, $MgSO_4$, $K_2SO_4$, $Na_2SO_4$, $NaCl$, $NaNO_3$, $NH_4Cl$, $NH_4NO_3$ |
| Case FON: add perturbation to $NO_3^-$ | The $NO_3^-$ content was raised up to 40 times (6.6%, FON) of that in SGL (0.16%), the other 14 species stayed the same with SGL (the cumulative percentage of listed species was 51.9%), the proportion of unlisted components decreased to 48.1%. | 2.53 | 2.52 | 1.9 | $CaSO_4$, $MgSO_4$, $K_2SO_4$, $Na_2SO_4$, $NaCl$, $NaNO_3$, $NH_4Cl$, $NH_4NO_3$ |
| Case OHA: add perturbation to $NH_4^+$ | The $NH_4^+$ content was raised up to 100 times (2.2%, OHA) of that in SGL (0.02%), the other 14 species stayed the same with SGL (the cumulative percentage of listed species was 47.7%), the proportion of unlisted components decreased to 52.3%. | 3.58 | 2.52 | 2.95 | $CaSO_4$, $MgSO_4$, $K_2SO_4$, $CaCl_2$, $Ca(NO_3)_2$, $MgCl_2$, $Mg(NO_3)_2$, $KCl$, $KNO_3$, $NaCl$, $NaNO_3$, $NH_4Cl$, $NH_4NO_3$ |
| Case THA: add perturbation to $NH_4^+$ | The $NH_4^+$ content was raised up to 200 times (4.4%, THA) of that in SGL (0.02%), the other 14 species stayed the same with SGL (the cumulative percentage of listed species was 49.9%,), the proportion of unlisted components decreased to 50.1%. | 4.64 | 2.52 | 4.02 | $CaSO_4$, $MgSO_4$, $K_2SO_4$, $CaCl_2$, $Ca(NO_3)_2$, $MgCl_2$, $Mg(NO_3)_2$, $KCl$, $KNO_3$, $NaCl$, $NaNO_3$, $NH_4Cl$, $NH_4NO_3$ |

Note:

1. SNA represent $SO_4^{2-}$, $NO_3^-$, and $NH_4^+$, Non-SNA represent other components in $PM_{2.5}$.

2. The listed components contain Al, Ca, Cl, EC, Fe, K, Mg, Mn, Na, OC, Si, Ti, $NH_4^+$, $NO_3^-$ and $SO_4^{2-}$, unlisted components are classified as Other.

3. The source profiles in all cases listed in the table were calculated based on the base case SGL. In the design of simulation cases, the reason why the disturbance amplitude of $NH_4^+$ and $NO_3^-$ were significantly higher than that of other components such as $SO_4^{2-}$ and Non-SNA, was because the percentages of $NH_4^+$ and $NO_3^-$

in the base source profile (SGL, based on the chemical composition of code 000002.5 in the EPA Speciate_5.0_0 database ) were very low, while the percentage of $NH_4^+$ and $NO_3^-$ in SPAPPC exhibited in section 2.2 were orders of magnitude higher than those in SGL.

**TE16**

*Data availability section: Please provide all inputs and model configuration data necessary to reproduce the results as well as all output data discussed in the article in a suitable data repository* (e.g., Zenodo, see also https://www.geoscientific-model-development.net/policies/code_and_data_policy.html). *For many readers, the SPAP database is behind a language barrier, please consider other ways to make it accessible* (e.g., if the license allows, provide the relevant data in another repository).

Thank you for your advice. We have provided all inputs and model configuration data necessary in Zenodo (**https://zenodo.org/record/7865675**). We also provide the tutorial guide (English version) for better use SPAP database. Furthermore, the English version of SPAP database has already proceed but still need some time. It will be accessible in the near future for more widely readers.

The tutorial guide is shown as follows:

Step1: Register

Access the site by typing the address "http://www.nkspap.com:9091/" through Google Chrome, Microsoft Edge, or other available web browser. (Or use our test account: [Account: NKUtest; Password: NKUtest2023])

[Figure]

[Figure]

Step 2: Log in

[Figure]

[Figure]

Step 3: View, download recommend source profile

Click the Data analysis window. Then click the [Data query] or [Recommend source profile] module.

[Figure]

Besides that, this page also contains statistical analysis and similarity analysis function which can draw column, column deviation or stacked area charts.

[revised manuscript text omitted]

---

## Author Response (AR3)

**Content**

**Response to Top Editor**

Dear Top Editor:

Thank you for giving us this opportunity to revise our manuscript. We really appreciate your constructive comments and suggestions on our manuscript. We have studied the comments carefully and revised our manuscript accordingly. Please see below for a point-by-point response to your comments and concerns. The comments are shown in *black italics*. Our replies are shown in indented black text.

Sincerely

Yinchang Feng and co-authors

*Dear authors*

*To make future correspondence more efficient: Please keep your replies concise. Since the review process is about your manuscript, please refrain from providing explanations, figures or tables that are not part of the revised manuscript. Also, please do not overly cite the changes you have made to the manuscript in your response, but rather simply indicate exactly where you have changed the manuscript (or supplement and data repositories).*

*TE1    Given the central importance of the R ratios in Table 2 (page 21) for Section 5, further clarification is required. By definition R1 >= R2 >= R3, so there is something wrong with at least the values for OHA and THA. In general, it is still not clear where the numerical values come from. In your reply you mention that they are monthly averages, but not in the manuscript. Also, are the R ratios averaged or are the concentrations used to calculate the ratios? And again, you need to discuss whether the temporal variability of the three ratios is small enough that it makes sense to consider their averages and the corresponding aerosol composition regimes.*

*TE2    Page 7, line 154: As one referee pointed out, the time period you consider (October 2018) should be motivated, indicating whether choosing a different time period would affect your results.*

**Response**: Thank you very much for your valuable comments. We are very sorry for not explaining this key issue clearly. Some discussions have been added in our revised manuscript (Section 5.2 on page 23). Regarding

comments *TE1* and *TE2*, we respond together paragraph by paragraph as follows:

**(1) Where the numerical values come from? Are the R ratios averaged or are the concentrations used to calculate the ratios?**

In chemical transport models (CTMs), species concentration ($C$) is the function of time ($t$), advection and diffusion, reaction ($R$), emission ($E$) and sink, etc (Eq. 1). Most CTMs choose ISORROPIA to determine the subsystem set of equilibrium equations and solves for the equilibrium state using the chemical potential method. (Fountoukis and Nenes, 2007). ISORROPIA have two key solution procedures: determine possible major species and calculate equilibrium reaction (as shown in Fig. TE1). The inputs need by ISORROPIA are the concentrations of Na, Ca, K, Mg, $NH_3$, $HNO_3$, HCl and $H_2SO_4$; Then, based on the R values, together with ambient relative humidity and temperature, the appropriate subset of equilibrium equations (which correspond to the possible species formation priority) for the conditions specified are solved to yield the equilibrium concentrations.

[Figure]

$$\frac{\partial C_i}{\partial t} = \underbrace{-\nabla \bullet UC_i + \nabla \bullet (D\nabla \bullet C_i)}_{} + \underbrace{R_i + E_i - L_i}_{} \qquad \ldots\ldots (1)$$

Where $C_i$-Concentration of species $i$ at time $t$

[Figure]

Fig. TE1 The general solution path of ISORROPIA for each grid at time $t$

As shown in Fig. TE1, R values are calculated by the concentrations of some components and have relevance to meteorological conditions: temperature and humidity affecting ISORROPIA solving procedure, wind field affecting flux in and flux out for each grid. R's value generates at every new integration time step (e.g., the subroutine aero_subs.F module in CMAQ model). Then the corresponding species concentrations are determined by the results of pervious moment.

In the previous version of our manuscript, R values in Table 2 are initial values which related to the concentration of relevant components emit into the environment under different sensitivity tests. As mentioned above, a new R value would generate at every timestep ($dt$) iteration, so R values would

change (Fig. TE2). Subsequently, it would have an impact on potential major species. Therefore, only providing initial value would be inappropriate. For better illustration, we have replaced Table 2 with R values' distribution under base case and different sensitivity test cases (Fig TE3) in revised manuscript (Fig. 6).

[Figure]

Fig. TE2 The solution procedure at each time step

[Figure]

Fig. TE3 R values' distribution under base case and different sensitivity test cases

**(2) What role does meteorological conditions play in the mechanism of how source profile affects the simulation results of PM$_{2.5}$ components?**

Our sensitivity experiment is focus on the influence of source profile changes on the simulated PM$_{2.5}$ components. For given meteorological condition, we analyze the sensitivity of simulated components to source chemical profile by comparing the simulation results of perturbed cases with base case. Fig. 5 (The sensitivity coefficients ($\delta$) of simulated components to the perturbation of adopted source profile in different cases) in the previous version represents the average results.

For each case, the distribution of R values is related to meteorological conditions (as shown in Fig TE3). To illustrate the role of meteorological conditions in the mechanism of how source profile affects the simulated PM$_{2.5}$ components, we have supplemented some discussion in Section 5.2 of the revised manuscript.

We incorporate the hourly simulation result of temperature and humidity (affecting ISORROPIA solving procedure), wind field (affecting inflow and outflow for each grid) into K-means clustering (The sketch figure is shown in Fig. TE4), when the number of clusters is equal to or greater than 4, there is a significant inflection point (elbow) between data points and their assigned cluster centroids (Fig. TE5). Hence, 4 patterns of meteorological conditions are selected to the subsequent analysis.

[Figure]

Fig. TE4 The sketch of stratified meteorological condition

[Figure]

Fig. TE5 The elbow plot of K-mean clustering

For pattern I: (1) in the case DBL and DBP, there were more Na, K, Mg, Ca, Cl participated in aerosol chemistry, which resulted in the increase of the simulated concentration of $SO_4^{2-}$ and $NO_3^-$. The number of anions that can bind to $NH_4^+$ decreased as the concentration of metal ions increased in the system. (2) In $SO_4^{2-}$ perturbation cases (DBS and TPS); in the presence of increased concentrations of $SO_4^{2-}$, the chemical reactions would favor the formation of $NH_4HSO_4$; As a result, the simulated concentrations of $NH_4^+$ in DBS and TPS were observed to be higher compared to base case. (3) When the proportion of $NO_3^-$ in source profile increased (Case FON and TWN), the

corresponding chemical equilibrium shifted towards the utilization of $NO_3^-$, such as $NH_4^+ + NO_3^- \rightarrow NH_4NO_3$, resulting in the consumption of more $NH_4^+$ and formation of more ammonium salt. (4) In the cases of $NH_4^+$ perturbation (Case OHA and THA), the chemical equilibrium associated with $NH_4^+$ shifted towards the direction of $NH_4^+$ consumption, such as in $NH_4^+ + H^+ + SO_4^{2-} \rightarrow NH_4HSO_4$, more $SO_4^{2-}$ was consumed simultaneously. For pattern II, III and IV: They had similar rules with pattern I (Fig. TE6). When we perturb source profile (other condition unchanged), some species/reactants increase (or reduce) in the system, the chemical equilibrium shift to the direction of consuming more (or less) reactants, as shown in Fig. TE7.

[Figure]

Fig. TE6 The sensitivity coefficients ($\delta$) under different hierarchical pattern

[Figure]

Fig. TE7 The shifted direction of chemical reaction equilibrium

From a global perspective, the subdivisional sensitivity of simulated $PM_{2.5}$ components to source chemical profile under different patterns were similar; From a local view, their sensitivity levels were slightly different. For example, in pattern II, the simulated $NH_4^+$ was very sensitive to the perturbation of $SO_4^{2-}$; While in pattern I, III and VI was sensitive, but it remained the major component that underwent change (These results were also shown in Table S28 of supplementary material). Under different patterns of meteorological conditions (determining the values of R), the influence pathways of chemical source profile changed on the simulated $PM_{2.5}$ components had the same laws with general results in section 5.1 of our manuscript.

**(3) *Whether choosing a different time period would affect your results?***

In this paper, we aim to introduce a framework for evaluating how much the source profile could affect the simulation results. Our paper highlights the necessity that the representativeness and timeliness of the source profile should be paid enough attention when using CTMs for simulation. The

selected time period or location/site is a study carrier. The same kind of sensitivity  experiment designing method is also applicable to other time period and location/station.

*TE3   Page 13, Fig. 3:The percentages in the figure (and accordingly also on page 13 in lines 289f) do not agree with the values in Table S13 (for $NH_4^+$, Cl). Compared to D3_10_LOC_S1-S10.xlsx, $PM_{2.5}$ is also different. In one version of the table in D3_10_LOC_S1-S10.xlsx, a factor of 100 is missing in the values for station 8.*

**Response**: We are sorry for these typos. Figure 3 and text have been modified on page 12, lines 287-288 and 292 in the revised manuscript. We also updated the data used (add the factor of 100).

*TE4   Page 19, line 280: The referee asks for the motivation for choosing station 8 and some information about the site.*

**Response**: We selected one air quality monitoring station to explore the effect of emission source chemical profiles on simulated $PM_{2.5}$ components, then used other stations to further illustrate the conclusions suggested. In fact, any station could be available. Due to limited length of the article, the simulation results from other sites are shown in Table S14-S21 and sites information are shown in Table S12 of our supplementary material. The same kind of experiment is also applicable to any location.

*TE5-TE7* Some typos and errors

Thank you for your advice. We are sorry for these typos and errors. We

have made the corrections at corresponding position, and also checked the whole text and data again.

**TE5**  *Page 23, line 496: There is no Fig. 9*

**Response**: We have made the correction on page 22, lines 499 of our revised manuscript.

**TE6**  *Data availability*

*- The repository https://zenodo.org/record/7865675 has to be referenced in the manuscript.*

*-What are the units of the values in the files "Data used/Output/D3_10_\*/\*/\*.txt"?*

*- The tutorial for accessing the SPAP should included in the data repository.*

**Response**: We have reuploaded the data used and referenced in revised manuscript (https://zenodo.org/record/7865675);

The units of the values in the files "Data used/Output/D3_10_\*/\*/\*.txt" are $\mu g/m^3$, we have provided a note at corresponding position ("Data used/Output/D3_10_\*/unit_.txt");

We have updated Data availability part (data used and tutorial guide for accessing the SPAP). These corrections are on page 26, lines 596-597 of our revised manuscript.

**TE7**  *Supplement, Table S27, last 2 cases: "NaNO$_4$" should read "NaNO$_3$"*

**Response**: Table S27 has been corrected.

Again, thank you for your very valuable comments and suggestions. Your continuous assistance in improving the quality of our paper with patience and precise scientific ideas are highly appreciated.

**Response to Anonymous referee #3**

Dear Anonymous referee #3

Thank you very much for your helpful comments and advices. We have studied the comments carefully and revise our manuscript accordingly. Please see below for a point-by-point response to your comments and concerns. The comments are shown in *black italics*. Our replies are shown in indented black text.

Sincerely

Yinchang Feng and co-authors

*Anonymous referee #3*

*This article investigates the influence of source profile changes used in the chemical transport model on the simulation of PM$_{2.5}$ chemical composition. The research results are convincing and have significant implications for improving the simulation effect of chemical transport models. I recommend the acceptance for publication after minor revisions. Several editorial comments for improving the information content and presentation of the paper are listed as follows.*

*Comments:*

*RE1. Abstract: The sentences in the abstract part are almost exactly the same as those in the conclusion part of the text. Please try to avoid this situation and make appropriate modifications.*

*L26-29: It is unnecessary to have these sentences regarding the aims of this paper in the abstract. Please remove them.*

*L32: it should be "……PM$_{2.5}$ concentrations". There are many English errors in the text part. Please correct all of them before publication.*

**Response**: Thank you for your advice, we have removed unnecessary sentences and corrected these English errors in L32 of revised manuscript. The abstract part has also been rewritten at page 2. We also checked and corrected other grammar errors in the revised manuscript (L132, L270, L306, L309, L327, L332, L428, L430, L432-433, L435, L438).

*RE2. Model configuration: You used the MEICv1.3 source emission inventory. This type of emission inventory can vary greatly from year to year, which can have a significant impact on the simulation results. Please provide additional information on which year's emission inventory was used and explain the reasons.*

*L138-140: Regarding the CMAQ, more references are needed such as*

*(1) Eder, B., and S. Yu, 2006. A performance evaluation of the 2004 release of Models-3 CMAQ. Atmospheric Environment, 40: 4811-4824. (2) Yu, et al., 2014. Aerosol indirect effect on the grid-scale clouds in the two-way coupled WRF-CMAQ: model description, development, evaluation and regional analysis. Atmos. Chem. Phys. 14, 11247–11285, doi:10.5194/acp-14-1-2014.*

**Response**: Thank you for your advice. We have added the relevant references into our revised manuscript in L140. Anthropogenic emission data from the monthly MEIC in the year 2017 were used (Detail information also could be seen in Table S2 of supplementary material). In order to make this issue clearly, we add extra illustration as follows:

In CTMs, the $PM_{2.5}$ emission inventory is speciated in the chemical-composition dimension (Reff et al., 2009). Some commonly used emission inventories are listed in Table RE1.

Table RE1 The air pollutants in emission inventory

| Scale | Name | Air pollutants |
|---|---|---|
| Global | EDGAR[1] | $CO$, $NO_x$, NMVOC, $CH_4$; $NH_3$, $NO_x$, $SO_2$; $PM_{10}$, $PM_{2.5}$, BC, OC |

| Global | EDGAR-HTAP[2] | $SO_2$, $NO_x$, CO, NMVOC, $PM_{10}$, $PM_{2.5}$, BC, OC, $NH_3$ |
|---|---|---|
| Global | GAINS[3] | $SO_2$, $NO_X$, VOC, PM, $NH_3$, $CO_2$, $CH_4$, $N_2O$ and the F-gases |
| Reginal | MIX, MEIC[4] | $SO_2$, $NO_x$, CO, NMVOC, $PM_{10}$, $PM_{2.5}$, BC, OC, $NH_3$, and $CO_2$ |
| Reginal | NEI[5] | CO, NOx, $PM_{10}$, $PM_{2.5}$, $SO_2$, VOC, $NH_3$ |
| Reginal | REAS[6] | $SO_2$, $NO_x$, CO, NMVOC, $PM_{10}$, $PM_{2.5}$, BC, OC, $NH_3$, and $CO_2$ |
| Reginal | EMEP[7] | $SO_2$, $NO_x$, NMVOCs, $PM_{2.5}$, $NH_3$ |

Note:

1, Emissions Database for Global Atmospheric Research (EDGAR) (1970-). https://edgar.jrc.ec.europa.eu/dataset_ap61

2, The Task Force Hemispheric Transport of Air Pollution (HTAP) (2000-2010). https://jeodpp.jrc.ec.europa.eu/ftp/jrc-opendata/EDGAR/datasets/htap_v2_2/ALL/

3, Greenhouse Gas and Air Pollution Interactions and Synergies (GAINS) (1990-).https://gains.iiasa.ac.at/gains/download/GAINS-tutorial.pdf.

4, A new Asian anthropogenic emission inventory (MIX) (2008, 2010); Multi-resolution Emission Inventory for China (MEIC) (2008-). http://meicmodel.org/

5, National emission inventory (NEI) (1970-), https://www.epa.gov/air-emissions-inventories/national-emissions-inventory-nei

6, Regional Emission inventory in Asia (REAS) (1950-2015). https://www.nies.go.jp/REAS/index.html#REASv3.2.1

7, European Monitoring and Evaluation Programme (EMEP) (1990-), https://www.eea.europa.eu/data-and-maps/dashboards/national-air-pollutant-emissions-data

As total $PM_{2.5}$ need to be speciated into its chemical components to match the chemical mechanism in CTMs, emission source profiles, which can provide "species" and "split factor" for $PM_{2.5}$, are key inputs for creating chemically-resolved emission inventories for CTMs. However, the actual emission source profile of $PM_{2.5}$ and the sensitivity of simulated components' concentrations to the variation in $PM_{2.5}$ source profiles are currently not well considered.

In this paper, we aim to introduce a framework for evaluating how much the source profile affect the simulation result. When we perturb source profile (other condition unchanged), some species/reactants increase (or reduce) in the system, the chemical equilibrium shift to the direction of consuming more (or less) reactants, as shown in Fig. RE1 below. Our paper

highlights the necessity that the representativeness and timeliness of the source profile should be paid enough attention when using CTMs for simulation. The selected emission inventory is a study carrier. The same kind of experiment is also applicable to other emission inventories (e.g. NEI, EEI, REAS, HATP, etc.).

[Figure]

Fig. RE1 The shift direction of chemical reaction equilibrium

*RE3. Fig.3 Why are there Chinese characters in the picture?*

**Response**: We are sorry for this error and make the correction in Line 217, and combine four pictures into one.

[Figure]

*RE4. There are some English grammar errors*

*such as*

*L262: It should be "…p is the…".*

*L266: It should be "…the value is close…".*

*L337: It should be "Evaluation index for simulation results".*

*L356: It should be "…the simulated results of…".*

*Please correct other English grammar errors in the text.*

**Response**: Thank you for your suggestion. We have revised these grammar errors (in L190, L194, L337, L355). We also check the whole text and corrected other grammar errors in the article (L132, L270, L306, L309, L327, L332, L428, L430, L432-433, L435, L438).

*RE5. Part 3 (L274~277,P13): Please provide additional information on the specific location of the site you have chosen and explain the reason for your choice*

**Response**: Thank you for your comments. We selected one air quality monitoring station to explore the effect of emission source chemical profiles on simulated $PM_{2.5}$ components, then used other stations to further illustrate the conclusions suggested. In fact, any station could be available. Due to the limited length of the article, the simulation results from other sites are shown in Table S14-S21 and sites information are shown in Table S12 of our supplementary material. The same kind of experiment is also applicable to any location.

Again, we are grateful for your insightful comments and suggestions, thank you for your expertise, attention to detail, and for helping us improve this paper.

**Reference**

Fountoukis, C., Nenes, A.: ISORROPIA II: a computationally efficient thermodynamic equilibrium model for $K^+$–$Ca^{2+}$–$Mg^{2+}$–$NH_4^+$–$Na^+$–$SO_4^{2-}$–$NO_3^-$–$Cl^-$–$H_2O$ aerosols, Atmos. Chem. Phys., 7, 4639-4659, https://doi.org/10.5194/acp-7-4639-2007, 2007.

Reff, A., Bhave, P. V., Simon, H., Pace, T. G., Pouliot, G. A., Mobley, J. D., Houyoux, M.: Emissions Inventory of $PM_{2.5}$ Trace Elements across the United States, Environ. Sci. Technol., 43, 5790-5796, http://doi.org/10.1021/es802930x, 2009.

---

## Author Response (AR4)

Dear Top Editor Klaus Klingmüller:

Thank you very much for your kind work and consideration on publication of our manuscript. We really appreciate your constructive comments and suggestions for helping us improve this paper. Please see below for a response to some necessary corrections. The comments are shown in *black italics*. Our replies are shown in indented black text.

Sincerely

Yinchang Feng and co-authors

*1 The input datasets for WRF have a DOI which should be used instead of the link in the Data availability section, i.e., https://rda.ucar.edu/datasets/ds351.0/index.html should be replaced by https://doi.org/10.5065/39C5-Z211. Also, please adjust the link to the MEIC dataset, the current link leads to the citation rules, not to the data itself.*

**Response**: Thank you for your suggestion. We have replaced the link to the input datasets of WRF with a DOI. Also, the emission data (MEIC dataset) used in this paper are available at https://zenodo.org/record/7865675. These corrections are on page 26, lines 589 and 595-596 of our revised manuscript and Table S2 of *Supplementary Material* (ds083.2: https://doi.org/10.5065/D6M043C6; ds351.0: https://doi.org/10.5065/39C5-Z211; ds461.0: https://doi.org/10.5065/4F4P-E398).

*2 Regarding the figures 1, S2, S3, S4, S5: please ensure that the colour schemes used in your maps and charts allow readers with colour vision deficiencies to correctly interpret your findings. Please check your figures using the Coblis – Color Blindness Simulator (https://www.color-blindness.com/coblis-color-blindness-simulator/) and revise the colour schemes accordingly.*

**Response**: Thank you for your advice. We have revised the color schemes (Figures 1, S2, S3, S4, S5) accordingly. And sorry for not considering this important factor.